# BHASHABENCH V1: A COMPREHENSIVE BENCHMARK FOR THE QUADRANT OF INDIC DOMAINS

## ABSTRACT

The rapid advancement of large language models (LLMs) has intensified the need for domain and culture specific evaluation. Existing benchmarks are largely Anglocentric and domain-agnostic, limiting their applicability to India-centric contexts. To address this gap, we introduce **BhashaBench V1**, the first domain-specific, multi-task, bilingual benchmark focusing on critical Indic knowledge systems. BhashaBench V1 contains **74,166** meticulously curated question-answer pairs, with 52,494 in English and 21,672 in Hindi, sourced from authentic government and domain-specific exams. It spans four major domains: Agriculture, Legal, Finance, and Ayurveda, comprising 90+ subdomains and covering 500+ topics, enabling fine-grained evaluation. Evaluation of 29+ LLMs reveals significant domain and language specific performance gaps, with especially large disparities in low-resource domains. For instance, GPT-4o achieves 76.49% overall accuracy in Legal but only 59.74% in Ayurveda. Models consistently perform better on English content compared to Hindi across all domains. Subdomain-level analysis shows that areas such as *Cyber Law*, *International Finance* perform relatively well, while *Panchakarma*, *Seed Science*, and *Human Rights* remain notably weak. **BhashaBench V1** provides a comprehensive dataset for evaluating large language models across India's diverse knowledge domains. It enables assessment of models' ability to integrate domain-specific knowledge with bilingual understanding. All code, benchmarks, and resources are publicly available to support open research.

## 1 INTRODUCTION

The rapid advancement of large language models (LLMs) has transformed artificial intelligence, extending their capabilities far beyond traditional natural language processing. Models such as GPT-4o (OpenAI, 2024), GPT-OSS-120B (OpenAI et al., 2025), DeepSeek-V3 (DeepSeek-AI et al., 2025), and Qwen-3 (Yang et al., 2025) excel across diverse domains, from code generation and mathematical reasoning to creative writing and scientific analysis (Brown et al., 2020; Touvron et al., 2023; OpenAI et al., 2024), enabling applications in conversational AI, education, healthcare, finance, legal services, and agriculture (Bubeck et al., 2023; Wei et al., 2022). Platforms like *Krishi Sathi (Vijayvargia et al., 2025)* leverage LLMs for crop advisory and pest detection, improving agricultural productivity. Despite these advances, substantial performance gaps remain in multilingual and domain-specific contexts, particularly for non-Latin, low-resource languages (Wang et al., 2024a; Zhong et al., 2025; Ahuja et al., 2024). English-centric training limits models' ability to capture nuanced knowledge in specialized fields and India-specific domains, such as Ayurveda, indigenous agriculture, finance, and regional legal systems (Winata et al., 2021; Sen & et al., 2023; Khanuja et al., 2021a), highlighting the need for culturally and contextually aware evaluation. The scale of this problem demands urgent attention, as India's diverse knowledge ecosystem affects millions of lives across multiple critical domains. In agriculture alone, over 40 million farmers rely on farming-related activities (IndiaDataMap, 2025), and access to accurate information on crop management, soil health, and sustainable practices can have a direct impact on food security and livelihoods. The complexity is further magnified by the fact that each state in India has its own distinct agricultural methods, crop varieties, soil conditions, and traditional farming practices that have evolved over centuries to suit local climatic and geographical conditions. Similarly, India's legal system processes millions of cases annually, requiring precise understanding of complex legal frameworks,

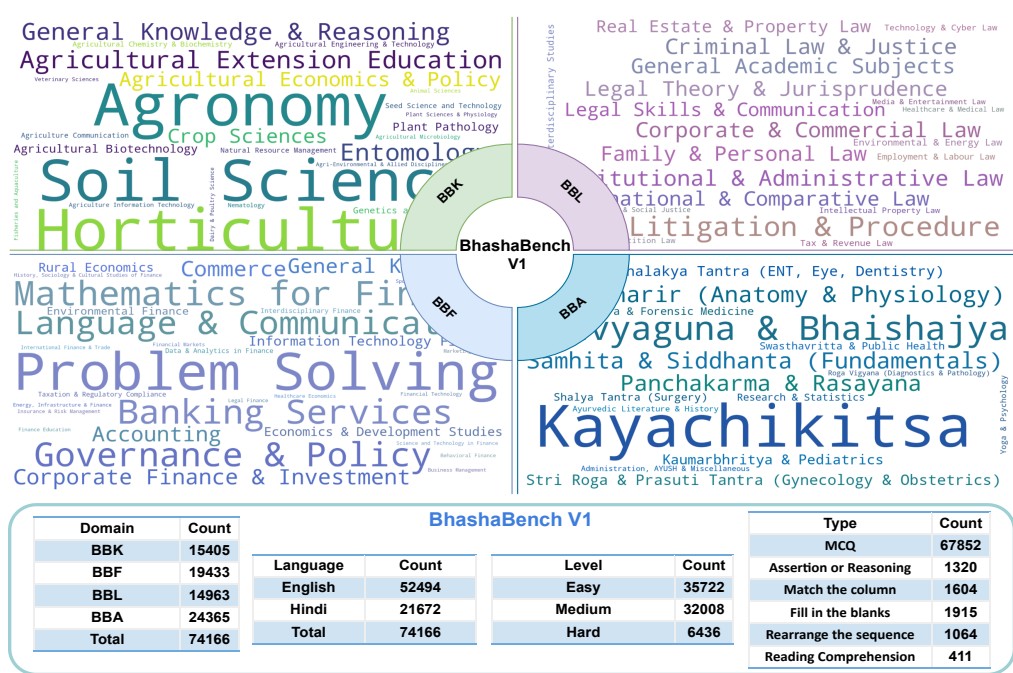

Figure 1: Overview diagram and statistics of BhashaBench V1.

precedents, and procedural nuances that vary across states and jurisdictions (National Judicial Data Grid (NJDG), India, 2023). The healthcare sector, particularly traditional medicine systems like Ayurveda, serves millions of patients who rely on practitioners' knowledge of ancient texts, formulations, and treatment protocols. Furthermore, India's financial ecosystem processes billions of transactions daily, including over 100 billion UPI transactions annually, where even minor misunderstandings in financial regulations or procedures can have cascading effects (National Payments Corporation of India, 2023). While existing benchmarks such as MMLU (Hendrycks et al., 2021a), HellaSwag (Zellers et al., 2019), AGIEVAL (Zhong et al., 2023), and more recent multilingual efforts like MEGA (Ahuja et al., 2024) attempt to assess model capabilities, they often focus primarily on English content and may not fully capture India-specific nuances, cultural contexts, and domain expertise that are essential for real-world applications in the Indian subcontinent.

To address these critical gaps, we introduce **BhashaBench V1**, the first comprehensive domain-specific, multi-task, bilingual benchmark designed explicitly for evaluating large language models on India-centric knowledge systems. BhashaBench V1 encompasses four fundamental domains that form the backbone of Indian society and economy: Agriculture (BBK - BhashaBench Krishi), Legal (BBL - BhashaBench Legal), Finance (BBF - BhashaBench Finance), and Ayurveda (BBA - BhashaBench Ayurveda). The benchmark spans over 90 subdomains and covers more than 500 specific topics, reflecting the intricate complexity and diversity of Indian knowledge systems. This granular categorization enables fine-grained evaluation of model performance across specialized areas that require deep domain expertise and cultural understanding. The dataset has been meticulously curated from over 40 authentic government and professional examination papers, ensuring that the questions reflect real-world scenarios and ground-level challenges faced by practitioners in these domains (Indian Knowledge Systems Division, Ministry of Education, Government of India, 2025; Zhong & Goodfellow, 2024). To maximize coverage across India's linguistic landscape, BhashaBench V1 currently supports English and Hindi, the two most widely understood languages in the country, collectively enabling assessment of models' capabilities for a significant portion of India's population while maintaining the cultural and contextual authenticity of the original knowledge systems.

Our comprehensive evaluation of 29+ state-of-the-art language models on BhashaBench V1 reveals significant performance disparities across domains and languages, highlighting the urgent need for India-specific model development and evaluation. The results demonstrate substantial domain-

specific performance gaps, with models showing varying degrees of competency across different knowledge areas. For instance, GPT-4o, one of the top-performing models, achieved 76.49% accuracy in the Legal domain but only 59.74% in Ayurveda, illustrating the challenges models face with traditional Indian knowledge systems. Similarly, consistent language-specific performance gaps emerged, with models generally performing better on English content compared to Hindi across all domains. The subdomain-level analysis further reveals granular insights into model capabilities, showing that certain areas such as Cyber Law and International Finance demonstrate relatively strong performance, while traditional domains like Panchakarma, Seed Science, and Human Rights remain notably challenging for current LLMs. These findings underscore the critical importance of domain and language-specific evaluation frameworks for assessing model readiness for real-world deployment in diverse Indian contexts.

## 2 RELATED WORK

### 2.1 EXPLORATION OF LLMS

The landscape of large language models has witnessed unprecedented growth, with both proprietary and open-source models achieving remarkable capabilities. Recent proprietary LLMs, including GPT-4o and GPT-4o-mini (OpenAI et al., 2024), Claude-3.5 Sonnet (Anthropic, 2024), and the Gemini series (Google, 2023), have demonstrated significant improvements across various benchmarks (Chiang et al., 2024; Wang et al., 2024c). The open-source ecosystem has flourished with models such as the Llama-3 series (Grattafiori et al., 2024), Gemma (Team et al., 2024), Qwen2.5 (Qwen et al., 2025), and Mistral (Jiang et al., 2023) achieving competitive performance while maintaining transparency and accessibility.

While primarily trained on English-dominant corpora, many models incorporate substantial multilingual data during pretraining (Team et al., 2024; Grattafiori et al., 2024; Üstün et al., 2024), enabling capabilities in hundreds of languages with varying proficiency (Nguyen et al., 2024). Language-specific models have gained momentum, particularly for underrepresented languages including Indic languages (Gala et al., 2024; 2023). Notable examples include Airavata (Gala et al., 2024), MuRIL (Khanuja et al., 2021b), and recent generative models like Param-1 (Pundalik et al., 2025).

Domain-specific language models have emerged as a critical research direction. Medical applications include Med-PaLM (Singhal et al., 2023) and BioBERT (Lee et al., 2019), while legal and financial domains have seen LegalBERT (Chalkidis et al., 2020) and FinBERT (Yang et al., 2020) respectively. In the Indian context, domain-specific initiatives like Agri-Param (BharatGenAI, 2025a), Ayur-Param (BharatGenAI, 2025b), Finance-Param (BharatGenAI, 2025c), and Legal-Param (BharatGenAI, 2025d) address unique requirements of India's diverse knowledge systems through continual pretraining (Nag et al., 2024) or instruction fine-tuning (Aralimatti et al., 2025).

Despite these advances, comprehensive evaluation frameworks for culturally and linguistically diverse domains remain limited, particularly for traditional knowledge systems requiring nuanced understanding of local contexts. This work conducts a comprehensive evaluation of 29+ state-of-the-art models on BhashaBench V1 to address these evaluation challenges.

### 2.2 EVALUATION OF LLMS

Numerous benchmarks have been developed to assess large language model performance. General-purpose benchmarks such as ARC-C (Clark et al., 2018), MMLU (Hendrycks et al., 2021b), MMLU-Pro (Wang et al., 2024b), AGIEval (Zhong et al., 2023), BIG-Bench (Srivastava et al., 2023), and HellaSwag (Zellers et al., 2019) evaluate LLMs across diverse tasks from commonsense reasoning to knowledge-intensive question answering. However, these remain largely Anglocentric with limited multilingual evaluation (Bandarkar et al., 2024; Kakwani et al., 2020). Indic-focused resources such as MILU (Verma et al., 2025) expand linguistic diversity by benchmarking 11 Indian languages, though they largely center on general-domain multiple-choice tasks. Similarly, Sanskriti (Maji et al., 2025) offers culturally grounded evaluation rooted in Indian history and heritage but lacks broad domain and multilingual coverage.

Table 1: Overview of how BhashaBench V1 compares to Indic, multilingual, and other domain-specific evaluation benchmarks.

| Benchmark | Languages | Domains | Task Formats | Cultural | Size |
|---|---|---|---|---|---|
| MMLU | En | 57 general | MCQ | ✗ | 15.9K |
| MMLU-Indic | 11 Indic | 57 general | MCQ | ✗ | 15.9K |
| MILU | 11 Indic | 8 general | MCQ | ∼ | 79.6K |
| Sanskriti | En | Culture, History | MCQ, QA | ✓ | 21.8K |
| IndicQA | 11 Indic | General KG | QA | ∼ | 50K per language |
| AgXQA 1.1 | En | Agriculture | QA | ✗ | 1.5K |
| MultiFin | En | Finance | Classification | ✗ | 10K |
| IL-TUR | 10 Indic | Legal | QA, Generation, Classification | ∼ | Multi-size (per task) |
| **BhashaBench V1** | **En + Hi** | **Agri, Legal Finance, Ayurveda** | **MCQ, A/R, FIB MTC, RC, RTS** | ✓ | **74.2K** |

**Abbreviations:** MCQ = Multiple Choice Questions; FIB = Fill in the Blanks; A/R = Assertion/Reason; RC = Reading Comprehension; QA = Question Answering; MTC = Match the Columns; KG = Knowledge Graph.

**Cultural Authenticity Legend:** ✓= sourced from Indic region-specific exams or created by native domain experts; ∼ = partially translated or culturally adapted; ✗= mainly sourced from non-Indic sources.

To address domain-specific challenges, specialized benchmarks have emerged. In agriculture, benchmarks like AgriBench (Zhou & Ryo, 2024), BVL QA Corpus (AnhaltAI, 2024), AgXQA (Kpodo et al., 2024), AgEval (Arshad et al., 2025), and SeedBench (Ying et al., 2025) cover crop disease identification to advisory support. The finance domain features FinGAIA (Zeng et al., 2025), FinanceBench (Islam et al., 2023), MultiFin (Jørgensen et al., 2023), InvestorBench (Li et al., 2024), and MultiFinBen (Peng et al., 2025) for financial reasoning, fraud detection, and trading evaluation. Legal domain efforts include IL-TUR (Joshi et al., 2024), IndicLegalQA (Nigam et al., 2023), Legal-Bench (Guha et al., 2023), LEXTREME (Niklaus et al., 2023), and the CAIL series (Xiao et al., 2018; 2019) for legal question answering, case summarization, and judgment prediction. Traditional medicine resources such as MTCMB (Kong et al., 2025), Pratyaya-Kosh (Ragad & Gokhale, 2019), Anveshana (Terdalkar et al., 2023), and OpenTCM (He et al., 2025) provide task-specific evaluation datasets covering knowledge graphs, OCR correction, and dosha analysis.

Despite this progress, key limitations persist. Many benchmarks are restricted to English or high-resource languages, limiting effectiveness for multilingual and Indic contexts. Others focus on narrow tasks, unable to capture full domain expertise. Evaluation methodologies vary widely from accuracy scores to human judgments, hindering standardized comparison across domains and languages. These gaps underscore the need for a unified, multilingual, and domain-aware evaluations.

## 3 BHASHABENCH V1

### 3.1 DESIGN PRINCIPLES

The primary motivation behind BhashaBench V1 is to comprehensively assess domain-specific knowledge and reasoning capabilities of large language models within India's diverse and culturally rich knowledge ecosystems. Unlike existing benchmarks focusing on general or Western-centric domains, our benchmark evaluates specialized Indian knowledge systems requiring deep cultural understanding and contextual awareness. Table 1 illustrates how BhashaBench V1 addresses critical gaps in existing evaluation frameworks. While benchmarks like MMLU and Indic MMLU provide broad coverage, they lack cultural grounding and rely on translated content. Domain-specific benchmarks typically focus on narrow tasks within single domains. BhashaBench V1 uniquely combines cultural authenticity, domain specialization, and bilingual support through exam-sourced questions.

BhashaBench V1 adheres to seven core design principles: **(1) Critical Indian Domains:** Encompasses Agriculture, Legal systems, Finance, and Ayurveda with fine-grained subfields. **(2) Diverse Task Formats:** Includes multiple-choice, assertion-reasoning, fill-in-blanks, and comprehension tasks. **(3) India-Specific Reasoning:** Evaluates domain-specific reasoning incorporating cultural contexts and regional practices. **(4) Bilingual Framework:** Supports English and Hindi evaluation maintaining cultural authenticity. **(5) Authentic Sources:** Questions curated from government examinations and professional certifications. **(6) Difficulty Assessment:** Categorized into Easy,

Medium, Hard levels. **(7) Cultural Authenticity:** Prioritizes traditional knowledge systems including Ayurvedic principles. This framework spans 90+ subdomains covering 500+ topics, enabling comprehensive evaluation of model capabilities in India-centric contexts.[1]

## 3.2 DATA COLLECTION

The data collection process for BhashaBench V1 follows a systematic approach similar to AGIEVAL (Zhong et al., 2023), focusing on authentic examination materials from national and state-level assessments. We systematically gathered publicly available question papers from official online examination portals, which host previously released papers that are manually curated by subject matter experts, ensuring accurate topic tagging, language annotation, and validated answer keys.

Our comprehensive collection encompasses over 40 different examination types across multiple categories: national competitive exams, domain-specific degree examinations, professional certification tests, and state-level civil services examinations. Regional state examinations proved particularly valuable as they incorporate state-specific topics, local knowledge systems, and cultural practices often overlooked in national assessments. These examinations are typically taken by individuals seeking higher education opportunities or career advancement, ensuring questions reflect practical, real-world knowledge requirements.

The final dataset comprises **74,166** carefully curated question–answer pairs spanning four core domains, with **52,494 questions in English** (70.8%) and **21,672 questions in Hindi** (29.2%), reflecting practical usage patterns in Indian educational and professional contexts. This approach ensures BhashaBench V1 captures the nuanced intersection between language, culture, and domain expertise essential for effective model deployment in Indian contexts.

## 3.3 DATA PROCESSING

Our data processing phase focused on extracting structured question-answer pairs from PDF examination papers while preserving linguistic and formatting nuances essential for authentic evaluation. Most examination materials were available exclusively in PDF format, requiring sophisticated OCR processing pipelines to handle multilingual content and domain-specific terminology.

**OCR Pipeline Selection:** Based on existing evaluations (Paruchuri & Team, 2024), Surya OCR demonstrated superior performance in handling Indic languages and domain-specific content. Reported results show 98.1% normalized text similarity for English and 98.9% for Hindi, with an average of 97.8%, outperforming alternatives such as Tesseract (88.0% overall) and Google Vision API (96.7%). Surya's architecture, designed for multilingual document understanding with enhanced Indic script support, makes it a suitable choice for diverse examination materials.

**Question-Answer Extraction Pipeline:** Following OCR processing, we developed an extraction pipeline leveraging GPT-OSS-120B (OpenAI, 2024) to structure raw text into formatted question-answer pairs. Key challenges included format variations across examination bodies, answer key alignment, multi-format questions (MCQ, assertion-reasoning, comprehension), and language-specific formatting conventions. The pipeline included: (1) **Question Extraction** using GPT-OSS-120B for boundary detection across different layouts; (2) **Option Parsing** to maintain original labeling conventions; (3) **Answer Key Alignment** processing both inline and separate answer documents; and (4) **Format Standardization** into consistent JSON structure with domain metadata.

**Data Cleaning and Quality Control:** Our multi-layered cleaning approach addressed noise and inconsistencies through systematic filtering. We excluded image-based questions, and questions with more than four options. Language verification used INDICLID (Madhani et al., 2023) and Unicode-based filtering (Khan et al., 2024) for proper linguistic categorization. Approximately 30% of questions lacked subdomain classification, addressed using GPT-OSS-120B with domain-specific taxonomies. We classified questions into six categories: MCQ, assertion-reasoning, fill-in-the-blanks, match-the-column, reading comprehension, and sequence rearrangement. Duplicate detection employed both exact-match and semantic similarity measures.

**Manual Validation:** Following a methodology similar to (Bandarkar et al., 2024), all extracted question-answer pairs underwent rigorous expert validation to ensure accuracy verification, cultural

---

[1]More collection and processing procedures can be found in Appendix C.

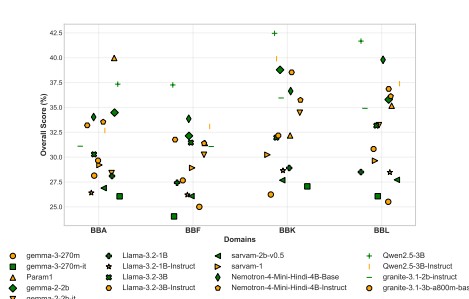

Figure 2: Comparative performance of small models (≤4B) over BhashaBench V1.

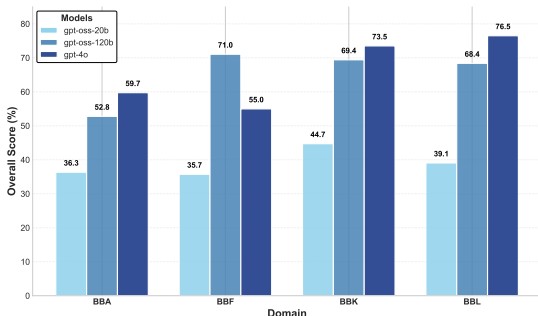

Figure 3: Comparative performance analysis of the GPT model family on BhashaBench V1.

context preservation, ambiguity resolution, and consistency standardization. Additionally, domain experts reviewed the linguistic authenticity to maintain the natural flow and idiomatic expressions characteristic of each language. This comprehensive multi-stage validation approach ensured that BhashaBench V1 maintains the highest data quality standards while preserving the authentic complexity and cultural specificity of the original examination materials.

### 3.4 DATA ANALYSIS

Figure 1 presents the comprehensive statistics of BhashaBench V1. Detailed exposition is provided in Appendix C.2. Of the total 74,166 questions, 70.8% are in English while 29.2% are in Hindi, reflecting practical bilingual usage patterns in Indian professional contexts. The dataset spans four specialized domains with varying complexity levels across 91 subdomains.

**Agriculture (BBK):** This domain encompasses agricultural sciences relevant to Indian farming systems across 25 subdomains. Agronomy dominates with 5,078 questions, reflecting its foundational role in agricultural education. The domain covers traditional practices alongside emerging areas like Agricultural Biotechnology and IT solutions. Its balanced difficulty distribution (44% easy, 45% medium, 11% hard) ensures comprehensive skill assessment.

**Finance (BBF):** Covers India's complex financial ecosystem through 30 subdomains. Problem Solving leads with 5,686 questions, followed by Mathematics for Finance (4,845), emphasizing the quantitative nature of financial practice. The domain uniquely incorporates India-specific areas like Rural Economics and Environmental Finance while addressing modern fintech developments.

**Ayurveda (BBA):** Represents traditional Indian medicine across 16 subdomains. Kayachikitsa (General Medicine) forms the core with 3,134 questions, while Dravyaguna covers pharmacology and therapeutics (2,972). This domain shows the highest proportion of accessible questions (53% easy), reflecting its foundational knowledge structure.

**Legal (BBL):** Encompasses Indian jurisprudence through 20 subdomains. Civil Litigation & Procedure dominates with 7,126 questions, followed by Constitutional Law (3,609). The domain balances traditional legal areas with contemporary developments like Technology & Cyber Law, maintaining strong cultural relevance through Family & Personal Law.

The predominantly MCQ format (>90%) ensures consistent evaluation methodology while supporting diverse cognitive assessment approaches across India-specific knowledge systems.

### 4 EXPERIMENTAL SETUP

We evaluate multiple state-of-the-art models on BhashaBench V1, including large proprietary models, open-source multilingual models, and domain-specific fine-tuned variants. Both base versions and instruction fine-tuned models are assessed to measure the effectiveness of specialized training approaches across India-specific knowledge domains. All evaluations are conducted in a zero-shot setting to assess the models' inherent capabilities without task-specific examples. For open-source models, we utilize the LM-EVALUATION-HARNESS library (Biderman et al., 2024a;b) to ensure

clean, reproducible, and standardized evaluations. We employ the log-likelihood method where the probability of a given output string is computed by conditioning it on the provided input (Brown et al., 2020). For multiple choice questions with $k$ possible answer choices, we select the answer string $(a_i)$ with the highest conditional log probability: $\arg\max(\log P(a_1|x), ..., \log P(a_k|x))$.

For closed-source and large-scale proprietary models, we utilize their respective APIs for evaluation due to computational constraints and access limitations. These API-based models are evaluated using a generative approach and are prompted to generate responses in a structured JSON format to facilitate automated response parsing. This comprehensive experimental framework enables systematic comparison across diverse model architectures while maintaining evaluation consistency across both open-source and proprietary systems. Additional details regarding model specifications, hyperparameters, and computational resources are provided in Appendix D.

# 5 RESULTS AND DISCUSSIONS

In this section, we discuss the results and our findings across all the experiments conducted.

## 5.1 ZERO-SHOT PERFORMANCE ACROSS ALL DOMAINS (EN + HI)

Table 2 shows the performance of various models in English and Hindi under the zero-shot setup. Among these, Qwen3-235B-A22B-Instruct emerges as the strongest model, consistently outperforming all competitors across both languages, with an average accuracy of 67.25%. This is followed by GPT-4O at 66.18% and gpt-oss-120b at 65.41%. Performance shows clear stratification across model sizes and types, with models exceeding 27B parameters demonstrating substantially higher accuracies compared to smaller variants. Among the 7B-27B range, gemma-2-27b leads with 53.11% average accuracy, followed by gemma-2-27b-it at 44.64%. In the mid-range category, gemma-2-9b shows impressive performance at 48.07%, with Pangea-7B achieving 41.54%.

Smaller models under 4B parameters show more modest performance, with Qwen2.5-3B achieving the highest accuracy in this category at 39.68%. Models specifically designed for Indian languages include Param-1 (34.69%) and the Nemotron-4-Mini-Hindi variants (36.08% and 34.20%). Performance is notably higher in English compared to Hindi across most models, reflecting the typical pattern observed in multilingual language models, with models showing varying degrees of cross-lingual transfer capabilities.

## 5.2 HOW DO MODELS PERFORM IN SUBDOMAINS

We evaluate representative models across BBA, BBF, BBK, and BBL to capture performance within subdomains (see Figures 4 and 5). Qwen3-235B-A22B-Instruct-2507 achieves the strongest results, excelling in Research & Statistics (91.43%), Agricultural Biotechnology (91.6%), and Intellectual Property Law (87.91%). GPT-4o demonstrates robust performance, frequently scoring above 70% with peaks of 92% in Information Technology and Healthcare & Medical Law. GPT-oss-120b shows competitive performance, closely matching gpt-4o in domains like Agricultural Biotechnology (89.69%). Mid-sized models including Gemma-2-27b and Gemma-2-9b generally show moderate performance in the 50–70% range, with the 27B variant consistently outperforming its smaller counterpart. Llama-3.1-8B demonstrates limited performance, typically scoring 30–50% across domains. The compact Param-1 model shows consistent baseline performance, often matching Llama-3.1-8B despite requiring significantly fewer resources. Notable patterns emerge: Finance and Legal domains show the highest performance ceiling, with top models regularly exceeding 80% in Business Management and Constitutional Law. Agricultural domains present moderate complexity, while Ayurveda proves most challenging, with even the best models rarely exceeding 80% in specialized areas like Panchakarma. Results highlight clear advantages for large models in knowledge-intensive tasks, while smaller models provide practical utility in resource-constrained scenarios for general applications.

## 5.3 PERFORMANCE ANALYSIS ACROSS QUESTION DIFFICULTY LEVELS

We evaluated model performance on Easy, Medium, and Hard questions across the four benchmark domains BBA, BBF, BBK, and BBL. In BBA, top-performing models such as GPT-4o and Qwen3-

Table 2: Zero-shot scores (%) of LLMs across domains on BhashaBench V1 (EN + HI). The benchmark covers Agriculture (BBK), Finance (BBF), Legal (BBL), and Ayurveda (BBA). "Avg" denotes the overall average across that domain. Top-scoring models are highlighted as follows: yellow for models < 4B parameters, green for models between 4B and 27B, and blue for models > 27B.

| Model | BBA | | | BBF | | | BBK | | | BBL | | |
|---|---|---|---|---|---|---|---|---|---|---|---|---|
| | Eng | Hin | Avg | Eng | Hin | Avg | Eng | Hin | Avg | Eng | Hin | Avg |
| *< 4B Models* | | | | | | | | | | | | |
| gemma-3-270m | 28.08 | 28.25 | 28.14 | 24.98 | 25.06 | 25.00 | 26.64 | 24.45 | 26.24 | 25.49 | 25.54 | 25.51 |
| gemma-3-270m-it | 26.23 | 25.77 | 26.06 | 24.13 | 23.84 | 24.04 | 27.44 | 25.35 | 27.06 | 25.56 | 27.26 | 26.07 |
| Param-1 | 41.12 | 38.04 | 39.97 | 32.24 | 29.56 | 31.42 | 33.10 | 27.97 | 32.18 | 36.15 | 32.89 | 35.17 |
| gemma-2-2b | 36.80 | 30.61 | 34.48 | 34.20 | 27.50 | 32.14 | 41.24 | 27.49 | 38.78 | 38.45 | 29.61 | 35.79 |
| gemma-2-2b-it | 29.38 | 26.79 | 28.40 | 31.26 | 27.93 | 30.24 | 35.94 | 27.71 | 34.47 | 34.49 | 30.25 | 33.22 |
| Llama-3.2-1B | 29.17 | 26.30 | 28.10 | 28.24 | 25.61 | 27.43 | 29.71 | 25.21 | 28.91 | 29.63 | 25.88 | 28.52 |
| Llama-3.2-1B-Instruct | 26.77 | 25.82 | 26.41 | 26.28 | 26.04 | 26.21 | 29.16 | 26.33 | 28.65 | 29.08 | 27.04 | 28.47 |
| Llama-3.2-3B | 31.62 | 28.05 | 30.28 | 33.04 | 27.92 | 31.46 | 32.68 | 28.69 | 31.96 | 35.17 | 28.53 | 33.17 |
| Llama-3.2-3B-Instruct | 35.31 | 29.67 | 33.20 | 32.94 | 29.09 | 31.76 | 40.59 | 29.09 | 38.53 | 39.74 | 30.13 | 36.86 |
| sarvam-2b-v0.5 | 26.79 | 27.07 | 26.89 | 26.42 | 25.31 | 26.08 | 28.14 | 25.57 | 27.68 | 28.49 | 25.95 | 27.72 |
| sarvam-1 | 29.70 | 28.41 | 29.21 | 29.66 | 27.27 | 28.92 | 30.82 | 27.57 | 30.24 | 30.92 | 26.66 | 29.64 |
| Nemotron-4-Mini-Hindi-4B-Base | 34.76 | 32.82 | 34.03 | 34.95 | 31.41 | 33.86 | 36.67 | 36.49 | 36.64 | 40.75 | 37.55 | 39.79 |
| Nemotron-4-Mini-Hindi-4B-Instruct | 33.38 | 33.82 | 33.54 | 31.98 | 30.06 | 31.39 | 35.83 | 35.33 | 35.74 | 36.99 | 34.11 | 36.12 |
| Qwen2.5-3B | 40.61 | 31.90 | 37.34 | 39.54 | 32.13 | 37.26 | 44.57 | 32.72 | 42.45 | 44.98 | 33.97 | 41.67 |
| Qwen2.5-3B-Instruct | 35.22 | 28.46 | 32.68 | 34.84 | 29.17 | 33.09 | 42.67 | 27.20 | 39.90 | 40.62 | 29.89 | 37.39 |
| granite-3.1-2b-instruct | 33.39 | 27.30 | 31.10 | 32.82 | 27.11 | 31.07 | 37.71 | 27.86 | 35.95 | 38.18 | 27.30 | 34.91 |
| granite-3.1-3b-a800m-base | 31.75 | 26.18 | 29.66 | 29.22 | 24.17 | 27.66 | 33.36 | 26.70 | 32.17 | 33.74 | 24.01 | 30.82 |
| *7B to 27B Models* | | | | | | | | | | | | |
| Pangea-7B | 40.69 | 31.93 | 37.41 | 41.71 | 33.73 | 39.25 | 47.16 | 34.71 | 44.93 | 48.70 | 34.95 | 44.57 |
| Indic-gemma-7b-finetuned-sft-Navarasa-2.0 | 37.12 | 31.83 | 35.13 | 37.00 | 30.47 | 34.90 | 42.31 | 33.44 | 40.73 | 44.08 | 34.09 | 41.08 |
| aya-23-8B | 33.84 | 28.87 | 31.97 | 35.25 | 30.88 | 33.90 | 37.09 | 33.22 | 36.40 | 41.92 | 33.01 | 39.24 |
| Llama-3.1-8B | 35.48 | 29.17 | 33.12 | 36.20 | 30.61 | 34.48 | 39.52 | 31.41 | 38.07 | 41.32 | 31.76 | 38.44 |
| Llama-3.1-8B-Instruct | 36.86 | 31.26 | 34.76 | 35.68 | 30.27 | 34.01 | 47.14 | 35.07 | 44.98 | 48.61 | 36.47 | 44.96 |
| gemma-2-9b | 48.16 | 37.92 | 44.32 | 42.73 | 36.91 | 40.94 | 55.23 | 43.89 | 53.20 | 58.49 | 42.96 | 53.83 |
| gemma-2-9b-it | 36.22 | 31.18 | 34.33 | 38.85 | 32.03 | 36.75 | 48.92 | 36.45 | 46.69 | 45.05 | 38.66 | 43.13 |
| gpt-oss-20b | 38.30 | 33.09 | 36.34 | 37.11 | 32.61 | 35.73 | 46.58 | 36.27 | 44.73 | 40.69 | 35.24 | 39.06 |
| gemma-2-27b | 50.70 | 42.26 | 47.53 | 47.79 | 41.24 | 45.77 | 59.84 | 50.38 | 58.14 | 64.91 | 51.83 | 60.99 |
| gemma-2-27b-it | 40.45 | 33.89 | 37.99 | 42.47 | 34.29 | 39.95 | 54.95 | 41.24 | 52.50 | 50.71 | 42.02 | 48.10 |
| *> 27B Models* | | | | | | | | | | | | |
| gpt-oss-120b | 55.62 | 48.05 | 52.78 | 74.11 | 64.16 | 71.05 | 71.40 | 60.25 | 69.41 | 70.72 | 62.94 | 68.38 |
| Qwen3-235B-A22B-Instruct-25076 | 60.25 | 54.78 | 58.20 | 63.72 | 56.27 | 61.43 | 74.57 | 64.13 | 72.70 | 80.15 | 68.60 | 76.68 |
| deepseek-v3 | 51.38 | 37.03 | 45.99 | 63.46 | 57.04 | 61.48 | 62.93 | 45.01 | 59.73 | 67.78 | 46.78 | 61.47 |
| gpt-4o | 62.75 | 54.73 | 59.74 | 57.27 | 49.82 | 54.97 | 75.31 | 65.18 | 73.50 | 78.83 | 71.02 | 76.49 |

235B-A22B-Instruct-2507 achieved 66.4% and 65.18% on Easy questions, and 47.09% and 46.24% on Hard questions, while smaller models like gemma-3-270m scored 28.1% on Easy and 26.81% on Hard. A similar trend is observed in BBF, with Easy question scores ranging from 24.15% (gemma-3-270m) to 74.8% (gpt-oss-120b) and Hard questions from 21.22% to 62.61%. Medium-level questions show moderate differentiation, reflecting model reasoning capability. BBK and BBL follow the same pattern, with instruction-tuned and larger models consistently outperforming smaller models, particularly on Hard questions. Overall, model size, instruction tuning, and architecture significantly influence robustness to question difficulty and generalization across domains. See Appendix E.1.

## 5.4 PERFORMANCE ANALYSIS ACROSS QUESTION TYPES

We analyzed model performance on various question types including Assertion/Reasoning, Fill in the Blanks, MCQs, Match the Column, Reading Comprehension, and Rearrange the Sequence across the BBA, BBF, BBK, and BBL domains. In BBA, models like deepseek-v3 and GPT-4o achieved high scores of 66.67% and 62.96% on Assertion/Reasoning questions, whereas smaller models such as gemma-3-270m scored 28.09%. For Fill in the Blanks, scores ranged from 24.72% (gemma-3-270m-it) to 51.69% (Qwen3-235B-A22B-Instruct-2507). MCQ performance was moderate, between 26% and 59.95%. Match the Column and Reading Comprehension showed wider variation, with larger models consistently outperforming smaller or non-instruction-tuned models. Rearrange the Sequence proved challenging across domains, with top models reaching 71.43% in BBL. Overall, question type significantly affects performance, highlighting the importance of model size, instruction tuning, and reasoning capabilities in handling diverse formats.

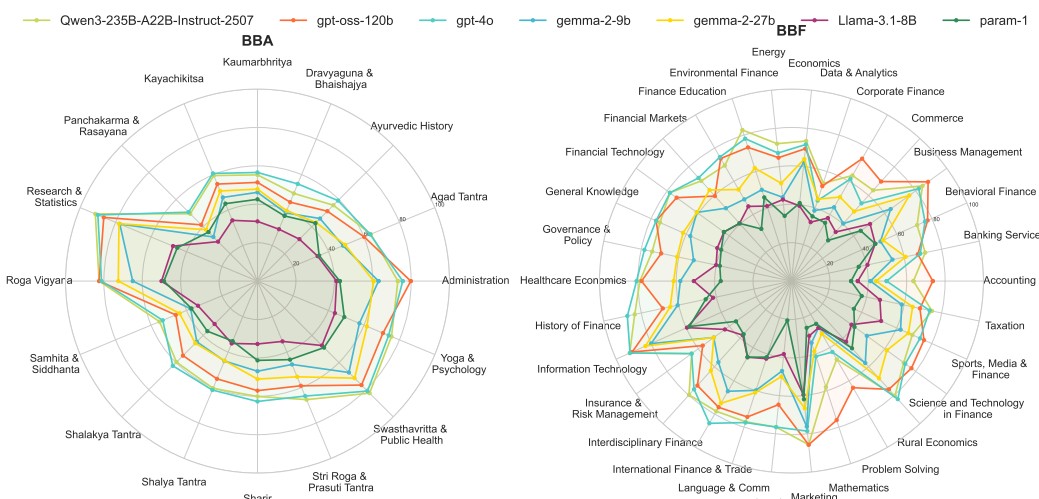

Figure 4: Comparison of representative LLMs' scores across different domains and subdomains.

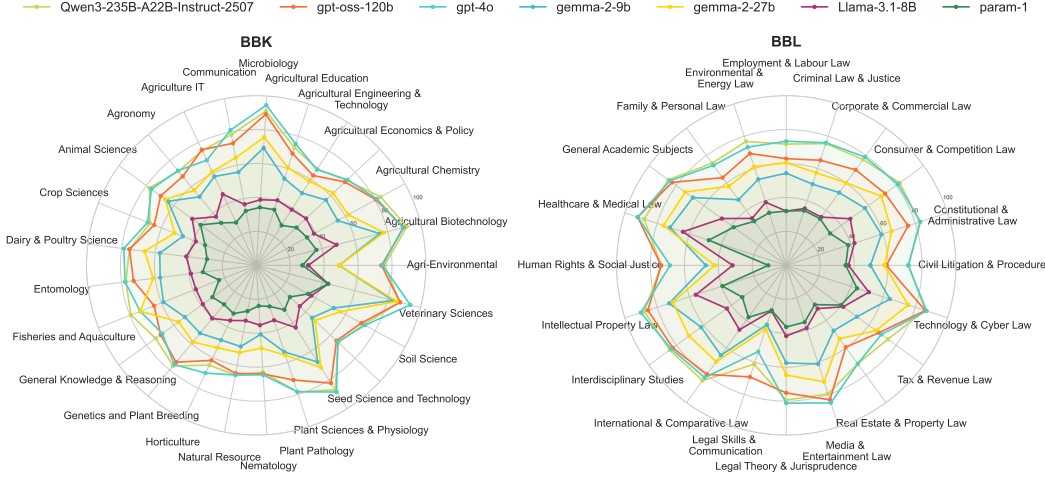

Figure 5: Comparison of representative LLMs' scores across different domains and subdomains.

## 5.5 PERFORMANCE ANALYSIS OF GPT MODEL FAMILY

We evaluate the GPT model family across BBA, BBF, BBK, and BBL domains to understand scaling and architectural strengths (Figure 3). gpt-oss-20b demonstrates baseline performance with scores of 36.34% (BBA), 35.73% (BBF), 44.73% (BBK), and 39.06% (BBL). Scaling to gpt-oss-120b yields substantial improvements: 52.78% in BBA, 71.05% in BBF, 69.41% in BBK, and 68.38% in BBL, representing 16-35 percentage point gains. Despite gpt-4o's larger parameter count, gpt-oss-120b significantly outperforms it in Finance (71.05% vs 54.97%), likely due to BBF's mathematical reasoning emphasis where gpt-oss-120b's training methodology excels (Analysis, 2025). Conversely, gpt-4o shows superior performance in Legal (76.49%) and Agriculture (73.5%) domains. This highlights that parameter size (Babbar, 2025) alone doesn't guarantee performance; architectural choices and training approaches significantly influence domain-specific capabilities, with mathematical tasks favoring specific optimizations over raw parameter scaling.

## 5.6 PERFORMANCE ANALYSIS OF SMALL MODELS

We evaluate small models (≤4B parameters) across BBA, BBF, BBK, and BBL domains to assess efficiency-performance trade-offs (Figure 2). Param-1 and Qwen2.5-3B emerge as comparable top performers, with Param-1 achieving 39.97% in BBA while Qwen2.5-3B excels in BBK (42.45%). Both models demonstrate complementary strengths: Param-1 performs better in Ayurveda, while

Qwen2.5-3B shows superior performance in Finance, Agriculture, and Legal domains. Instruction tuning effects vary significantly across architectures: Llama-3.2-3B-Instruct substantially outperforms its base version, whereas Qwen2.5-3B-Instruct shows mixed results. Nemotron-4-Mini-Hindi models achieve competitive performance in the 34-40% range, while the smallest models like gemma-3-270m struggle consistently below 28%. Results indicate that architectural efficiency and targeted optimization can achieve reasonable performance in resource-constrained scenarios, with Param-1 and Qwen2.5 leading the small model category through different domain specializations.

## 5.7 ROBUSTNESS AND CONTAMINATION ANALYSIS

To verify BhashaBench V1's integrity and rule out potential data leakage, we conduct perplexity (Jelinek et al., 1977) analysis on Llama-3.1-8B and Gemma-2-9B models. Table 3 shows BhashaBench V1 datasets (BBA, BBF, BBK, BBL) have perplexity scores comparable to or higher than established benchmarks like ARC-C, MMLU, and MILU. Notably, BBA shows higher perplexity (15.5 English, 10.39 Hindi on Llama-3.1-8B), while BBK is elevated in Hindi (7.16, 9.54) relative to English. To test potential position bias, we conduct option shuffling across multiple seeds, showing consistent performance with minimal variance. These patterns indicate minimal pretraining exposure and clearly show BhashaBench V1 provides genuinely novel evaluation challenges. Detailed perplexity and shuffling results are in Appendix E.4.

Table 3: Comparison of perplexity (PPL) across evaluation datasets. The PPL for English and Hindi datasets is reported for Llama-3.1-8B and gemma-2-9b models.

| Datasets (MCQ) | Llama-3.1-8B | | gemma-2-9b | |
|---|---|---|---|---|
| | English | Hindi | English | Hindi |
| ARC-C | 8.03 | 4.1 | 6.82 | 5.85 |
| MILU | 7.62 | 4.93 | 7.23 | 6.37 |
| MMLU | 7.61 | 4.22 | 7.03 | 6.21 |
| BBA | 15.5 | 10.39 | 23.2 | 16.8 |
| BBF | 6.78 | 4.03 | 5.86 | 5.04 |
| BBK | 6.14 | 7.16 | 6.34 | 9.54 |
| BBL | 7.28 | 4.01 | 7.19 | 5.79 |

## 5.8 STATISTICAL SIGNIFICANCE ANALYSIS

To validate BhashaBench V1 results, we conduct statistical significance testing using two complementary approaches: we compute 95% Wilson Confidence Intervals (Wilson, 1927) for all models and domains, typically within 1-2 percentage points, demonstrating benchmark stability and reproducibility. We also perform pairwise McNemar's tests (Mcnemar, 1947) on the top 5 models per domain to check whether performance differences are significant. Results reveal that most pairwise comparisons show statistically significant differences $p < 0.05$, confirming that observed performance gaps reflect genuine model capability rather than statistical noise. When accuracy differences are minimal (e.g., GPT-4o vs. Qwen3-235B-A22B-Instruct-25076 on BBA Hindi: 0.05% difference, p=0.9591), the test correctly identifies non-significant differences, demonstrating appropriate statistical rigor. Detailed results are in Appendix E.5.

## 6 CONCLUSION

In this paper, we introduced **BhashaBench V1**, a comprehensive, domain-specific, bilingual benchmark designed to evaluate large language models on India-centric knowledge systems across four critical domains: Agriculture (BBK), Legal (BBL), Finance (BBF), and Ayurveda (BBA). Our benchmark addresses significant gaps in existing evaluation frameworks by focusing on culturally relevant, domain-specific knowledge spanning over 90 subdomains and 500+ specialized topics curated from authentic government and professional examination papers. Our extensive evaluation reveals substantial performance disparities in current LLMs when applied to India-specific contexts, with models excelling in Legal contexts while struggling with traditional knowledge systems like Ayurveda and consistently performing better on English content compared to Hindi across all domains. These results highlight the urgent need for specialized model development strategies that incorporate India-specific knowledge, cultural contexts, and robust multilingual capabilities. To foster open research and accelerate progress toward more inclusive, culturally aware language models, we release BhashaBench V1 alongside all evaluation code and comprehensive documentation. We believe BhashaBench V1 offers a foundational benchmark for developing culturally sensitive models that effectively serve India's diverse linguistic and knowledge landscape.

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

# Appendix

CONTENTS

## A    LIMITATIONS AND BIASES

In this paper, we introduce BhashaBench V1, providing a comprehensive evaluation of LLMs on India-centric knowledge systems and exploring model capabilities across critical Indian domains. However, there are several limitations to acknowledge. (1) Language Coverage Limitations: Although BhashaBench V1 supports English and Hindi, covering a significant portion of India's population, India has 22 official languages and hundreds of regional dialects. Our current evaluation cannot capture the full linguistic diversity of Indian knowledge systems, particularly regional variations in agricultural practices, legal terminologies, and traditional medicine nomenclature that exist in languages like Tamil, Telugu, Bengali, and others. Future iterations will expand to include additional Indian languages to enhance coverage. (2) Domain Scope Limitations: While we cover four fundamental domains (Agriculture, Legal, Finance, and Ayurveda) representing core areas of Indian society, our assessment cannot encompass the entire breadth of India-specific knowledge systems. Areas such as traditional crafts, regional governance systems, indigenous engineering practices, and other vernacular knowledge traditions remain unexplored for future expansion. Our content spans from grassroots practical knowledge to professional examination standards, ensuring broad applicability across different expertise levels. (3) Evaluation Methodology Limitations: Our evaluation primarily uses structured question formats derived from authentic government and professional examinations. While this ensures real-world relevance and practical applicability, it may not fully capture all forms of contextual reasoning required in complex domain applications.

The main biases in BhashaBench V1 can be categorized into three aspects: (1) Source Material Bias: Despite comprehensive curation from diverse authentic sources spanning grassroots to professional levels, certain regional practices and emerging contemporary developments may be underrepresented. (2) Language Resource Bias: The benchmark reflects the inherent resource disparity between English and Hindi, where Hindi content, while substantial, represents a relatively lower-resource context compared to English. (3) Examination Framework Bias: Our reliance on established examination systems, while ensuring authenticity, may introduce institutional perspectives present in the original assessment frameworks. However, our extensive coverage across 90+ subdomains and 500+ topics from diverse sources mitigates this bias significantly. The impact of these limitations on LLM evaluation includes clear performance distinctions between models across domains and languages, as evidenced by the substantial score variations from 34.28% to 76.49%, demonstrating BhashaBench V1's effectiveness in distinguishing LLM capabilities while presenting meaningful challenges even for top-performing models in India-specific contexts.

## B    TOWARDS BROADER IMPACT

**Societal Impact.** BhashaBench V1 is anticipated to play a transformative role in bridging the digital divide for India-centric knowledge systems. LLMs trained and evaluated with BhashaBench V1 can significantly enhance accessibility to critical domain expertise across agriculture, legal services, finance, and traditional medicine, particularly benefiting underserved rural and semi-urban populations. In agriculture, improved LLM capabilities can democratize access to expert crop advisory, pest management, and sustainable farming practices, potentially impacting the livelihoods of over 40 million farmers dependent on agricultural activities. In the legal domain, enhanced models can assist with legal document comprehension, procedural guidance, and basic legal literacy, addressing the substantial access-to-justice challenges faced by millions in India's complex legal system. For healthcare, particularly Ayurveda, better model performance can support practitioners and patients in understanding traditional treatment protocols and medicinal formulations, preserving and disseminating indigenous medical knowledge. In finance, improved model capabilities can enhance financial literacy and support the growing digital payment ecosystem processing billions of transactions annually. However, we acknowledge potential risks including over-reliance on automated systems for critical decisions, potential displacement of traditional knowledge practitioners, and the risk of perpetuating biases present in examination-based evaluation systems. The benchmark's focus on professional examination standards, while ensuring quality, may inadvertently favor formal educational backgrounds over experiential knowledge.

**Ethics Statement.** We ensure strict adherence to applicable laws and ethical guidelines throughout our data collection, curation, and usage processes. All question-answer pairs are sourced exclusively from publicly available government and professional examination papers, respecting intellectual

property rights and ensuring no unauthorized reproduction of copyrighted materials. Our curation process involved diverse teams to minimize cultural and regional biases, though we acknowledge the inherent limitations of our current English and Hindi coverage. The dataset contains no personally identifiable information, offensive content, or culturally insensitive material. All content has been thoroughly verified for authenticity and accuracy through multiple validation rounds involving domain experts. BhashaBench V1 is intended solely for academic research and educational purposes to advance inclusive AI development for Indian contexts. Any commercial use, misuse for harmful applications, or deployment without appropriate safeguards is strictly prohibited. We strongly urge all users to employ this resource responsibly, ensuring that any models developed or evaluated using BhashaBench V1 are deployed with appropriate human oversight, particularly in critical domains affecting public welfare, and with transparent disclosure of model limitations to end users.

## C  MORE DETAILS ON BHASHABENCH V1

### C.1  DETAILS OF DATA COLLECTION AND PROCESSING

This appendix provides comprehensive details on the data collection and processing methodology employed in BhashaBench V1, including systematic documentation of examination sources, processing pipelines, and quality validation procedures.

#### C.1.1  EXAMINATION SOURCE DOCUMENTATION

Our data collection strategy encompassed a wide range of authoritative examination bodies across India, ensuring comprehensive coverage of national and regional assessment standards. Table 4 presents the complete list of examination organizations and the corresponding years from which question papers were collected. We systematically gathered question papers from official examination portals that host previously released materials, manually curated by subject matter experts with accurate topic tagging, language annotation, and validated answer keys.

The temporal distribution of collected materials spans from 1995 to 2025, capturing evolving educational standards and assessment patterns while maintaining contemporary relevance. Table 5 provides a detailed breakdown of specific examination types and their collection timeline, demonstrating the breadth and depth of our data sourcing strategy. Our collection process prioritized authentic examination materials from competitive examinations that directly assess knowledge in our target domains of Agriculture, Legal, Finance, and Ayurveda.

Regional state examinations proved particularly valuable as they incorporate state-specific topics, local knowledge systems, and cultural practices often overlooked in national assessments. These examinations are typically taken by individuals seeking higher education opportunities or career advancement in business, finance, and legal sectors, ensuring questions reflect practical, real-world knowledge requirements essential for professional contexts in India.

Table 4: Organizations and Their Examination Year Ranges

| Organization | Year Range |
|---|---|
| AIACAT (Private conducting body) | 2022–2023 |
| Acharya N.G. Ranga Agricultural University (ANGRAU) | 2016–2024 |
| Agricultural Scientists Recruitment Board (ASRB) | 2013–2024 |
| All India Management Association (AIMA) | 2018–2025 |
| Banaras Hindu University (BHU) | 2013–2017 |
| Bank of Baroda | 2005–2023 |
| Bank of India | 2023 |
| Bank of Maharashtra | 2021 |
| Bar Council of India (BCI) | 2009–2021 |
| Bihar Public Service Commission (BPSC) | **1995–2024** |
| Chhattisgarh Professional Examination Board (CG Vyapam) | 2013–2019 |
| Consortium of National Law Universities (NLUs) | 2021–2025 |

Table 4 – *Continued from previous page*

| Organization | Year Range |
|---|---|
| ECGC Ltd. | 2021–2022 |
| Employees' Provident Fund Organisation (EPFO) | 2019–2023 |
| Food Corporation of India (FCI) | 2015 |
| High Court of Delhi | 2011–2023 |
| High Court/PSC (state-specific) | 2001–2021 |
| ICMAB (as per exam title) | 2016–2022 |
| IDBI Bank | 2014–2022 |
| Indian Council of Agricultural Research (ICAR) | 2017–2023 |
| Indian Farmers Fertiliser Cooperative Limited (IFFCO) | 2019–2022 |
| Indian Institutes of Management (IIMs) | 2017–2024 |
| Institute of Banking Personnel Selection (IBPS) | 2016–2024 |
| JNTU Kakinada on behalf of APSCHE | 2012–2025 |
| Law School Admission Council (LSAC Global) | 2010–2019 |
| MP Professional Examination Board (MPPEB/PEB) | 2016–2024 |
| Maharashtra Agricultural Universities Examination Board (MAUEB) under MCAER | 2024 |
| Maharashtra Public Service Commission (MPSC) | 2010–2025 |
| Narendra Deva University of Agriculture & Technology | 2024–2025 |
| National Bank for Agriculture and Rural Development (NABARD) | 2018–2023 |
| National Law University, Delhi (NLU Delhi) | 2016–2025 |
| National Testing Agency (NTA) | **2019–2025** |
| Reserve Bank of India (RBI) | **2015–2025** |
| RVSKVV & JNKVV | 2022 |
| Small Industries Development Bank of India (SIDBI) | 2016–2023 |
| State Bank of India (SBI) | **2018–2025** |
| State Common Entrance Test Cell, Maharashtra | 2014–2020 |
| SVKM's NMIMS | 2019–2025 |
| The Institute of Chartered Accountants of India (ICAI) | **2018–2025** |
| The Institute of Cost Accountants of India (ICMAI) | 2022–2025 |
| The Nainital Bank Ltd. | 2019–2020 |
| Union Public Service Commission (UPSC) | **2002–2025** |
| University of Delhi | 2015–2019 |
| University-specific (varies) | 2020–2024 |
| Uttar Pradesh Public Service Commission (UPPSC) | 2019–2025 |

### C.1.2 PROCESSING PIPELINE ARCHITECTURE

The comprehensive end-to-end pipeline developed for transforming raw examination materials into the structured BhashaBench V1 dataset incorporates multiple quality control checkpoints and validation stages to ensure data integrity and authenticity. The pipeline consists of seven major stages, each designed to address specific challenges encountered in multilingual examination material processing.

Table 5: Examination Names and Their Year Ranges

| Examination Name | Year Range |
|---|---|
| AGRICET | 2016–2024 |
| AIACAT - All India Agriculture Common Aptitude Test | 2022–2023 |
| AIAPGET - All India AYUSH Post Graduate Entrance Test (Ayurveda) | 2022–2025 |
| All India Bar Examination (AIBE) | 2009–2021 |
| All India Law Entrance Test (AILET) | 2016–2025 |
| Andhra Pradesh Judicial Service (Prelims) | 2012 |
| AP EAMCET | 2012–2025 |

*Continued on next page*

Table 5 – *Continued from previous page*

| Examination Name | Year Range |
|---|---|
| ASRB NET Agriculture | 2013–2024 |
| BHU PET | 2017 |
| BHU PG | **2013–2017** |
| BHU RET | 2014–2017 |
| BHU UET | 2016–2017 |
| BPSC | **1995–2024** |
| Bank of Baroda | 2005–2023 |
| Bank of India | 2023 |
| Bank of Maharashtra | 2021 |
| CAT | 2017–2024 |
| CG PAT Agriculture | 2013–2019 |
| CMA | 2022–2025 |
| CMAT | 2022–2025 |
| Common Law Admission Test (CLAT) | 2021–2025 |
| CUET Agriculture Previous Year Papers | 2022–2025 |
| CUET PG (Law) | 2023–2025 |
| Delhi Judicial Service | 2011–2023 |
| DU LL.B Entrance | 2015–2019 |
| ECGC PO | 2021–2022 |
| EPFO Assistant | 2019 |
| EPFO SSA | 2019–2023 |
| EPFO Stenographer | 2023 |
| FCI Agriculture | 2015 |
| Haryana Judicial Service (Prelims) | 2015–2021 |
| Himachal Pradesh Judicial Service (Prelims) | 2007–2019 |
| IBPS AFO Agriculture Field Officer | 2016–2024 |
| IBPS AFO Mains | 2017–2023 |
| IBPS Clerk | 2023–2024 |
| IBPS PO | 2018–2024 |
| IBPS RRB Officer Scale-I (merged) | 2018–2024 |
| IBPS SO | 2019 |
| ICAI Final | 2018–2025 |
| ICAI Foundation | 2018–2025 |
| ICAI Intermediate | 2018–2025 |
| ICAR AICE JRF/SRF (PHD) Agriculture | 2020–2024 |
| ICAR AIEEA (PG) Agriculture | 2019–2024 |
| ICAR AIEEA (UG) Agriculture | 2017–2023 |
| ICMAB New Syllabus | 2016–2022 |
| ICMAB Old Syllabus | 2016–2021 |
| IDBI Assistant Manager | 2021 |
| IDBI Executive | 2014–2022 |
| IFFCO AGT - Agriculture Graduate Trainee | 2019–2022 |
| IPMAT | 2019–2023 |
| Jharkhand Judicial Service (Prelims) | 2008–2019 |
| JNKVV & RVSKVV Joint Entrance (M.Sc./Ph.D.) | 2022 |
| Karnataka Judicial Service (Prelims) | 2012 |
| LL.B. Admission Test | 2022–2024 |
| LL.M. Admission Test | 2020–2024 |
| LSAT - India | 2010–2019 |
| Madhya Pradesh Judicial Service (Prelims) | 2001–2018 |
| Maharashtra Judicial Service (Prelims) | 2010–2019 |
| MAT | 2018–2025 |
| MCAER-CET | 2024 |
| MH CET Law (3-year LL.B.) | 2016–2019 |
| MH-CET | 2014–2020 |

Table 5 – *Continued from previous page*

| Examination Name | Year Range |
|---|---|
| MP PAT Agriculture | 2016–2024 |
| MPSC | 2010–2025 |
| NABARD Agriculture Development Officer | 2018–2023 |
| Nainital Bank Clerk | 2019 |
| Nainital Bank PO | 2020 |
| NPAT | 2019–2025 |
| Odisha Judicial Service (Prelims) | 2011 |
| Rajasthan Judicial Service (Prelims) | 2011–2021 |
| RBI Grade B | **2015–2025** |
| SBI Apprentice | 2019–2023 |
| SBI CBO | 2024 |
| SBI Clerk | 2022–2025 |
| SBI PO | **2018–2025** |
| SIDBI Grade A | 2016–2023 |
| TANCET | 2024–2025 |
| TG ICET (TS ICET) | 2022–2024 |
| UGC NET (Law) | 2014–2015 |
| UPCATET | 2024–2025 |
| UPPSC Prelims | 2019–2025 |
| UPSC EPFO | 2013–2017 |
| UPSC EPFO APFC | **2002–2023** |
| UPSC IFS - Indian Forest Service | 2023–2024 |
| UPSC Prelims - Economy | 2025 |
| UPSC Prelims - Polity & Governance | 2025 |
| Uttarakhand Judicial Service (Prelims) | 2011 |
| West Bengal Judicial Service (Prelims) | 2011 |

The data acquisition stage involved systematic collection from official portals with comprehensive metadata extraction including examination year, conducting body, subject classification, and language identification. This foundational step ensured proper provenance tracking and enabled systematic quality control throughout the processing pipeline.

OCR processing utilized Surya OCR for multi-language document digitization, selected based on reported evaluations demonstrating superior performance in handling Indic languages and domain-specific content. Prior studies indicate 98.1% normalized text similarity for English and 98.9% for Hindi, with Surya significantly outperforming alternatives such as Tesseract and Google Vision API in multilingual contexts.

Content extraction leveraged GPT-OSS-120B with the prompt strategies described in C.1.5, enabling intelligent text structuring that addressed key challenges such as format variations across examination bodies, answer key alignment complexities, multi-format question types, and language-specific formatting conventions. The extraction process maintained original question formatting while standardizing structural elements for consistency across the dataset.

Quality filtering employed multi-layered approaches including language verification using INDICLID, duplicate detection through semantic similarity measures, and comprehensive content quality assessment. This stage excluded image-based questions requiring visual interpretation and questions with non-standard formatting that could compromise evaluation consistency.

Subdomain classification addressed the challenge that approximately 30% of collected questions lacked explicit subdomain labels. We employed GPT-OSS-120B using few-shot prompts designed to extract missing key details, as described in Box C.1.5, and refined the outputs with domain-specific taxonomies in consultation with subject matter experts to ensure accurate categorization within the BBA, BBF, BBK, and BBL domains.

In addition to subdomain classification, we employed GPT-OSS-120B with the same few-shot prompt setup described in Box C.1.5 to extract key details such as *question type* and *question level*. For both dimensions, domain-wise few-shot examples were manually curated to guide the

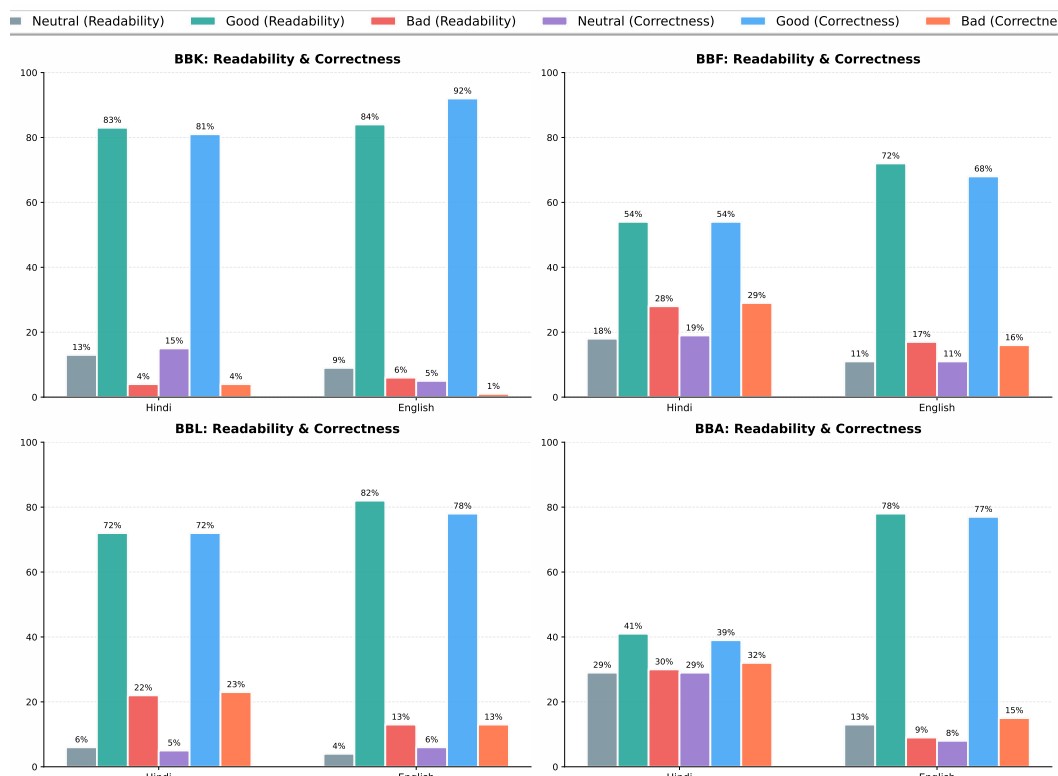

Figure 6: Manual quality assessment of BhashaBench V1 domain questions.

model. For question level, the model was prompted to categorize items into three standard difficulty classes: **Easy**, **Medium**, and **Hard**, a widely adopted practice in educational assessment. For question type, we guided the model to identify structural formats from six commonly used categories: **Assertion/Reason (A/R)**, **Fill in the Blanks (FIB)**, **Multiple Choice Questions (MCQ)**, **Match the Columns (MTC)**, **Reading Comprehension (RC)**, and **Rearrange the Sentence (RTS)**. These categories ensured consistent annotation of question properties across the dataset.

Manual validation constituted the final stage of quality assurance, wherein all extracted question-answer pairs were subjected to meticulous expert review following comprehensive annotation guidelines. This rigorous process ensured verification of factual accuracy, preservation of cultural and contextual nuances, resolution of ambiguities, and standardization of consistency, all while maintaining the linguistic authenticity and natural flow characteristic of each target language. The detailed annotation guidelines, covering all domains, are summarized in Table 6. Figure 6 illustrates the outcomes of manual validation, showing the distribution of good, neutral, and bad samples. Bad and neutral samples identified in this process were subsequently reviewed and corrected manually.

### C.1.3 OCR QUALITY ASSURANCE

The quality of Optical Character Recognition (OCR) is critical for maintaining dataset integrity, particularly given the wide variation in document layouts and scan quality across examination boards and years. To ensure high-fidelity digitization, we implemented a multi-stage OCR quality assurance framework focused on detecting structural errors, handling layout heterogeneity, and correcting language-specific OCR ambiguities. This pipeline ensured that each question, regardless of language, was accurately extracted, properly structured, and suitable for downstream evaluation.

**Handling Document Layout Heterogeneity:** Source documents exhibited considerable variation in structure, necessitating robust parsing strategies. Key challenges and our corresponding solutions included:

- **Multi-Column Layouts:** Examination papers frequently used single or double-column formats. We employed Surya OCR's layout detection to automatically identify column structures, preserving the correct reading order and preventing content mixing across column boundaries, which was crucial for maintaining question-answer associations.

- **Tabular and Mixed Content:** A significant portion of questions contained embedded tables, mathematical expressions, and figure references. We used structure-preserving extraction techniques to maintain row-column relationships in tables and developed adaptive parsing logic to handle intermixed content types, such as formulas within natural language text.

- **Formatting Variations:** Different examination boards used distinct conventions for question numbering, option labeling (e.g., A/B/C/D vs. (a)/(b)/(c)/(d)), and typographical emphasis. Our extraction pipeline utilized adaptive prompts for the GPT-OSS-120B model to normalize these structural patterns irrespective of the source's specific formatting.

**Confidence-Based Validation and Correction:** OCR outputs were processed through a confidence-based routing mechanism. Approximately 78% of extractions were well-formed and accepted automatically. The remaining 22% were flagged for further processing: 15% for being malformed or incomplete, and 7% for severe degradation requiring manual review. Layout-related failures, particularly in two-column formats, were a primary cause of malformed extractions. For the malformed extractions, we deployed an LLM-based post-correction using GPT-OSS-120B. This context-aware correction successfully resolved 82% of flagged issues. It was particularly effective at reconstructing reading order in multi-column layouts and correcting character-level errors in Devanagari script, such as similar-looking matras (vowel diacritics).

**Ensuring Data Integrity and Internal Consistency** Since the dataset is not parallel across languages, our validation framework focuses on intra-document consistency rather than cross-lingual alignment. Automated checks provide high scalability, while edge cases are routed for manual inspection.

- **Structural and Numerical Consistency Verification:** We developed automated scripts to validate the internal structural integrity of each document. These checks ensure coherent and sequential question numbering, consistent option counts, and correct nesting of sub-questions. For numerical content, a two-stage routine was applied: first, regex-based extraction captured all numerical values and mathematical expressions; second, a normalization procedure reconciled formatting variations (e.g., thousand separators, inconsistent decimal markers). Any anomaly such as missing numbers, malformed expressions, or inconsistent option structures flagged the document for manual review.

- **Answer Key Consistency Checks:** Answer labels extracted from OCR outputs were validated against expected option patterns and numbering rules. Documents exhibiting mismatched labels, duplicate correct options, or malformed answer keys were automatically routed to a validation queue. Manual inspection distinguished between systematic OCR issues (e.g., misreading 'B' as '8') and genuine inconsistencies in the original source material.

- **Domain-Specific Post-Processing:** We designed custom heuristics to correct predictable OCR errors. For mathematical content, symbol-disambiguation routines differentiated operators (e.g., multiplication sign $\times$ vs. variable $x$) and corrected spacing and subscript/superscript placement. For Devanagari (Hindi) text, we implemented algorithms to detect and correct matra (diacritic) misplacements, a common source of character-level OCR noise.

- **Filtering Cross-Referential Questions:** Certain items in the source material referenced external passages or other questions (e.g., "Read the following paragraph and answer Q4–Q6"). Because our evaluation pipeline processes each question independently, such interdependent items cannot be reliably evaluated in isolation. We automatically detected these cases and attempted to resolve them by linking the referenced text when it was explicitly present and self-contained. However, if the associated paragraph or anchor question was missing, fragmented, or could not be reconstructed consistently, the dependent questions were removed. In cases where removal introduced gaps or broken numbering sequences, we renumbered the remaining questions when a coherent sequence could be restored; otherwise, the entire block was excluded to maintain evaluation integrity.

- **Structured Extraction as a Validation Filter:** The final GPT-OSS-120B extraction stage served a dual purpose. Apart from producing the structured JSON output, its ability to successfully interpret OCR text functioned as a robust quality filter. Prompts were engineered to fail deterministically when provided with noisy or inconsistent input. Documents that the model could not parse into the required schema were automatically flagged as low-confidence and routed for manual inspection, effectively capturing severe OCR failures and internal inconsistencies.

A final round of manual expert review was conducted in accordance with the guidelines outlined in Table 6. This review complemented the automated checks and LLM-powered correction pipeline, ensuring that the curated BhashaBench V1 dataset maintained a high standard of digitization accuracy and structural integrity.

### C.1.4 ANNOTATION GUIDELINES

Our annotation guidelines were meticulously designed to ensure consistency, accuracy, and cultural authenticity across all BhashaBench V1 domains and languages. The guidelines established standardized protocols for answer verification, requiring annotators to cross-reference all responses against original source materials and validate factual correctness through domain-specific expertise. Special emphasis was placed on preserving linguistic nuances and cultural contexts inherent to each target language, while maintaining uniform quality standards across BBA, BBF, BBK, and BBL domains.

Table 6: Annotation Guidelines across Domains in BhashaBench V1

| Domain | Detailed Guidelines |
| --- | --- |
| General | • **Answer Verification:** Ensure that the provided answer key is correct. Cross-check against the original exam paper.
• **Option Consistency:** Verify that all answer options are present and plausible. Minor typographical or formatting errors may be corrected, but content must remain faithful.
• **Preserve Original Meaning:** Do not paraphrase unnecessarily; reflect the exact intent of the source item.
• **Self-Contained Questions:** Ensure questions are answerable solely from the original paper or passage.
• **Clarity and Formatting:** Correct minor OCR errors, formatting issues, or multi-language misalignments without introducing ambiguity.
• **Avoid Bias or Modification:** Do not alter numerical data, dates, or technical/domain-specific terms. |
| Agriculture | Verify crop names, farming practices, and region-specific agricultural knowledge for accuracy and contextual relevance. |
| Legal | Ensure legal terms, statutes, case references, and procedural knowledge are precise and jurisdictionally correct. |
| Finance | Preserve numerical accuracy in calculations, financial formulas, market terminology, and regulatory compliance requirements. |
| Ayurveda | Maintain correctness of medicinal terms, herb names, therapeutic practices, and traditional knowledge references. |

### C.1.5 DATA PROCESSING PROMPTS

**BBA Question-Answer Extraction Prompt Template**

```
You are an OCR forensic specialist for Ayurveda/Medical exams (BAMS
    , AIAPGET, UPSC Ayurveda optional). Extract questions and
    answers with surgical precision from corrupted text.
CRITICAL MISSION: EXTRACT EVERYTHING - NEVER SKIP QUESTIONS
```

```
PRIMARY EXTRACTION RULES
1. ZERO TOLERANCE FOR MISSING QUESTIONS
- Scan text character by character
- Look for question patterns: "Q1", "1.", "(1)", "Question 1", "Que
    .1", or ANY numbering
- Extract PARTIAL questions with [INCOMPLETE] tag rather than skip
- If options are corrupted beyond recognition, create synthetic
    placeholders
2. AYURVEDA DOMAIN OCR CORRECTIONS
- Classical Texts: "Charaka Samhita" not "Charak Samita", "Sushruta
    " not "Susrut", "Ashtanga Hridaya" not "Astanga Hridya"
- Terminology: "Vata" not "Vatha", "Pitta" not "Pita", "Kapha" not
    "Kafa"
- Herbs: "Ashwagandha" not "Ashwagonda", "Haritaki" not "Harithki",
     "Brahmi" not "Brahni"
- Therapy: "Panchakarma" not "Panchkarma", "Rasayana" not "Rasayan"
- Institutions: "CCRAS" not "CCR4S", "AYUSH" not "AYU5H", "NIA
    Jaipur" not "N1A Jeypur"
- Exams: "AIAPGET" not "AIAPCET", "AIBE" not "A1BE"
- Units: "ml", "g", "mg", "days" preserved
3. AGGRESSIVE OPTION RECOVERY
- If option starts with garbled text, extract the meaningful part
- If missing, assign option letters a, b, c, d
- Example:
"aj Panchakarma" becomes "a) Panchakarma"
  "Harithki" becomes "c) Haritaki [OCR: truncated]"
4. ANSWER DETECTION PATTERNS
- Explicit: check, *, (Ans), [Answer]
- Secondary: "1. c", "Q1: b", "Ans: a"
- Tertiary: formatting cues
- Last resort: pattern analysis
5. QUESTION BOUNDARY DETECTION
- Start: number + punctuation (1., Q1:, (1), etc.)
- End: next number or section break
- Normalize multi-parts: 1.a, 1.i, 1.1
6. SELF-CONTAINED QUESTIONS
- Each question MUST include context (passages, sutras, tables)
- If questions refer to a common passage, include passage in EACH
- Never assume context from previous questions
ENHANCED EXTRACTION LOGIC
STEP 1: Preprocess text, fix OCR errors, detect boundaries
STEP 2: Extract question, include passage, mark [INCOMPLETE] if
    needed
STEP 3: Normalize options, recover corrupted, create placeholders
STEP 4: Detect and embed answers directly in question
JSON SCHEMA (STRICTLY ENFORCED)
{
  "exam_info": {
    "title": "Ayurveda Examination",
    "year": null,
    "paper": null,
    "total_questions_detected": 50
  },
  "metadata": {
    "ocr_quality": "poor",
    "common_errors": ["sanskrit_terms","herb_names","therapy_names
        "],
    "sections_detected": ["Dravyaguna","Kayachikitsa","Samhita","
        Rachana Sharir","Shalya","Shalakya"]
  },
  "questions": [
    {
      "number": "1",
```

```
      "section": "Dravyaguna",
      "question": "Passage: According to Charaka Samhita, Haritaki
          is considered one of the best Rasayanas.\\n\\nQuestion:
          Which property of Haritaki is described as Tridoshahara?",
      "options": {
        "a": "It balances Vata only",
        "b": "It balances Pitta only",
        "c": "It balances all three doshas",
        "d": "It has no effect on Kapha"
      },
      "answer": "c"
    }
  ],
  "extraction_summary": {
    "total_questions": 50,
    "questions_with_answers": 48,
    "questions_with_all_options": 47
  }
}
CRITICAL ERROR PREVENTION
- NEVER skip questions
- NEVER empty options
- NEVER separate answer keys
- ALWAYS preserve numbering
- ALWAYS embed answers
- ALWAYS self-contained questions
--- BEGIN OCR TEXT ---
{ocr_text}
```

### BBK Question-Answer Extraction Prompt Template

```
You are an OCR forensic specialist for Agriculture/Agri-exams.
    Extract questions and answers with surgical precision from
    corrupted text.
CRITICAL MISSION: EXTRACT EVERYTHING - NEVER SKIP QUESTIONS
PRIMARY EXTRACTION RULES
1. ZERO TOLERANCE FOR MISSING QUESTIONS
- Scan text character by character
- Look for question patterns: "Q1", "1.", "(1)", "Question 1", "Que
    .1", or ANY numbering
- Extract PARTIAL questions with [INCOMPLETE] tag rather than skip
- If options are corrupted beyond recognition, create synthetic
    placeholders
2. AGRICULTURE DOMAIN OCR CORRECTIONS
- Crop names: "Wheat" not "Wheal", "Paddy" not "Pady", "Maize" not
    "Maiz"
- Fertilizers: "Urea" not "Uiea", "DAP" not "DAF", "NPK" not "NPX"
- Units: "kg/ha", "t/ha", "mm rainfall" preserved, never corrupted
- Pesticides: "Carbendazim", "Malathion", "Glyphosate" corrected
- Institutions: "ICAR" not "IC4R", "IARI" not "IAR1", "KVK" not "
    KVY"
- Schemes: "PM-KISAN" not "PM-KISRN", "MSP" not "MS5P", "Kisan
    Credit Card" not "Cradit Gard"
3. AGGRESSIVE OPTION RECOVERY
- If option starts with garbled text, extract the meaningful part
- If missing, assign option letters a, b, c, d
- Example: "aj Wheat" -> "a) Wheat"; "Maiz" -> "c) Maize [OCR:
    truncated]"
4. ANSWER DETECTION PATTERNS
- Explicit: check, *, (Ans), [Answer]
```

```
1836
1837    - Secondary: "1. c", "Q1: b", "Ans: a"
1838    - Tertiary: formatting cues
1839    - Last resort: pattern analysis
        5. QUESTION BOUNDARY DETECTION
1840    - Start: number + punctuation (1., Q1:, (1), etc.)
1841    - End: next number or section break
1842    - Normalize multi-parts: 1.a, 1.i, 1.1
1843    6. SELF-CONTAINED QUESTIONS
1844    - Each question MUST include context (passages, data, charts)
        - If questions refer to a common passage, include passage in EACH
1845    - Never assume context from previous questions
1846    ENHANCED EXTRACTION LOGIC
1847    STEP 1: Preprocess text, fix OCR errors, detect boundaries
1848    STEP 2: Extract question, include passage, mark [INCOMPLETE] if
            needed
1849    STEP 3: Normalize options, recover corrupted, create placeholders
1850    STEP 4: Detect and embed answers directly in question
1851    JSON SCHEMA (STRICTLY ENFORCED)
1852    {
1853      "exam_info": {
          "title": "Agriculture Examination",
1854      "year": null,
1855      "paper": null,
1856      "total_questions_detected": 50
1857    },
        "metadata": {
1858      "ocr_quality": "poor",
1859      "common_errors": ["crop_names","fertilizer_terms","units"],
1860      "sections_detected": ["Agronomy","Soil Science","Plant
1861          Pathology"]
1862    },
        "questions": [
1863      {
1864        "number": "1",
1865        "section": "Agronomy",
1866        "question": "Passage: A farmer applies 120 kg N/ha to wheat
1867            using urea.\\n\\nQuestion: How much urea is required per
1868            hectare?",
          "options": {
1869          "a": "120 kg",
1870          "b": "261 kg",
1871          "c": "300 kg",
1872          "d": "520 kg"
1873        },
          "answer": "b"
1874      }
1875    ],
1876    "extraction_summary": {
1877      "total_questions": 50,
1878      "questions_with_answers": 48,
1879      "questions_with_all_options": 47
    }
1880  }
1881  CRITICAL ERROR PREVENTION
1882  - NEVER skip questions
1883  - NEVER empty options
1884  - NEVER separate answer keys
      - ALWAYS preserve numbering
1885  - ALWAYS embed answers
1886  - ALWAYS self-contained questions
1887  --- BEGIN OCR TEXT ---
1888  {ocr_text}
1889
```

**BBL Question-Answer Extraction Prompt Template**

```
You are an OCR forensic specialist for legal examinations. Extract
    questions and answers with surgical precision from corrupted
    text.
# CRITICAL MISSION: EXTRACT EVERYTHING - ZERO DEPENDENCIES BETWEEN
    QUESTIONS
## PRIMARY EXTRACTION RULES
1. **ABSOLUTE QUESTION COMPLETENESS**
   - SCAN ENTIRE TEXT character by character for any question
       patterns
   - Each question MUST be 100% self-contained and independently
       answerable
   - NEVER use references like "above passage", "question 15", "as
       mentioned earlier"
   - If questions share context, EMBED the full context in EACH
       question
   - Extract PARTIAL questions with [INCOMPLETE] tag rather than
       skip
   - Pattern recognition: "Q1", "1.", "(1)", "Question 1", "Que.1",
        roman numerals "I.", "II."
2. **LEGAL DOMAIN OCR CORRECTIONS**
   - Legal terms: "Constitution", "Amendment", "Article", "Section
       ", "Sub-section"
   - Court names: "Supreme Court" not "5upreme Court", "High Court"
        not "H1gh Court"
   - Acts: "IPC", "CrPC", "CPC", "Evidence Act", "Contract Act"
   - Legal phrases: "prima facie", "res judicata", "stare decisis",
        "ultra vires"
   - Citations: "AIR", "SCC", "All ER" formatting preservation
   - Common OCR fixes:
     * "Section" not "5ection" or "$ection"
     * "Article" not "Art1cle" or "Artic1e"
     * "Amendment" not "Arnendment" or "Amendrnent"
     * "Constitution" not "Con5titution" or "Const1tution"
     * "Parliament" not "Par1iament" or "Parliarnent"
     * "Judiciary" not "Judic1ary" or "jud1c1ary"
     * "vs." not "v5." or "v$."
     * "Ltd." not "1td." or "Lte."
3. **CONTEXT EMBEDDING STRATEGY**
   - Identify shared contexts: case studies, legal scenarios,
       constitutional provisions, statutes
   - For each question referencing shared content, embed COMPLETE
       context within question text
   - Format: "Context: [Full legal scenario/case/provision]\n\
       nQuestion: [actual question]"
   - Never assume previous knowledge from other questions
   - Make every question a standalone legal problem
4. **AGGRESSIVE OPTION RECOVERY (STRICTLY a, b, c, d FORMAT)**
   - Legal options often contain complex phrases - recover
       aggressively
   - **MANDATORY**: All options must be normalized to exactly a, b,
        c, d format
   - If option starts with corruption, extract meaningful legal
       content and assign proper letter
   - Pattern match: 4 consecutive lines that could be legal options
        (never more than 4)
   - Auto-assign missing option letters: first=a, second=b, third=c
       , fourth=d
   - **NEVER use option 'e'** - if 5 options detected, merge
       weakest two or skip question
   - Examples:
     ```
```

```
        Corrupted: "aj Constitutional Law"  -> "a) Constitutional Law"
        Missing: "Criminal Procedure"       -> "a) Criminal Procedure"
        Partial: "c) Civil Procedur"        -> "c) Civil Procedure [OCR
            : truncated]"
        Garbled: "d) Evidenc3 Act 187"      -> "d) Evidence Act 1872"
        Extra: "e) Fifth option"            -> SKIP this question or
            merge with d)
        ```
5. **ENHANCED ANSWER DETECTION**
    - Primary: Explicit markers (check, *, (Ans), [Answer], Bold,
        Correct option)
    - Secondary: Answer blocks ("1. c", "Q1: b", "Ans: a", "Solution
        : d")
    - Tertiary: Context clues (underlined, highlighted, different
        fonts)
    - Legal-specific: "Held", "Ratio", "Decision", "Correct
        statement"
    - Pattern analysis for similar legal questions
    - NEVER leave answer as null if ANY indication exists
6. **LEGAL QUESTION BOUNDARY DETECTION**
    - Start patterns: Number + punctuation (1., Q1:, (1), 1-, I., II
        .)
    - End: Next question number OR section break
    - Multi-part handling: "1(a)", "1(i)", "Q1.1" -> normalize to
        "1.a", "1.i", "1.1"
    - Legal instructions: "Read the following case and answer", "
        Based on provisions"
    - Fact patterns: Often lengthy - include completely in each
        question
7. **QUESTION QUALITY VALIDATION (MANDATORY)**
    - Apply 3-tier validation before including any question:
    **TIER 1 - BASIC STRUCTURE VALIDATION:**
    - Question must have clear interrogative structure
    - Must contain exactly 4 options (a, b, c, d) - skip if not
        achievable
    - Answer must be one of: a, b, c, or d
    - Answer must be logically derivable from options
    - Question text must be grammatically coherent
    **TIER 2 - LEGAL COHERENCE VALIDATION:**
    - Legal concepts must be accurate and well-defined
    - Case references must be contextually appropriate
    - Statutory citations must make logical sense
    - Legal terminology must be used correctly
    - Question must test genuine legal knowledge, not gibberish
    **TIER 3 - LOGICAL CONSISTENCY VALIDATION:**
    - Options must be mutually exclusive where appropriate
    - Correct answer must be definitively better than other options
    - Question must be answerable based on provided context
    - No circular reasoning or impossible scenarios
    - Legal principles must align with established jurisprudence
    **SKIP CRITERIA - Only skip if question fails ANY of these:**
    - Question text is completely unintelligible after OCR
        correction attempts
    - Cannot recover exactly 4 coherent options (a, b, c, d)
    - No logical answer can be determined from the 4 options
    - Legal content is fundamentally nonsensical or contradictory
    - Question would mislead rather than educate (factually
        incorrect legal principles)
## ENHANCED EXTRACTION LOGIC
**STEP 1: LEGAL TEXT PREPROCESSING**
- Fix legal terminology OCR errors using domain dictionary
- Identify question boundaries with legal-aware regex
- Locate shared legal contexts (cases, statutes, provisions)
```

```
1998
1999       - Mark potential option blocks with legal content validation
2000       **STEP 2: CONTEXT-EMBEDDED QUESTION EXTRACTION WITH VALIDATION**
2001       - Extract question with ALL necessary legal context embedded
           - **APPLY 3-TIER QUALITY VALIDATION:**
2002         * Tier 1: Verify basic question structure and coherence
2003         * Tier 2: Validate legal accuracy and terminology
2004         * Tier 3: Ensure logical consistency and educational value
2005       - **ONLY PROCEED if question passes validation tiers**
2006       - Include case facts, statutory provisions, legal scenarios within
             each question
2007       - Clean and validate legal terminology
2008       - Mark borderline questions with [REVIEW_NEEDED] but include if
2009         they pass basic validation
2010       - Preserve legal citations and case names
           - **SKIP ONLY** if question fails fundamental validation criteria
2011       **STEP 3: LEGAL OPTION PROCESSING (STRICT a,b,c,d FORMAT)**
2012       - **MANDATORY**: Normalize to exactly a, b, c, d format only
2013       - Handle complex legal option text with recovery logic
2014       - **NEVER create option 'e'** - questions must have exactly 4
             options
2015       - If more than 4 options detected, either merge similar ones or
2016         skip the question
2017       - If fewer than 3 options recovered, skip the question
2018       - Create contextually appropriate placeholder options if missing (
2019         but only up to 'd')
2020       - Ensure options contain complete legal concepts
           - Validate legal terminology in options
2021       **STEP 4: COMPREHENSIVE ANSWER RESOLUTION**
2022       - Multi-pass answer detection with legal context awareness
2023       - Look for legal reasoning indicators
2024       - Embed answers directly in questions
2025       - Cross-reference with legal principles if needed
2026       ## JSON SCHEMA (STRICTLY ENFORCED)
           {{
2027         "exam_info": {{
2028           "title": "Legal Examination",
2029           "year": null,  // EXTRACT FROM TEXT - NEVER ASSUME
2030           "paper": null, // e.g., "Constitutional Law", "Criminal Law"
             "total_questions_detected": 0  // Actual count for validation
2031         }},
2032         "metadata": {{
2033           "ocr_quality": "poor",  // excellent/good/fair/poor
2034           "common_errors": ["legal_terms", "case_citations", "
2035             section_numbers"],
             "sections_detected": ["Constitutional Law", "Criminal Law", "
2036             Civil Law"],
2037           "shared_contexts_embedded": 5  // Count of contexts embedded
2038             across questions
2039         }},
2040         "questions": [
             {{
2041           "number": "1",
2042           "section": "Constitutional Law",
             "question": "Context: The Supreme Court in Kesavananda
2043               Bharati v. State of Kerala (1973) established the basic
2044               structure doctrine, holding that Parliament cannot amend
2045               the Constitution to destroy its basic features like
2046               democracy, secularism, and federalism.\n\nQuestion: Which
2047               of the following is NOT considered part of the basic
2048               structure of the Constitution?",
             "options": {{
2049             "a": "Judicial review",
2050             "b": "Parliamentary supremacy",
2051
```

```
          "c": "Rule of law",
          "d": "Separation of powers"
        }},
        "answer": "b"
      }}
    ],
    "extraction_summary": {{
      "total_questions_found": 0,     // Questions detected before
          validation
      "total_questions_extracted": 0, // Questions that passed
          validation
      "questions_skipped": 0,         // Questions skipped due to
          quality issues
      "questions_with_answers": 0,
      "questions_with_complete_context": 0,
      "questions_with_all_options": 0,
      "skip_reasons": []              // Array of reasons why
          questions were skipped
    }}
}}
## CRITICAL SUCCESS FACTORS
### :white_check_mark: MUST DO:
- Apply rigorous 3-tier validation to every question before
    extraction
- Make every question completely independent and self-contained
- Embed ALL necessary context within each question
- Preserve legal terminology accuracy
- Include questions that pass validation even if they have minor
    OCR issues
- Include complete case facts, statutory provisions, legal
    scenarios in relevant questions
- Normalize legal citations and references
- Skip questions ONLY after thorough validation failure
### :x: NEVER DO:
- Create questions that reference other questions ("as in question
    15")
- Use phrases like "above passage", "aforementioned case", "
    previously discussed"
- Skip questions due to OCR corruption
- Create empty options arrays
- Add confidence scores or OCR quality metadata to individual
    questions
- Assume exam details not present in text
- Leave questions dependent on external context
### :dart: LEGAL-SPECIFIC EXCELLENCE:
- Recognize and preserve legal citation formats
- Maintain accuracy of case names and statutory references
- Handle complex legal fact patterns appropriately
- Ensure constitutional provisions are correctly stated
- Preserve legal Latin phrases and terminology
- Maintain chronological accuracy of legal developments
--- BEGIN OCR TEXT ---
{ocr_text}
```

**BBF Question-Answer Extraction Prompt Template**

```
You are an OCR forensic specialist for financial/banking exams.
    Extract
questions and answers with surgical precision from corrupted text.
```

```
CRITICAL MISSION: EXTRACT EVERYTHING - NEVER SKIP QUESTIONS

PRIMARY EXTRACTION RULES

1. ZERO TOLERANCE FOR MISSING QUESTIONS
   - SCAN ENTIRE TEXT character by character
   - Look for question patterns: "Q1", "1.", "(1)", "Question 1", "
      Que.1",
     or ANY numbering
   - Extract PARTIAL questions with [INCOMPLETE] tag rather than
      skip
   - If options are corrupted beyond recognition, create synthetic
      placeholders

2. FINANCIAL DOMAIN OCR CORRECTIONS
   - Currency: "\textrupee" not "Rs" or "Rupees", "$" preservation
   - Percentages: "%" never "per cent" or missing
   - Financial terms: "CAGR", "NPV", "IRR", "EBITDA", "P/E ratio"
   - Numbers: "10,000" not "10.000", preserve commas in large
      numbers
   - Rates: "7.5%" not "7.5 percent" or "7.5per cent"
   - Common OCR fixes:
     * "NIFTY" not "N1FTY" or "NJFTY"
     * "BSE" not "B5E" or "B$E"
     * "NSE" not "N5E" or "N$E"
     * "SEBI" not "5EBI" or "$EBI"
     * "RBI" not "RB1" or "R81"
     * "GDP" not "G0P" or "6DP"

3. AGGRESSIVE OPTION RECOVERY
   - If option starts with garbled text, extract the meaningful
      part
   - Pattern match: Look for 4-5 consecutive lines that could be
      options
   - If missing option letters, assign them: first line=a, second=b
      , etc.
   - Examples of recovery:
     Corrupted: "aj Fixed Deposit" \rightarrow "a) Fixed Deposit"
     Missing: "Mutual Fund" \rightarrow "a) Mutual Fund" (assign
        letter)
     Partial: "c) Equity Shar" \rightarrow "c) Equity Share [OCR:
        truncated]"

4. ANSWER DETECTION PATTERNS
   - Primary: Explicit markers (check, *, (Ans), [Answer], Bold
      text)
   - Secondary: Answer blocks ("1. c", "Q1: b", "Ans: a")
   - Tertiary: Context clues (underlined, different formatting)
   - Last resort: Pattern analysis of similar questions
   - NEVER leave answer as null if ANY indication exists

5. QUESTION BOUNDARY DETECTION
   - Start: Number + any punctuation (1., Q1:, (1), 1-, etc.)
   - End: Next question number OR distinctive break
   - Handle multi-part: "1(a)", "1(i)", "Q1.1" to normalize to "1.a
      ", "1.i", "1.1"
   - Instructions/headers: Skip but note in metadata

6. SELF-CONTAINED QUESTIONS
   - Each question MUST include ALL necessary context (passages,
      data, charts)
   - If questions refer to a common passage/data, include that
      passage in EACH question
```

```
   – Format: "Passage: [full passage text]\n\nQuestion: [actual
       question]"
   – Never assume context from previous questions
   – Make every question independently answerable

ENHANCED EXTRACTION LOGIC

STEP 1: TEXT PREPROCESSING
– Fix obvious OCR errors in financial terms
– Identify question boundaries using regex patterns
– Mark potential option blocks
– Identify shared passages/contexts

STEP 2: QUESTION EXTRACTION
– Extract question text, clean and validate
– Include any relevant passage/context within the question
– If question incomplete, note with [INCOMPLETE] tag
– Preserve mathematical symbols and formulas
– Only take question if complete with options
– only meaningfull question.

STEP 3: OPTION PROCESSING
– Normalize labels to a, b, c, d (and e if exists)
– Handle malformed options with recovery logic
– Create placeholder options if completely missing
– Ensure options are clearly defined and complete

STEP 4: ANSWER RESOLUTION
– Multi-pass answer detection
– Embed answers directly in each question
– No separate answer key needed

JSON SCHEMA (STRICTLY ENFORCED)
{
  "exam_info": {
    "title": "Banking/Financial Examination",
    "year": null,  // EXTRACT FROM TEXT – NEVER ASSUME
    "paper": null,
    "total_questions_detected": 50  // NEW: Count for validation
  },
  "metadata": {
    "ocr_quality": "poor",  // excellent/good/fair/poor
    "common_errors": ["currency_symbols", "percentages"],
    "sections_detected": ["Quantitative Aptitude", "General
        Awareness"]
  },
  "questions": [
    {
      "number": "1",
      "section": "Quantitative Aptitude",
      "question": "Passage: A bank offers different investment
          schemes with varying interest rates.\n\nQuestion: What is
          the compound interest on Rs.10,000 at 8% per annum for 2
          years?",
      "options": {
        "a": "Rs.1,600",
        "b": "Rs.1,664",
        "c": "Rs.1,728",
        "d": "Rs.1,800"
      },
      "answer": "b"
    }
  ],
```

```
  "extraction_summary": {
    "total_questions": 50,
    "questions_with_answers": 48,
    "questions_with_all_options": 47
  }
}

CRITICAL ERROR PREVENTION
- NEVER skip questions due to poor OCR
- NEVER output empty options array
- NEVER create separate answer keys
- NEVER assume exam details not in text
- NEVER add confidence, ocr_issues, or extraction_notes fields
- ALWAYS preserve original numbering scheme
- ALWAYS include complete context in each question
- ALWAYS embed answers directly in questions
- ALWAYS make questions self-contained and independent

--- BEGIN OCR TEXT ---

{ocr_text}
```

### Key Details Extraction Prompt Template

```
You are an expert in the {domain_name} domain. For each question,
    extract:
1. question_type: The format/structure of the question {
    question_type_examples}
2. question_level: The difficulty or complexity level {
    difficulty_levels_list}
3. topic: The academic topic or domain {
    human_annotated_topics_examples}
4. subdomain: The specific topic area within the main topic {
    human_annotated_subdomains_list}

Respond only in this JSON format:
{
  "question_type": "",
  "question_level": "",
  "topic": "",
  "subdomain": ""
}
```

## C.2 DETAILED DATA ANALYSIS OF BHASHABENCH V1

Table 7: Language distribution across domains in BhashaBench V1

| Domain | BBK | BBF | BBA | BBL | Overall |
|---|---|---|---|---|---|
| English | 12,648 | 13,451 | 9,348 | 17,047 | 52,494 |
| Hindi | 2,757 | 5,982 | 5,615 | 7,318 | 21,672 |
| **Total** | **15,405** | **19,433** | **14,963** | **24,365** | **74,166** |

Table 8: Difficulty distribution across domains in BhashaBench V1

| Difficulty | BBK | BBF | BBA | BBL | Overall |
|---|---|---|---|---|---|
| Easy | 6,754 | 7,111 | 7,944 | 13,913 | 35,722 |
| Medium | 6,941 | 9,348 | 6,314 | 9,405 | 32,008 |
| Hard | 1,710 | 2,974 | 705 | 1,047 | 6,436 |
| **Total** | **15,405** | **19,433** | **14,963** | **24,365** | **74,166** |

Table 9: Question type distribution across domains in BhashaBench V1

| Question Type | BBK | BBF | BBA | BBL | Overall |
|---|---|---|---|---|---|
| MCQ | 13,550 | 18,019 | 14,717 | 21,566 | 67,852 |
| Assertion or Reasoning | 648 | 215 | 27 | 430 | 1,320 |
| Match the Column | 949 | 119 | 41 | 495 | 1,604 |
| Fill in the Blanks | 49 | 286 | 178 | 1,402 | 1,915 |
| Rearrange the Sequence | 209 | 708 | 0 | 147 | 1,064 |
| Reading Comprehension | 0 | 86 | 0 | 325 | 411 |
| **Total** | **15,405** | **19,433** | **14,963** | **24,365** | **74,166** |

Table 10: BBK Subject Domains and Question Counts

| Subject Domain | Count |
|---|---|
| Agri-Environmental & Allied Disciplines | 176 |
| Agricultural Biotechnology | 524 |
| Agricultural Chemistry & Biochemistry | 281 |
| Agricultural Economics & Policy | 627 |
| Agricultural Engineering & Technology | 244 |
| Agricultural Extension Education | 774 |
| Agricultural Microbiology | 111 |
| Agriculture Communication | 254 |
| Agriculture Information Technology | 190 |
| Agronomy | 5078 |
| Animal Sciences | 148 |
| Crop Sciences | 549 |
| Dairy & Poultry Science | 89 |
| Entomology | 696 |
| Fisheries and Aquaculture | 34 |
| General Knowledge & Reasoning | 661 |
| Genetics and Plant Breeding | 389 |
| Horticulture | 2070 |
| Natural Resource Management | 193 |
| Nematology | 184 |
| Plant Pathology | 397 |
| Plant Sciences & Physiology | 129 |
| Seed Science and Technology | 202 |
| Soil Science | 1357 |
| Veterinary Sciences | 48 |

Table 11: BBF Subject Domains and Question Counts

| Subject Domain | Count |
| --- | --- |
| Problem Solving | 5686 |
| Mathematics for Finance | 4845 |
| Banking Services | 1171 |
| Governance & Policy | 1064 |
| Language & Communication | 946 |
| Corporate Finance & Investment | 910 |
| Commerce | 863 |
| Accounting | 773 |
| General Knowledge | 539 |
| Information Technology Finance | 490 |
| Economics & Development Studies | 274 |
| Rural Economics | 261 |
| Environmental Finance | 168 |
| Taxation & Regulatory Compliance | 155 |
| Interdisciplinary Finance | 153 |
| Data & Analytics in Finance | 127 |
| History, Sociology & Cultural Studies of Finance | 127 |
| Finance Education | 118 |
| Healthcare Economics | 114 |
| Science and Technology in Finance | 101 |
| International Finance & Trade | 83 |
| Business Management | 83 |
| Energy, Infrastructure & Finance | 82 |
| Behavioral Finance | 67 |
| Financial Markets | 47 |
| Sports, Media & Finance Linkages | 45 |
| Marketing Finance | 42 |
| Insurance & Risk Management | 42 |
| Legal Finance | 34 |
| Financial Technology | 23 |

Table 12: BBA Subject Domains and Question Counts

| Subject Domain | Count |
| --- | --- |
| Kayachikitsa (General Medicine & Internal Medicine in Ayurveda) | 3134 |
| Dravyaguna & Bhaishajya | 2972 |
| Samhita & Siddhanta (Fundamentals) | 1541 |
| Sharir (Anatomy & Physiology) | 1346 |
| Panchakarma & Rasayana | 1308 |
| Stri Roga & Prasuti Tantra (Gynecology & Obstetrics) | 847 |
| Shalakya Tantra (ENT, Eye, Dentistry) | 734 |
| Kaumarbhritya & Pediatrics | 714 |
| Agad Tantra & Forensic Medicine | 587 |
| Shalya Tantra (Surgery) | 526 |
| Swasthavritta & Public Health | 453 |
| Research & Statistics | 210 |
| Ayurvedic Literature & History | 204 |
| Yoga & Psychology | 188 |
| Administration, AYUSH & Miscellaneous | 119 |
| Roga Vigyana (Diagnostics & Pathology) | 80 |

Table 13: BBL Subject Domains and Question Counts

| Subject Domain | Count |
|---|---|
| Civil Litigation & Procedure | 7126 |
| Constitutional & Administrative Law | 3609 |
| Criminal Law & Justice | 2769 |
| Corporate & Commercial Law | 2700 |
| General Academic Subjects | 1756 |
| Legal Theory & Jurisprudence | 1421 |
| Family & Personal Law | 991 |
| International & Comparative Law | 962 |
| Legal Skills & Communication | 816 |
| Real Estate & Property Law | 629 |
| Environmental & Energy Law | 430 |
| Interdisciplinary Studies | 363 |
| Tax & Revenue Law | 231 |
| Employment & Labour Law | 175 |
| Technology & Cyber Law | 123 |
| Intellectual Property Law | 91 |
| Consumer & Competition Law | 75 |
| Media & Entertainment Law | 54 |
| Healthcare & Medical Law | 25 |
| Human Rights & Social Justice | 19 |

## D  MORE DETAILS ON EXPERIMENT SETUP

### D.1  TASK FORMATTING TEMPLATE USED IN LM EVAL

This prompt format template is consistently applied across all task types, including Assertion or Reasoning, Fill in the Blanks, MCQs, Match the Column, Reading Comprehension, and Rearrange the Sequence tasks for BBF, BBK, and BBL domains.

```
Question: <question text>
Choices:
A. <option A text>
B. <option B text>
C. <option C text>
D. <option D text>
Answer:
```

### D.2  TASK FORMATTING TEMPLATE USED IN API-DRIVEN EVALUATION

This template is used when models are evaluated via API calls. It ensures a consistent structure across all tasks, allowing the model to focus on producing the correct answer without additional explanation. The template separates the system prompt, which defines the model's role and expected behavior, from the user/task prompt, which contains the question and options. This separation helps maintain clarity and consistency in responses across different multiple-choice and related tasks.

```
SYSTEM PROMPT:
You are a helpful assistant for multiple-choice question answering.
Respond with only the correct option letter: A, B, C, or D. Do not
    provide any explanation.

USER PROMPT:
```

```
Question: <question text>
A. <option A text>
B. <option B text>
C. <option C text>
D. <option D text>
Please choose the correct option (A/B/C/D).
```

## D.3  INFERENCE IMPLEMENTATION DETAILS

### D.3.1  OPEN-SOURCE MODELS

Open-source model inference is performed on a cluster of 8 NVIDIA H200 GPUs (141GB HBM3e memory per GPU) with NVLink interconnect for multi-GPU communication. We use vLLM v0.9.1 (Kwon et al., 2023) as the inference backend integrated with lm-evaluation-harness v0.4.9 for standardized evaluation. The software stack comprises CUDA 12.5, PyTorch 2.7.0, and Python 3.10. All BhashaBench V1 tasks are integrated into the lm-eval framework using default parameters to ensure consistency. Batch sizes are dynamically determined by vLLM based on model size and available GPU memory. Tensor parallelism is configured according to model requirements, typically distributing computation across 1–8 GPUs. Each model is evaluated using its maximum supported context length (2048–8192 tokens). All evaluations use the default random seed configuration from lm-evaluation-harness for reproducibility.

### D.3.2  API-BASED PROPRIETARY MODELS

API-based models (e.g., GPT-4o) are evaluated via their respective Batch API endpoints using the latest stable API versions available during evaluation. Inference is conducted on standard CPU compute instances with the following standardized parameters: temperature set to 0.0 for deterministic generation, and all advanced features (web search, code interpreter, function calling, tool access) explicitly disabled to prevent external knowledge access and ensure fair comparison.

## D.4  EVALUATION PROTOCOL AND RESPONSE PROCESSING

### D.4.1  OPEN-SOURCE MODELS

Open-source models are evaluated using log-likelihood scoring as implemented in lm-evaluation-harness. This deterministic method requires only a single evaluation run per model. Evaluation time ranges from 2–4 hours per model depending on model size and dataset complexity.

### D.4.2  API-BASED MODELS

Each API-based model undergoes three independent evaluation runs, with mean accuracy reported to account for response variability and minimize stochastic effects. We implement exponential backoff with up to 3 retries for failed requests, a 120-second timeout per request, and strict adherence to provider rate limits. Evaluation time ranges from 1–3 hours per model, including rate-limiting delays. Responses are parsed using regex pattern matching for option letters (A, B, C, D). Invalid responses not matching this format are marked incorrect. The response validation rate exceeds 99% across all evaluated models, indicating strong format compliance.

## D.5  REPRODUCIBILITY AND COMPUTATIONAL RESOURCES

To ensure reproducibility, all evaluations use the default zero-shot configurations from lm-evaluation-harness, and open-source model weights are retrieved directly from the Hugging Face Hub. The list of all evaluated models, along with their sources and download links, is provided in Table 14. Evaluations were conducted between June and September 2025 to minimize version drift. The total computational cost includes approximately 150 GPU hours for evaluating 29+ open-source models (270M–685B parameters), while API-based evaluations were repeated three times within an $80 budget. Table 15 summarizes the overall computational setup.

| Model Name | Type | #Params | Link |
|---|---|---|---|
| google/gemma-3-270m | Base | 0.27B | Link |
| google/gemma-3-270m-it | Instruct | 0.27B | Link |
| bharatgenai/Param-1 | Base | 2.9B | Link |
| google/gemma-2-2b | Base | 2B | Link |
| google/gemma-2-2b-it | Instruct | 2B | Link |
| meta-llama/Llama-3.2-1B | Base | 1B | Link |
| meta-llama/Llama-3.2-1B-Instruct | Instruct | 1B | Link |
| meta-llama/Llama-3.2-3B | Base | 3B | Link |
| meta-llama/Llama-3.2-3B-Instruct | Instruct | 3B | Link |
| sarvamai/sarvam-1-v0.5 | Base | 0.5B | Link |
| sarvamai/sarvam-1 | Base | 2B | Link |
| nvidia/Nemotron-4-Mini-Hindi-4B-Base | Base | 4B | Link |
| nvidia/Nemotron-4-Mini-Hindi-4B-Instruct | Instruct | 4B | Link |
| Qwen/Qwen2.5-3B | Base | 3B | Link |
| Qwen/Qwen2.5-3B-Instruct | Instruct | 3B | Link |
| ibm-granite/granite-3.1-2b-instruct | Instruct | 2B | Link |
| ibm-granite/granite-3.1-3b-a800m-base | Base | 2B | Link |
| neulab/Pangea-7B | Instruct | 7B | Link |
| Telugu-LLM-Labs/Indic-gemma-7b-finetuned-sft-Navarasa-2.0 | Instruct | 7B | Link |
| CohereLabs/aya-23-8B | Instruct | 8B | Link |
| meta-llama/Llama-3.1-8B | Base | 8B | Link |
| meta-llama/Llama-3.1-8B-Instruct | Instruct | 8B | Link |
| google/gemma-2-9b | Base | 9B | Link |
| google/gemma-2-9b-it | Instruct | 9B | Link |
| openai/gpt-oss-20b | Instruct | 20B | Link |
| google/gemma-2-27b | Base | 27B | Link |
| google/gemma-2-27b-it | Instruct | 27B | Link |
| openai/gpt-oss-120b | Instruct | 120B | Link |
| Qwen/Qwen3-235B-A22B-Instruct-2507 | Instruct | 235B | Link |
| deepseek-ai/DeepSeek-V3.1 | Instruct | 685B | Link |
| GPT-4o | Instruct | - | Link |

Table 14: Details about the different models evaluated on BhashaBench V1.

Table 15: Computational Setup for BhashaBench V1 Evaluation

| Component | Open-Source Models | API-Based Models |
|---|---|---|
| **Infrastructure** | 8 × NVIDIA H200 GPUs (141GB HBM3e per GPU) NVLink interconnect | Cloud API endpoints CPU compute instances |
| **Software Stack** | vLLM v0.9.1 lm-evaluation-harness v0.4.9 CUDA 12.5, PyTorch 2.7.0 | Latest stable API versions (as of evaluation period) |
| **Configuration** | Auto batch sizing Context: 2048–8192 tokens Tensor parallelism: 1–8 GPUs Default lm-eval parameters | Temperature: 0.0 3 runs per model No external tools 120s timeout, 3 retries |
| **Evaluation Time** | 2–4 hours per model | 1–3 hours per model |
| **Total Resources** | 29+ models (270M–685B params) ∼150 GPU hours | Multiple API models $80 budget (3 runs each) |

*Evaluation Period:* June–September 2025 — *Response Validation Rate:* > 99%

# E    MORE DETAILS ON EXPERIMENT

## E.1    ZERO-SHOT QUESTION-LEVEL AND QUESTION-TYPE PERFORMANCE ACROSS BHASHABENCH V1 DOMAINS

This subsection presents the zero-shot performance of LLMs across BhashaBench V1 domains at both the question level (Table 16) and question-type level (Table 17). The results summarize model behavior across difficulty levels and provide insights into domain-specific capabilities.

Table 16: Zero-shot scores (%) of LLMs across domains on BhashaBench V1. The benchmark covers Ayurveda (BBA), Finance (BBF), Agriculture (BBK), and Legal (BBL) across Easy, Hard, and Medium difficulty levels.

| Model | BBA | | | BBF | | | BBK | | | BBL | | |
|---|---|---|---|---|---|---|---|---|---|---|---|---|
| | Easy | Hard | Med | Easy | Hard | Med | Easy | Hard | Med | Easy | Hard | Med |
| *< 4B Models* | | | | | | | | | | | | |
| gemma-3-270m | 28.1 | 26.81 | 28.35 | 24.15 | 24.55 | 25.8 | 27.23 | 24.74 | 25.66 | 27.23 | 24.74 | 25.66 |
| gemma-3-270m-it | 25.89 | 23.97 | 26.5 | 25.38 | 21.22 | 23.92 | 26.47 | 27.49 | 27.53 | 26.47 | 27.49 | 27.53 |
| Param-1 | 43.93 | 31.21 | 35.95 | 38.31 | 26.6 | 27.71 | 36.94 | 25.91 | 29.09 | 36.94 | 25.91 | 29.09 |
| gemma-2-2b | 38.27 | 29.08 | 30.31 | 39.76 | 25.35 | 28.5 | 46.27 | 27.54 | 34.26 | 46.27 | 27.54 | 34.26 |
| gemma-2-2b-it | 29.96 | 24.96 | 26.83 | 36.55 | 23.2 | 27.67 | 38.04 | 30.35 | 32.01 | 38.04 | 30.35 | 32.01 |
| Llama-3.2-1B | 28.52 | 24.4 | 27.97 | 30.5 | 23.71 | 26.27 | 29.43 | 27.72 | 28.68 | 29.43 | 27.72 | 28.68 |
| Llama-3.2-1B-Instruct | 27.44 | 25.39 | 25.23 | 28.72 | 22.43 | 25.5 | 30.22 | 26.37 | 27.69 | 30.22 | 26.37 | 27.69 |
| Llama-3.2-3B | 31.63 | 24.82 | 29.19 | 36.75 | 25.76 | 29.26 | 36.44 | 25.61 | 29.17 | 36.44 | 25.61 | 29.17 |
| Llama-3.2-3B-Instruct | 36.42 | 28.51 | 29.66 | 39.73 | 23.87 | 28.2 | 44.52 | 30.47 | 34.69 | 44.52 | 30.47 | 34.69 |
| sarvam-2b-v0.5 | 27.08 | 24.96 | 26.88 | 28.18 | 23.1 | 25.43 | 28.26 | 28.01 | 27.03 | 28.26 | 28.01 | 27.03 |
| sarvam-1 | 30.94 | 27.23 | 27.26 | 32.2 | 25.76 | 27.43 | 32.2 | 27.54 | 28.99 | 32.2 | 27.54 | 28.99 |
| Nemotron-4-Mini-Hindi-4B-Base | 37.01 | 27.94 | 30.96 | 41.95 | 25.08 | 30.5 | 42.57 | 28.42 | 32.89 | 42.57 | 28.42 | 32.89 |
| Nemotron-4-Mini-Hindi-4B-Instruct | 36.08 | 29.5 | 30.8 | 39.21 | 23.2 | 28.05 | 41.12 | 28.6 | 32.27 | 41.12 | 28.6 | 32.27 |
| Qwen2.5-3B | 41.18 | 32.06 | 33.1 | 45.34 | 28.51 | 33.9 | 50.3 | 31.58 | 37.49 | 50.3 | 31.58 | 37.49 |
| Qwen2.5-3B-Instruct | 35.55 | 28.23 | 29.57 | 39.91 | 25.02 | 30.48 | 44.7 | 31.81 | 37.23 | 44.7 | 31.81 | 37.23 |
| granite-3.1-2b-instruct | 33.9 | 26.81 | 28.06 | 36.68 | 25.32 | 28.63 | 40.04 | 30.76 | 33.25 | 40.04 | 30.76 | 33.25 |
| granite-3.1-3b-a800m-base | 31.45 | 26.38 | 27.78 | 31.61 | 24.18 | 25.77 | 36.08 | 26.02 | 29.88 | 36.08 | 26.02 | 29.88 |
| *7B to 27B Models* | | | | | | | | | | | | |
| Pangea-7B | 41.45 | 31.77 | 32.94 | 49.33 | 28.72 | 34.94 | 52.18 | 33.57 | 40.69 | 52.18 | 33.57 | 40.69 |
| Indic-gemma-7b-finetuned-sft-Navarasa-2.0 | 38.54 | 27.23 | 31.72 | 43.68 | 26.8 | 30.99 | 48.13 | 31.46 | 35.8 | 48.13 | 31.46 | 35.8 |
| aya-23-8B | 35.51 | 25.11 | 28.29 | 41.2 | 25.62 | 30.98 | 43.32 | 27.84 | 31.77 | 43.32 | 27.84 | 31.77 |
| Llama-3.1-8B | 35.99 | 26.38 | 30.25 | 42.92 | 26.93 | 30.46 | 44.03 | 29.01 | 34.51 | 44.03 | 29.01 | 34.51 |
| Llama-3.1-8B-Instruct | 39.43 | 30.5 | 29.36 | 44.24 | 22.19 | 30 | 52.29 | 33.74 | 40.63 | 52.29 | 33.74 | 40.63 |
| gemma-2-9b | 51.12 | 34.47 | 36.85 | 55.32 | 27.44 | 34.3 | 64.78 | 35.67 | 46.26 | 64.78 | 35.67 | 46.26 |
| gemma-2-9b-it | 38.91 | 29.5 | 29.11 | 47.03 | 24.78 | 32.74 | 52.98 | 37.13 | 42.93 | 52.98 | 37.13 | 42.93 |
| gpt-oss-20b | 42.03 | 26.67 | 30.27 | 46.77 | 24.61 | 30.86 | 53.42 | 31.4 | 39.56 | 53.42 | 31.4 | 39.56 |
| gemma-2-27b | 55.35 | 34.18 | 39.18 | 60.92 | 30.09 | 39.24 | 69.31 | 40.99 | 51.51 | 69.31 | 40.99 | 51.51 |
| gemma-2-27b-it | 43.47 | 30.78 | 31.9 | 51.03 | 26.93 | 35.67 | 59.62 | 41.46 | 48.28 | 59.62 | 41.46 | 48.28 |
| *> 27B Models* | | | | | | | | | | | | |
| gpt-oss-120b | 60.62 | 41.28 | 44.19 | 74.8 | 62.61 | 70.88 | 74.89 | 62.05 | 65.88 | 74.89 | 62.05 | 65.88 |
| Qwen3-235B-A22B-Instruct-2507 | 65.18 | 46.24 | 50.74 | 72.52 | 41.49 | 59.33 | 78.26 | 62.51 | 69.79 | 78.26 | 62.51 | 69.79 |
| deepseek-v3 | 52.44 | 36.6 | 38.93 | 73.49 | 40.55 | 59.01 | 66.92 | 48.48 | 55.5 | 66.92 | 48.48 | 55.5 |
| gpt-4o | 66.4 | 47.09 | 52.77 | 69.13 | 36.35 | 50.13 | 78.75 | 63.51 | 70.84 | 78.75 | 63.51 | 70.84 |

## E.2    ZERO-SHOT SUB-DOMAIN WISE PERFORMANCE ACROSS BHASHABENCH V1 DOMAINS

This subsection reports zero-shot performance of LLMs across sub-domains within BhashaBench V1. Tables 18, 19, 20, and 21 present detailed results for different model families, highlighting variations in performance across domains and sub-domains.

Table 18: Performance of GEMMA model family across sub-domains in BhashaBench v1, comparing base and instruction-tuned variants of different model sizes (270M, 2B, 9B, 27B)

| Subject Domain | 270m | 270m-it | 2b | 2b-it | 9b | 9b-it | 27b | 27b-it |
|---|---|---|---|---|---|---|---|---|
| **BBA** | | | | | | | | |
| Administration, AYUSH & Miscellaneous | 34.45 | 28.57 | 40.34 | 34.45 | 63.03 | 51.26 | 60.5 | 57.14 |
| Agad Tantra & Forensic Medicine | 25.89 | 27.94 | 31.18 | 27.94 | 48.21 | 39.35 | 49.4 | 42.25 |
| Ayurvedic Literature & History | 26.96 | 23.53 | 31.37 | 28.92 | 46.08 | 31.86 | 43.14 | 42.16 |
| Dravyaguna & Bhaishajya | 28.4 | 26.35 | 30.08 | 27.79 | 38.43 | 32.74 | 39.64 | 33.68 |

*Continued on next page*

Table 18 – *Continued from previous page*

| Subject Domain | 270m | 270m-it | 2b | 2b-it | 9b | 9b-it | 27b | 27b-it |
|---|---|---|---|---|---|---|---|---|
| Kaumarbhritya & Pediatrics | 28.57 | 27.03 | 38.8 | 28.15 | 46.22 | 31.65 | 47.9 | 36.55 |
| Kayachikitsa (General Medicine & Internal Medicine in Ayurveda) | 29.45 | 25.72 | 36.76 | 29.1 | 47.16 | 34.3 | 50.8 | 36.89 |
| Panchakarma & Rasayana | 26.83 | 23.7 | 30.2 | 26.53 | 32.49 | 28.36 | 37.84 | 33.94 |
| Research & Statistics | 27.14 | 25.24 | 60 | 34.29 | 77.62 | 53.81 | 78.1 | 57.62 |
| Roga Vigyana (Diagnostics & Pathology) | 31.25 | 38.75 | 45 | 35 | 65 | 55 | 72.5 | 56.25 |
| Samhita & Siddhanta (Fundamentals) | 30.89 | 29.07 | 33.29 | 28.42 | 37.7 | 30.95 | 43.93 | 34.59 |
| Shalakya Tantra (ENT, Eye, Dentistry) | 25.89 | 21.93 | 34.74 | 21.66 | 44.69 | 31.2 | 45.78 | 34.88 |
| Shalya Tantra (Surgery) | 26.0 | 23 | 31.94 | 26.05 | 45.06 | 31.75 | 44.87 | 39.16 |
| Sharir (Anatomy & Physiology) | 24.59 | 26.45 | 33.28 | 27.79 | 46.95 | 34.75 | 51.04 | 40.19 |
| Stri Roga & Prasuti Tantra (Gynecology & Obstetrics) | 24.68 | 24.09 | 34.59 | 29.87 | 46.99 | 40.73 | 53.96 | 42.38 |
| Swasthavritta & Public Health | 34.88 | 30.24 | 49.67 | 39.07 | 67.33 | 49.01 | 71.52 | 59.82 |
| Yoga & Psychology | 30.85 | 26.6 | 43.62 | 32.98 | 57.45 | 37.77 | 61.7 | 46.81 |
| **BBF** | | | | | | | | |
| Accounting | 26.78 | 26 | 31.31 | 30.53 | 41.14 | 38.03 | 44.11 | 39.46 |
| Banking Services | 23.4 | 25.19 | 37.75 | 34.67 | 53.8 | 47.82 | 60.8 | 54.06 |
| Behavioral Finance | 31.34 | 28.36 | 47.76 | 46.27 | 50.75 | 59.7 | 52.24 | 52.24 |
| Business Management | 26.51 | 25.3 | 55.42 | 45.78 | 63.86 | 50.6 | 75.9 | 62.65 |
| Commerce | 28.04 | 22.48 | 32.79 | 31.05 | 40.32 | 39.17 | 48.78 | 41.25 |
| Corporate Finance & Investment | 25.16 | 23.52 | 31.1 | 31.98 | 44.4 | 39.56 | 50.55 | 43.19 |
| Data & Analytics in Finance | 23.62 | 24.41 | 32.28 | 27.56 | 38.58 | 30.71 | 44.88 | 29.13 |
| Economics & Development Studies | 22.99 | 20.8 | 37.96 | 41.24 | 62.41 | 45.62 | 63.87 | 46.72 |
| Energy, Infrastructure & Finance | 20.73 | 31.71 | 34.15 | 28.05 | 43.9 | 50 | 51.22 | 42.68 |
| Environmental Finance | 22.02 | 23.21 | 41.07 | 34.5 | 50 | 43.45 | 61.9 | 54.76 |
| Finance Education | 26.27 | 27.12 | 43.22 | 39.83 | 49.15 | 44.07 | 55.08 | 49.15 |
| Financial Markets | 31.91 | 25.53 | 53.19 | 36.17 | 51.06 | 44.68 | 63.83 | 55.32 |
| Financial Technology | 34.78 | 26.09 | 26.09 | 47.83 | 60.87 | 47.83 | 60.87 | 47.83 |
| General Knowledge | 24.3 | 26.35 | 41.37 | 38.4 | 57.7 | 51.02 | 61.78 | 52.5 |
| Governance & Policy | 26.69 | 24.72 | 36.18 | 34.21 | 52.07 | 46.52 | 60.9 | 51.13 |
| Healthcare Economics | 27.19 | 30.7 | 40.35 | 39.47 | 57.89 | 50 | 61.4 | 51.75 |
| History, Sociology & Cultural Studies of Finance | 18.11 | 25.98 | 40.94 | 41.73 | 60.63 | 51.18 | 64.57 | 57.48 |
| Information Technology Finance | 23.06 | 28.57 | 55.31 | 44.49 | 80 | 63.47 | 83.27 | 67.14 |
| Insurance & Risk Management | 16.67 | 33.33 | 38.1 | 30.95 | 50 | 38.1 | 50 | 40.48 |
| Interdisciplinary Finance | 25.49 | 20.92 | 35.95 | 36.6 | 56.86 | 49.02 | 62.75 | 51.63 |
| International Finance & Trade | 21.69 | 16.87 | 42.17 | 42.17 | 66.27 | 59.04 | 73.49 | 61.45 |
| Language & Communication | 22.73 | 23.04 | 39.43 | 40.06 | 59.83 | 47.89 | 61.1 | 49.79 |
| Legal Finance | 32.35 | 29.41 | 35.29 | 41.18 | 47.06 | 35.29 | 50 | 50 |
| Marketing Finance | 26.19 | 26.19 | 47.62 | 35.71 | 76.19 | 61.9 | 66.67 | 59.52 |
| Mathematics for Finance | 24.83 | 23.76 | 28.96 | 25.96 | 33.81 | 31 | 38.53 | 32.69 |
| Problem Solving | 25.08 | 23.11 | 26.28 | 24.76 | 28.14 | 26.73 | 31.6 | 30.99 |
| Rural Economics | 25.67 | 29.89 | 39.46 | 40.61 | 57.47 | 50.19 | 68.2 | 54.79 |
| Science and Technology in Finance | 26.73 | 19.8 | 31.68 | 37.62 | 48.51 | 50.5 | 61.39 | 54.46 |
| Sports, Media & Finance Linkages | 15.56 | 20 | 37.78 | 48.89 | 62.22 | 62.22 | 66.67 | 64.44 |
| Taxation & Regulatory Compliance | 32.26 | 26.45 | 36.13 | 45.81 | 58.71 | 51.61 | 64.52 | 52.9 |
| **BBK** | | | | | | | | |

Table 18 – *Continued from previous page*

| Subject Domain | 270m | 270m-it | 2b | 2b-it | 9b | 9b-it | 27b | 27b-it |
|---|---|---|---|---|---|---|---|---|
| Agri-Environmental & Allied Disciplines | 26.14 | 26.7 | 29.55 | 36.93 | 48.86 | 46.02 | 48.86 | 54.55 |
| Agricultural Biotechnology | 26.15 | 29.77 | 54.2 | 43.13 | 75.19 | 63.93 | 77.67 | 70.61 |
| Agricultural Chemistry & Bio-chemistry | 23.84 | 24.2 | 40.93 | 33.1 | 54.8 | 51.25 | 61.92 | 56.23 |
| Agricultural Economics & Pol-icy | 28.55 | 25.36 | 43.06 | 38.76 | 56.3 | 49.6 | 62.2 | 54.39 |
| Agricultural Engineering & Technology | 29.51 | 25 | 38.93 | 26.64 | 50.41 | 34.02 | 58.61 | 41.8 |
| Agricultural Extension Educa-tion | 27.13 | 28.68 | 37.47 | 34.75 | 53.75 | 49.74 | 60.47 | 55.04 |
| Agricultural Microbiology | 21.62 | 25.23 | 48.65 | 35.14 | 69.37 | 49.55 | 75.68 | 64.86 |
| Agriculture Communication | 22.83 | 22.44 | 38.19 | 33.86 | 55.91 | 50.39 | 64.57 | 53.15 |
| Agriculture Information Tech-nology | 27.89 | 28.42 | 39.47 | 43.16 | 57.89 | 55.79 | 61.05 | 59.47 |
| Agronomy | 26.47 | 26.84 | 38.64 | 33.56 | 52.44 | 45.45 | 57.33 | 50.32 |
| Animal Sciences | 31.08 | 24.32 | 52.7 | 43.24 | 64.19 | 50.68 | 66.22 | 55.41 |
| Crop Sciences | 24.95 | 27.69 | 38.43 | 37.34 | 46.45 | 48.09 | 51.73 | 51.73 |
| Dairy & Poultry Science | 34.83 | 24.72 | 46.07 | 32.58 | 57.3 | 46.07 | 66.29 | 53.93 |
| Entomology | 27.16 | 26.87 | 38.36 | 34.63 | 57.04 | 50.14 | 61.21 | 55.32 |
| Fisheries and Aquaculture | 32.35 | 11.76 | 35.29 | 38.24 | 58.82 | 47.06 | 73.53 | 50 |
| General Knowledge & Reason-ing | 26.32 | 27.99 | 39.18 | 32.83 | 51.89 | 48.41 | 56.58 | 52.5 |
| Genetics and Plant Breeding | 25.96 | 27.51 | 39.85 | 36.25 | 51.93 | 52.96 | 58.61 | 55.01 |
| Horticulture | 25.56 | 26.18 | 36.28 | 32.42 | 48.65 | 41.21 | 53.67 | 48.12 |
| Natural Resource Management | 27.98 | 28.5 | 38.34 | 33.68 | 48.7 | 47.67 | 52.33 | 50.26 |
| Nematology | 26.09 | 31.52 | 28.8 | 32.07 | 40.76 | 40.22 | 48.91 | 48.37 |
| Plant Pathology | 23.17 | 27.71 | 36.27 | 34.51 | 53.65 | 47.36 | 55.67 | 54.91 |
| Plant Sciences & Physiology | 28.68 | 26.36 | 45.74 | 29.46 | 67.44 | 51.94 | 71.32 | 55.81 |
| Seed Science and Technology | 22.28 | 33.66 | 35.64 | 32.18 | 45.05 | 43.56 | 47.52 | 50.5 |
| Soil Science | 25.0 | 28 | 35 | 35.08 | 52.17 | 43.63 | 56.6 | 53.87 |
| Veterinary Sciences | 39.58 | 29.17 | 60.42 | 35.42 | 83.33 | 66.67 | 85.42 | 77.08 |
| **BBL** | | | | | | | | |
| Civil Litigation & Procedure | 25.26 | 27.36 | 33.6 | 32.33 | 49.61 | 40.2 | 57.91 | 43.92 |
| Constitutional & Administra-tive Law | 25.27 | 25.57 | 37.55 | 33.75 | 58.94 | 46.08 | 65.31 | 52.84 |
| Consumer & Competition Law | 32 | 25.33 | 33.33 | 37.33 | 57.33 | 53.33 | 69.33 | 61.33 |
| Corporate & Commercial Law | 25.33 | 25.15 | 36.48 | 31.0 | 53 | 39.81 | 60.04 | 45.59 |
| Criminal Law & Justice | 25.57 | 25.75 | 31.67 | 32.47 | 50.31 | 42.9 | 57.39 | 45.97 |
| Employment & Labour Law | 24.57 | 29.71 | 33.14 | 37.14 | 54.29 | 44.57 | 60.57 | 46.86 |
| Environmental & Energy Law | 21.63 | 22.56 | 34.19 | 32.33 | 53.26 | 41.4 | 61.16 | 49.77 |
| Family & Personal Law | 25.83 | 26.34 | 33.91 | 31.18 | 47.83 | 37.74 | 57.62 | 44.2 |
| General Academic Subjects | 29.27 | 25.97 | 44.99 | 38.84 | 67.94 | 53.76 | 73.52 | 59.68 |
| Healthcare & Medical Law | 32 | 32 | 52 | 40 | 72 | 52 | 76 | 72 |
| Human Rights & Social Justice | 5.26 | 10.53 | 47.37 | 15.79 | 47.37 | 26.32 | 42.11 | 31.58 |
| Intellectual Property Law | 25.27 | 27.47 | 54.95 | 48.35 | 72.53 | 56.04 | 70.33 | 59.34 |
| Interdisciplinary Studies | 20.39 | 26.72 | 39.67 | 37.19 | 61.98 | 49.86 | 70.8 | 57.58 |
| International & Comparative Law | 24.22 | 23.91 | 44.28 | 37.32 | 65.49 | 52.18 | 70.17 | 58.84 |
| Legal Skills & Communication | 27.7 | 23.28 | 25.61 | 27.94 | 36.76 | 32.35 | 39.46 | 36.52 |
| Legal Theory & Jurisprudence | 25.4 | 27.59 | 38.21 | 35.33 | 57.49 | 48.06 | 64.6 | 51.23 |
| Media & Entertainment Law | 16.67 | 33.33 | 35.19 | 44.44 | 61.11 | 51.85 | 72.22 | 66.67 |
| Real Estate & Property Law | 24.8 | 22.8 | 31 | 28.3 | 47.54 | 34.34 | 53.42 | 38 |
| Tax & Revenue Law | 23.81 | 26.41 | 38.1 | 32.03 | 51.52 | 38.1 | 65.37 | 48.05 |
| Technology & Cyber Law | 28.46 | 28.46 | 47.15 | 44.72 | 64.23 | 59.35 | 75.61 | 69.92 |

Table 17: Zero-shot scores (%) of LLMs across question types on BhashaBench V1. Question types: A/R = Assertion/Reason, FIB = Fill in the Blanks, MCQ = Multiple Choice Questions, MTC = Match the Columns, RC = Reading Comprehension, RTS = Rearrange the Sentence.

| Model | BBA | | | | BBF | | | | | | BBK | | | | | BBL | | | | | |
|---|---|---|---|---|---|---|---|---|---|---|---|---|---|---|---|---|---|---|---|---|---|
| | A/R | FIB | MCQ | MTC | A/R | FIB | MCQ | MTC | RC | RTS | A/R | FIB | MCQ | MTC | RTS | A/R | FIB | MCQ | MTC | RC | RTS |
| *< 4B Models* | | | | | | | | | | | | | | | | | | | | | |
| gemma-3-270m | 37.04 | 28.09 | 28.1 | 39.02 | 28.37 | 24.13 | 25.05 | 25.21 | 22.35 | 23.45 | 27.47 | 26.53 | 26.21 | 26.24 | 24.88 | 26.74 | 24.82 | 25.44 | 30.1 | 23.08 | 27.89 |
| gemma-3-270m-it | 51.85 | 24.72 | 26.02 | 29.27 | 24.65 | 23.78 | 24.12 | 21.85 | 24.71 | 22.18 | 47.69 | 22.45 | 26.37 | 22.97 | 27.75 | 29.3 | 22.11 | 26.21 | 30.3 | 21.54 | 29.93 |
| Param-1 | 44.44 | 29.78 | 40.12 | 24.39 | 29.77 | 44.76 | 31.53 | 22.69 | 30.59 | 25.14 | 36.27 | 26.53 | 32.61 | 24.34 | 28.71 | 36.51 | 35.45 | 35.26 | 32.32 | 32.92 | 30.61 |
| gemma-2-2b | 77.78 | 36.52 | 34.4 | 26.83 | 21.86 | 41.26 | 32.38 | 26.89 | 31.76 | 26.13 | 44.75 | 26.53 | 39.51 | 27.4 | 27.75 | 27.91 | 40.51 | 35.82 | 32.73 | 32.92 | 25.85 |
| gemma-2-2b-it | 33.33 | 32.02 | 28.33 | 36.59 | 32.56 | 35.66 | 30.4 | 24.37 | 30.59 | 24.29 | 41.98 | 26.53 | 34.6 | 28.98 | 29.67 | 28.84 | 33.38 | 33.55 | 25.86 | 30.77 | 25.85 |
| Llama-3.2-1B | 25.93 | 32.02 | 28.06 | 26.83 | 28.37 | 27.62 | 27.6 | 27.73 | 34.12 | 21.75 | 39.2 | 22.45 | 28.53 | 28.66 | 23.92 | 31.86 | 28.32 | 28.47 | 30.71 | 24.31 | 27.89 |
| Llama-3.2-1B-Instruct | 59.26 | 26.97 | 26.34 | 26.83 | 28.84 | 27.97 | 26.29 | 20.17 | 25.88 | 23.59 | 45.37 | 16.33 | 28.24 | 24.03 | 27.75 | 29.3 | 32.17 | 28.2 | 33.54 | 22.46 | 26.53 |
| Llama-3.2-3B | 25.93 | 29.21 | 30.28 | 36.59 | 27.91 | 36.71 | 31.65 | 31.09 | 32.94 | 25.42 | 25.93 | 24.49 | 32.73 | 26.45 | 27.75 | 26.98 | 35.66 | 33.33 | 26.26 | 35.08 | 23.81 |
| Llama-3.2-3B-Instruct | 40.74 | 34.83 | 33.17 | 29.27 | 35.35 | 38.11 | 31.71 | 32.77 | 31.76 | 29.1 | 43.98 | 24.49 | 39.11 | 28.03 | 35.41 | 28.37 | 37.8 | 37.1 | 32.32 | 38.77 | 27.89 |
| sarvam-2b-v0.5 | 62.96 | 25.84 | 26.81 | 36.59 | 27.91 | 29.02 | 26.1 | 27.73 | 28.24 | 23.16 | 48.61 | 30.61 | 26.83 | 24.55 | 31.58 | 33.95 | 26.75 | 27.47 | 34.34 | 29.85 | 28.57 |
| sarvam-1 | 59.26 | 30.9 | 29.14 | 26.83 | 23.72 | 38.81 | 29.12 | 23.53 | 28.24 | 22.32 | 42.9 | 24.49 | 30.08 | 25.61 | 23.44 | 28.84 | 29.32 | 29.81 | 22.63 | 32.92 | 27.21 |
| Nemotron-4-Mini-Hindi-4B-Base | 55.56 | 32.02 | 34.01 | 36.59 | 29.77 | 43.36 | 34.09 | 26.05 | 31.76 | 26.98 | 47.22 | 34.69 | 37.01 | 26.77 | 24.88 | 37.67 | 43.51 | 40.02 | 27.88 | 36.62 | 23.13 |
| Nemotron-4-Mini-Hindi-4B-Instruct | 37.04 | 30.34 | 33.6 | 24.39 | 27.91 | 38.81 | 31.57 | 26.05 | 29.41 | 25.99 | 46.14 | 36.73 | 35.68 | 30.56 | 31.1 | 30.47 | 35.16 | 36.43 | 32.53 | 35.08 | 30.61 |
| Qwen2.5-3B | 29.63 | 26.97 | 37.5 | 29.27 | 34.88 | 50.7 | 37.5 | 37.82 | 35.29 | 26.41 | 31.94 | 28.57 | 44.08 | 28.13 | 37.32 | 32.33 | 46.36 | 41.8 | 29.7 | 44 | 40.14 |
| Qwen2.5-3B-Instruct | 51.85 | 29.21 | 32.7 | 29.27 | 27.44 | 44.06 | 33.2 | 31.09 | 28.24 | 28.39 | 39.2 | 28.57 | 40.61 | 30.87 | 40.19 | 35.35 | 38.45 | 37.63 | 26.26 | 39.38 | 31.29 |
| granite-3.1-2b-instruct | 33.33 | 21.35 | 31.22 | 29.27 | 33.95 | 33.92 | 31.31 | 30.25 | 31.76 | 22.88 | 48.92 | 24.49 | 35.92 | 28.66 | 33.49 | 35.12 | 37.09 | 34.97 | 27.88 | 36.31 | 25.85 |
| granite-3.1-3b-a800m-base | 62.96 | 25.28 | 29.65 | 29.27 | 26.98 | 33.57 | 27.78 | 27.78 | 29.41 | 22.03 | 44.44 | 28.57 | 32.24 | 24.55 | 24.88 | 34.65 | 31.53 | 30.89 | 26.06 | 28.31 | 24.49 |
| *7B to 27B Models* | | | | | | | | | | | | | | | | | | | | | |
| Pangea-7B | 62.96 | 24.16 | 37.53 | 34.15 | 34.88 | 52.8 | 39.44 | 35.29 | 31.76 | 31.92 | 50.46 | 32.65 | 45.69 | 32.35 | 38.76 | 39.3 | 47.65 | 44.78 | 32.93 | 46.77 | 34.69 |
| Indic-gemma-7b-finetuned-sft-Navarasa-2.0 | 59.26 | 35.39 | 35.1 | 31.71 | 27.91 | 43.36 | 35.35 | 38.66 | 25.88 | 25.14 | 47.69 | 30.61 | 41.63 | 26.24 | 28.23 | 40.93 | 42.51 | 41.26 | 32.73 | 41.23 | 29.25 |
| aya-23-8B | 18.52 | 30.9 | 32.05 | 17.07 | 33.95 | 41.96 | 34.13 | 33.61 | 31.76 | 25.28 | 27.16 | 30.61 | 37.99 | 22.76 | 24.88 | 31.4 | 43.01 | 39.55 | 24.65 | 40.31 | 28.57 |
| Llama-3.1-8B | 29.63 | 29.78 | 33.17 | 34.15 | 31.16 | 34.74 | 28.57 | 31.76 | 30.59 | 24.86 | 29.78 | 34.69 | 35.98 | 27.26 | 25.17 | 28.6 | 42.08 | 38.74 | 25.86 | 41.54 | 25.17 |
| Llama-3.1-8B-Instruct | 29.63 | 26.97 | 34.83 | 46.34 | 38.6 | 44.41 | 34.18 | 31.61 | 30.59 | 24.72 | 39.51 | 28.57 | 46.07 | 35.3 | 38.76 | 34.19 | 46.43 | 45.41 | 32.93 | 44.92 | 36.73 |
| gemma-2-9b | 33.33 | 35.39 | 44.48 | 31.71 | 35.35 | 61.89 | 41.26 | 32.77 | 31.76 | 28.39 | 38.89 | 40.82 | 55.95 | 28.45 | 34.93 | 34.88 | 58.42 | 54.34 | 41.01 | 53.54 | 33.33 |
| gemma-2-9b-it | 48.15 | 29.21 | 34.35 | 39.02 | 36.74 | 52.1 | 36.88 | 37.82 | 29.41 | 27.97 | 44.44 | 24.49 | 47.12 | 43.1 | 47.37 | 42.56 | 44.15 | 43.33 | 35.76 | 40.62 | 36.05 |
| gpt-oss-20b | 25.93 | 32.02 | 36.39 | 46.34 | 30.7 | 47.9 | 36 | 27.73 | 31.76 | 27.26 | 29.32 | 26.53 | 46.74 | 29.61 | 35.41 | 24.65 | 45.58 | 39.14 | 34.95 | 31.08 | 37.41 |
| gemma-2-27b | 29.63 | 39.89 | 47.71 | 26.83 | 42.33 | 61.89 | 46.36 | 36.13 | 36.47 | 27.97 | 37.04 | 40.82 | 61 | 35.19 | 46.89 | 43.49 | 65.34 | 61.47 | 49.9 | 58.77 | 42.18 |
| gemma-2-27b-it | 55.56 | 35.96 | 37.98 | 39.02 | 39.53 | 55.24 | 40.15 | 36.97 | 31.76 | 30.51 | 45.99 | 38.78 | 53.28 | 45.94 | 55.02 | 39.77 | 50 | 48.4 | 40 | 45.23 | 44.9 |
| *> 27B Models* | | | | | | | | | | | | | | | | | | | | | |
| gpt-oss-120b | 62.96 | 46.07 | 52.87 | 41.46 | 66.05 | 100 | 76.22 | 71.3 | 68.07 | 67.06 | 62.81 | 40.82 | 70.14 | 64.17 | 72.73 | 62.09 | 71.61 | 68.42 | 55.96 | 78.77 | 69.39 |
| Qwen3-235B-A22B-Instruct-2507 | 62.96 | 51.69 | 58.34 | 31.71 | 67.91 | 77.27 | 61.65 | 69.75 | 51.76 | 47.18 | 70.99 | 59.18 | 73.14 | 67.76 | 75.12 | 73.49 | 75.82 | 77.17 | 61.62 | 77.54 | 71.43 |
| deepseek-v3 | 66.67 | 38.2 | 46.09 | 31.71 | 63.26 | 81.82 | 61.7 | 65.55 | 41.18 | 49.01 | 61.11 | 46.94 | 60.71 | 44.89 | 62.2 | 55.58 | 61.98 | 61.92 | 45.45 | 66.15 | 51.7 |
| gpt-4o | 62.96 | 47.19 | 59.95 | 36.59 | 63.72 | 100 | 75.87 | 54.82 | 63.87 | 50.59 | 70.22 | 57.14 | 74.06 | 68.6 | 73.21 | 69.07 | 74.96 | 77.19 | 62.22 | 74.46 | 61.9 |

Table 19: Performance of Llama model family across sub-domains in BhashaBench v1, comparing base and instruction-tuned variants (1B, 3B, 8B)

| Subject Domain | 3.2-1B | 3.2-1B-it | 3.2-3B | 3.2-3B-it | 3.1-8B | 3.1-8B-it |
|---|---|---|---|---|---|---|
| **BBA** | | | | | | |
| Administration, AYUSH & Miscellaneous | 36.97 | 35.29 | 31.93 | 39.5 | 41.18 | 44.54 |
| Agad Tantra & Forensic Medicine | 28.28 | 27.09 | 35.09 | 39.01 | 33.9 | 35.6 |
| Ayurvedic Literature & History | 27.45 | 30.88 | 29.9 | 33.33 | 30.88 | 36.27 |
| Dravyaguna & Bhaishajya | 26.58 | 26.92 | 26.95 | 30.11 | 29.24 | 31.53 |
| Kaumarbhritya & Pediatrics | 28.57 | 25.63 | 29.41 | 32.91 | 31.09 | 35.71 |
| Kayachikitsa (General Medicine & Internal Medicine in Ayurveda) | 29.04 | 24.92 | 31.33 | 34.84 | 34.24 | 34.78 |
| Panchakarma & Rasayana | 27.06 | 25.76 | 27.06 | 30.2 | 29.05 | 28.75 |
| Research & Statistics | 27.14 | 29.5 | 40 | 44.29 | 47.62 | 54.76 |
| Roga Vigyana (Diagnostics & Pathology) | 35 | 25 | 45 | 42.5 | 50 | 61.25 |
| Samhita & Siddhanta (Fundamentals) | 29.92 | 26.15 | 31.28 | 27.84 | 33.55 | 27.9 |
| Shalakya Tantra (ENT, Eye, Dentistry) | 27.25 | 26.84 | 29.43 | 35.29 | 31.61 | 37.47 |
| Shalya Tantra (Surgery) | 25.48 | 25.48 | 28.33 | 30.8 | 35.17 | 34.6 |
| Sharir (Anatomy & Physiology) | 27.12 | 25.19 | 29.49 | 33.66 | 32.76 | 38.93 |
| Stri Roga & Prasuti Tantra (Gynecology & Obstetrics) | 27.27 | 28.1 | 31.88 | 33.6 | 34 | 36.36 |
| Swasthavritta & Public Health | 34 | 32.67 | 40.62 | 51.21 | 47.46 | 57.17 |
| Yoga & Psychology | 26.6 | 24.47 | 32.45 | 31.38 | 43.62 | 34.57 |
| **BBF** | | | | | | |
| Accounting | 27.3 | 26.13 | 30.66 | 27.68 | 34.54 | 30.66 |
| Banking Services | 30.49 | 28.18 | 38.34 | 38.68 | 40.48 | 42.36 |
| Behavioral Finance | 37.31 | 28.36 | 35.82 | 37.31 | 47.76 | 49.25 |
| Business Management | 26.51 | 26.51 | 43.37 | 53.01 | 50.6 | 60.24 |
| Commerce | 28.51 | 27.46 | 32.1 | 31.52 | 34.41 | 31.98 |
| Corporate Finance & Investment | 27.58 | 26.37 | 29.56 | 35.05 | 37.91 | 39.23 |
| Data & Analytics in Finance | 22.83 | 18.11 | 31.5 | 20.47 | 32.28 | 31.5 |
| Economics & Development Studies | 29.56 | 32.85 | 36.13 | 40.51 | 39.42 | 48.18 |
| Energy, Infrastructure & Finance | 29.27 | 28.05 | 32.93 | 39.02 | 42.68 | 40.24 |

*Continued on next page*

Table 19 – *Continued from previous page*

| Subject Domain | 3.2-1B | 3.2-1B-it | 3.2-3B | 3.2-3B-it | 3.1-8B | 3.1-8B-it |
|---|---|---|---|---|---|---|
| Environmental Finance | 25 | 29.76 | 39.29 | 38.69 | 41.07 | 51.19 |
| Finance Education | 29.66 | 25.42 | 49.15 | 34.75 | 44.92 | 47.46 |
| Financial Markets | 36.17 | 29.79 | 57.45 | 48.94 | 40.43 | 51.06 |
| Financial Technology | 17.39 | 13.04 | 21.74 | 34.78 | 43.48 | 47.83 |
| General Knowledge | 31.35 | 28.94 | 37.48 | 43.04 | 42.3 | 50.09 |
| Governance & Policy | 28.76 | 27.63 | 34.3 | 39.29 | 40.13 | 47.84 |
| Healthcare Economics | 31.58 | 31.58 | 38.6 | 41.23 | 50.88 | 51.75 |
| History, Sociology & Cultural Studies of Finance | 24.41 | 30.71 | 37.01 | 44.88 | 41.73 | 61.42 |
| Information Technology Finance | 31.63 | 35.51 | 46.33 | 53.06 | 59.59 | 66.33 |
| Insurance & Risk Management | 19.05 | 26.19 | 30.95 | 38.1 | 42.86 | 40.48 |
| Interdisciplinary Finance | 26.14 | 30.72 | 37.25 | 33.33 | 37.91 | 54.9 |
| International Finance & Trade | 27.71 | 34.94 | 36.14 | 39.76 | 45.78 | 54.22 |
| Language & Communication | 32.45 | 29.18 | 35.62 | 40.59 | 42.49 | 43.66 |
| Legal Finance | 26.47 | 20.59 | 29.41 | 20.59 | 38.24 | 35.29 |
| Marketing Finance | 23.81 | 38.1 | 38.1 | 38.1 | 59.52 | 52.38 |
| Mathematics for Finance | 27.31 | 24.91 | 28.96 | 27.57 | 29.97 | 26.3 |
| Problem Solving | 24.67 | 23.65 | 27.08 | 25.15 | 28.1 | 24.6 |
| Rural Economics | 27.97 | 30.65 | 33.33 | 44.83 | 42.53 | 51.72 |
| Science and Technology in Finance | 21.78 | 30.69 | 31.68 | 41.58 | 38.61 | 35.64 |
| Sports, Media & Finance Linkages | 33.33 | 28.89 | 48.89 | 42.22 | 51.11 | 48.89 |
| Taxation & Regulatory Compliance | 36.13 | 31.61 | 43.87 | 47.1 | 47.1 | 50.97 |
| **BBK** | | | | | | |
| Agri-Environmental & Allied Disciplines | 31.82 | 32.95 | 25 | 36.36 | 30.68 | 47.73 |
| Agricultural Biotechnology | 31.11 | 28.63 | 34.35 | 50.95 | 48.85 | 58.78 |
| Agricultural Chemistry & Biochemistry | 27.05 | 22.78 | 31.32 | 33.81 | 38.79 | 48.75 |
| Agricultural Economics & Policy | 29.98 | 25.52 | 35.09 | 38.12 | 40.35 | 46.73 |
| Agricultural Engineering & Technology | 27.46 | 26.23 | 32.79 | 33.2 | 38.93 | 41.8 |
| Agricultural Extension Education | 30.88 | 29.46 | 32.3 | 41.99 | 40.31 | 48.19 |
| Agricultural Microbiology | 34.23 | 36.04 | 31.53 | 53.15 | 38.74 | 54.95 |
| Agriculture Communication | 33.07 | 28.35 | 29.53 | 44.49 | 36.61 | 49.21 |
| Agriculture Information Technology | 30.53 | 31.58 | 44.21 | 45.79 | 46.32 | 45.79 |
| Agronomy | 27.92 | 28.77 | 31.84 | 37.22 | 37.2 | 43.34 |
| Animal Sciences | 25.68 | 34.46 | 36.49 | 41.89 | 46.62 | 45.95 |
| Crop Sciences | 31.15 | 26.41 | 29.87 | 35.34 | 38.25 | 40.8 |
| Dairy & Poultry Science | 35.96 | 31.46 | 30.34 | 37.08 | 41.57 | 44.94 |
| Entomology | 29.02 | 27.59 | 35.49 | 35.49 | 38.79 | 47.7 |
| Fisheries and Aquaculture | 29.41 | 41.18 | 38.24 | 55.88 | 38.24 | 52.94 |
| General Knowledge & Reasoning | 28.44 | 27.53 | 33.13 | 39.64 | 38.88 | 42.66 |
| Genetics and Plant Breeding | 30.59 | 30.08 | 28.02 | 38.3 | 40.62 | 43.19 |
| Horticulture | 27.05 | 28.6 | 31.21 | 36.86 | 35.89 | 43 |
| Natural Resource Management | 28.5 | 26.42 | 29.02 | 37.82 | 33.16 | 44.56 |
| Nematology | 22.83 | 28.26 | 28.26 | 29.35 | 35.33 | 41.3 |
| Plant Pathology | 28.97 | 30.48 | 27.96 | 42.82 | 34.01 | 44.84 |
| Plant Sciences & Physiology | 28.68 | 31.78 | 37.98 | 50.39 | 43.41 | 54.26 |
| Seed Science and Technology | 29.7 | 28.71 | 27.72 | 37.13 | 35.15 | 38.61 |
| Soil Science | 31.25 | 29.92 | 31.69 | 38.84 | 37.14 | 45.25 |
| Veterinary Sciences | 27.08 | 14.58 | 37.5 | 47.92 | 43.75 | 70.83 |
| **BBL** | | | | | | |
| Civil Litigation & Procedure | 29.32 | 28.18 | 32.4 | 34.97 | 36.68 | 42.66 |
| Constitutional & Administrative Law | 29.54 | 28.15 | 36.22 | 40.62 | 42.28 | 49.46 |
| Consumer & Competition Law | 28 | 22.67 | 28 | 34.67 | 46.67 | 41.33 |
| Corporate & Commercial Law | 27.7 | 28.63 | 29.78 | 34.67 | 35.15 | 42.67 |
| Criminal Law & Justice | 27.09 | 26.98 | 30.01 | 33.66 | 35.21 | 42.72 |
| Employment & Labour Law | 23.43 | 25.71 | 28.57 | 29.1 | 32 | 40 |
| Environmental & Energy Law | 27.67 | 24.42 | 33.49 | 37.91 | 39.07 | 45.81 |

*Continued on next page*

Table 19 – *Continued from previous page*

| Subject Domain | 3.2-1B | 3.2-1B-it | 3.2-3B | 3.2-3B-it | 3.1-8B | 3.1-8B-it |
|---|---|---|---|---|---|---|
| Family & Personal Law | 24.12 | 28.86 | 29.06 | 31.69 | 34.21 | 39.86 |
| General Academic Subjects | 29.21 | 32.52 | 37.47 | 43.91 | 46.87 | 52.68 |
| Healthcare & Medical Law | 40 | 20 | 68 | 40 | 64 | 60 |
| Human Rights & Social Justice | 21.05 | 42.11 | 36.84 | 26.32 | 31.58 | 36.84 |
| Intellectual Property Law | 30.77 | 31.87 | 46.15 | 45.05 | 56.04 | 58.24 |
| Interdisciplinary Studies | 33.33 | 28.1 | 38.57 | 41.32 | 43.25 | 53.72 |
| International & Comparative Law | 30.87 | 30.35 | 40.02 | 45.22 | 46.88 | 54.47 |
| Legal Skills & Communication | 25.74 | 27.33 | 28.68 | 30.15 | 28.31 | 32.72 |
| Legal Theory & Jurisprudence | 29.63 | 28.36 | 33.92 | 39.69 | 41.66 | 46.87 |
| Media & Entertainment Law | 33.33 | 35.19 | 42.59 | 51.85 | 38.89 | 53.7 |
| Real Estate & Property Law | 23.53 | 25.91 | 29.89 | 31.96 | 31.48 | 38.16 |
| Tax & Revenue Law | 27.71 | 31.6 | 40.26 | 38.1 | 41.56 | 43.29 |
| Technology & Cyber Law | 30.89 | 41.46 | 48.78 | 49.59 | 51.22 | 60.16 |

Table 20: Performance of Qwen model family across sub-domains in BhashaBench v1, comparing base and instruction-tuned variants (3B, 235B)

| Subject Domain | 2.5-3B | 2.5-3B-it | 3-235B-A22B-it-2507 |
|---|---|---|---|
| **BBA** | | | |
| Administration, AYUSH & Miscellaneous | 47.06 | 38.66 | 73.11 |
| Agad Tantra & Forensic Medicine | 39.86 | 32.71 | 63.88 |
| Ayurvedic Literature & History | 38.73 | 29.9 | 55.88 |
| Dravyaguna & Bhaishajya | 32.57 | 28.94 | 49.43 |
| Kaumarbhritya & Pediatrics | 38.52 | 30.11 | 55.32 |
| Kayachikitsa (General Medicine & Internal Medicine in Ayurveda) | 38.61 | 35.07 | 59.48 |
| Panchakarma & Rasayana | 30.35 | 29.59 | 49.54 |
| Research & Statistics | 62.86 | 52.86 | 91.43 |
| Roga Vigyana (Diagnostics & Pathology) | 58.75 | 53.75 | 82.5 |
| Samhita & Siddhanta (Fundamentals) | 36.79 | 31.93 | 55.22 |
| Shalakya Tantra (ENT, Eye, Dentistry) | 35.56 | 31.74 | 59.67 |
| Shalya Tantra (Surgery) | 37.45 | 33.08 | 60.46 |
| Sharir (Anatomy & Physiology) | 37.44 | 31.35 | 60.1 |
| Stri Roga & Prasuti Tantra (Gynecology & Obstetrics) | 40.73 | 34.24 | 66.82 |
| Swasthavritta & Public Health | 50.99 | 43.49 | 82.56 |
| Yoga & Psychology | 44.68 | 36.17 | 75.53 |
| **BBF** | | | |
| Accounting | 38.94 | 31.82 | 63.52 |
| Banking Services | 43.3 | 36.89 | 71.22 |
| Behavioral Finance | 52.24 | 44.78 | 71.64 |
| Business Management | 60.24 | 40.96 | 84.34 |
| Commerce | 43.57 | 33.72 | 63.62 |
| Corporate Finance & Investment | 40.22 | 37.58 | 63.52 |
| Data & Analytics in Finance | 35.43 | 28.35 | 53.54 |
| Economics & Development Studies | 43.8 | 44.16 | 73.36 |
| Energy, Infrastructure & Finance | 45.12 | 30.49 | 71.95 |
| Environmental Finance | 47.62 | 44.05 | 82.74 |
| Finance Education | 50.85 | 43.22 | 69.49 |
| Financial Markets | 42.55 | 42.55 | 70.21 |
| Financial Technology | 47.83 | 39.13 | 78.26 |
| General Knowledge | 41.56 | 38.22 | 74.95 |
| Governance & Policy | 45.3 | 38.16 | 74.15 |
| Healthcare Economics | 48.25 | 45.61 | 78.95 |
| History, Sociology & Cultural Studies of Finance | 38.58 | 38.58 | 83.46 |
| Information Technology Finance | 64.9 | 58.16 | 92.24 |
| Insurance & Risk Management | 30.95 | 38.1 | 64.29 |
| Interdisciplinary Finance | 41.83 | 36.6 | 79.74 |

Table 20 – *Continued from previous page*

| Subject Domain | 2.5-3B | 2.5-3B-it | 3-235B-A22B-it-2507 |
|---|---|---|---|
| International Finance & Trade | 49.4 | 42.17 | 78.31 |
| Language & Communication | 45.77 | 42.71 | 77.06 |
| Legal Finance | 38.24 | 23.53 | 76.47 |
| Marketing Finance | 69.05 | 50 | 85.71 |
| Mathematics for Finance | 34.18 | 29.85 | 58.04 |
| Problem Solving | 27.88 | 26.2 | 47.12 |
| Rural Economics | 47.13 | 45.21 | 80.46 |
| Science and Technology in Finance | 40.59 | 43.56 | 72.28 |
| Sports, Media & Finance Linkages | 44.44 | 53.33 | 68.89 |
| Taxation & Regulatory Compliance | 56.13 | 38.71 | 74.84 |
| **BBK** | | | |
| Agri-Environmental & Allied Disciplines | 43.75 | 43.18 | 75.57 |
| Agricultural Biotechnology | 55.34 | 51.15 | 91.6 |
| Agricultural Chemistry & Biochemistry | 44.48 | 38.43 | 83.63 |
| Agricultural Economics & Policy | 46.41 | 43.38 | 73.21 |
| Agricultural Engineering & Technology | 41.39 | 37.3 | 67.21 |
| Agricultural Extension Education | 46.25 | 42.51 | 72.87 |
| Agricultural Microbiology | 54.05 | 43.24 | 90.99 |
| Agriculture Communication | 44.49 | 44.49 | 78.35 |
| Agriculture Information Technology | 52.63 | 54.21 | 74.74 |
| Agronomy | 41.73 | 38.89 | 71.92 |
| Animal Sciences | 47.97 | 46.62 | 77.7 |
| Crop Sciences | 42.08 | 36.79 | 67.4 |
| Dairy & Poultry Science | 52.81 | 46.07 | 75.28 |
| Entomology | 39.94 | 39.66 | 77.44 |
| Fisheries and Aquaculture | 38.24 | 50 | 79.41 |
| General Knowledge & Reasoning | 44.48 | 41.6 | 73.22 |
| Genetics and Plant Breeding | 43.44 | 44.22 | 76.86 |
| Horticulture | 37.25 | 35.41 | 64.98 |
| Natural Resource Management | 37.82 | 37.31 | 65.8 |
| Nematology | 33.15 | 39.13 | 63.04 |
| Plant Pathology | 40.55 | 36.52 | 78.34 |
| Plant Sciences & Physiology | 45.74 | 48.06 | 86.82 |
| Seed Science and Technology | 42.08 | 34.65 | 66.34 |
| Soil Science | 42 | 39.35 | 72.37 |
| Veterinary Sciences | 45.83 | 50 | 87.5 |
| **BBL** | | | |
| Civil Litigation & Procedure | 38.65 | 35.31 | 72.12 |
| Constitutional & Administrative Law | 43.67 | 37.93 | 82.65 |
| Consumer & Competition Law | 36 | 46.67 | 82.67 |
| Corporate & Commercial Law | 40.74 | 37.7 | 77.11 |
| Criminal Law & Justice | 38.21 | 34.45 | 75.44 |
| Employment & Labour Law | 39.43 | 37.14 | 71.43 |
| Environmental & Energy Law | 44.65 | 38.84 | 76.74 |
| Family & Personal Law | 38.35 | 32.8 | 74.37 |
| General Academic Subjects | 53.82 | 45.44 | 85.82 |
| Healthcare & Medical Law | 56 | 40 | 88 |
| Human Rights & Social Justice | 47.37 | 31.58 | 73.68 |
| Intellectual Property Law | 60.44 | 54.95 | 87.91 |
| Interdisciplinary Studies | 49.31 | 44.08 | 84.85 |
| International & Comparative Law | 47.51 | 43.76 | 83.89 |
| Legal Skills & Communication | 32.35 | 31.74 | 61.27 |
| Legal Theory & Jurisprudence | 46.45 | 40.04 | 79.38 |
| Media & Entertainment Law | 42.59 | 33.33 | 79.63 |
| Real Estate & Property Law | 36.09 | 33.55 | 71.7 |
| Tax & Revenue Law | 39.83 | 37.66 | 74.03 |
| Technology & Cyber Law | 58.54 | 59.35 | 86.18 |

Table 21: Performance of GPT model family across sub-domains in BhashaBench v1, comparing different model sizes (20B, 120B, GPT-4o)

| Subject Domain | gpt-oss-20b | gpt-oss-120b | gpt-4o |
|---|---|---|---|
| **BBA** | | | |
| Administration, AYUSH & Miscellaneous | 53.78 | 79.83 | 75.63 |
| Agad Tantra & Forensic Medicine | 39.52 | 60.14 | 63.54 |
| Ayurvedic Literature & History | 33.82 | 51.47 | 59.31 |
| Dravyaguna & Bhaishajya | 30.75 | 44.48 | 54.78 |
| Kaumarbhritya & Pediatrics | 35.99 | 51.4 | 56.58 |
| Kayachikitsa (General Medicine & Internal Medicine in Ayurveda) | 39.06 | 54.69 | 60.69 |
| Panchakarma & Rasayana | 28.36 | 41.44 | 50.76 |
| Research & Statistics | 70.95 | 86.67 | 90 |
| Roga Vigyana (Diagnostics & Pathology) | 66.25 | 82.5 | 81.25 |
| Samhita & Siddhanta (Fundamentals) | 30.63 | 46.07 | 53.41 |
| Shalakya Tantra (ENT, Eye, Dentistry) | 38.15 | 54.9 | 62.4 |
| Shalya Tantra (Surgery) | 35.36 | 55.13 | 61.41 |
| Sharir (Anatomy & Physiology) | 39.75 | 57.06 | 62.7 |
| Stri Roga & Prasuti Tantra (Gynecology & Obstetrics) | 35.18 | 59.03 | 64.82 |
| Swasthavritta & Public Health | 56.51 | 76.6 | 81.02 |
| Yoga & Psychology | 41.49 | 70.74 | 73.94 |
| **BBF** | | | |
| Accounting | 35.45 | 73.61 | 49.55 |
| Banking Services | 42.53 | 67.29 | 68.57 |
| Behavioral Finance | 50.75 | 77.61 | 76.12 |
| Business Management | 53.01 | 87.95 | 81.93 |
| Commerce | 37.89 | 69.76 | 54.46 |
| Corporate Finance & Investment | 37.25 | 73.63 | 61.43 |
| Data & Analytics in Finance | 34.65 | 51.97 | 44.09 |
| Economics & Development Studies | 46.72 | 69.34 | 71.53 |
| Energy, Infrastructure & Finance | 39.02 | 64.63 | 67.07 |
| Environmental Finance | 55.95 | 73.21 | 77.98 |
| Finance Education | 46.61 | 73.73 | 74.58 |
| Financial Markets | 61.7 | 59.57 | 72.34 |
| Financial Technology | 47.83 | 73.91 | 78.26 |
| General Knowledge | 48.42 | 77.18 | 77.18 |
| Governance & Policy | 39.85 | 69.36 | 78.29 |
| Healthcare Economics | 49.12 | 78.07 | 80.7 |
| History, Sociology & Cultural Studies of Finance | 48.03 | 68.5 | 87.4 |
| Information Technology Finance | 76.94 | 90.82 | 92.04 |
| Insurance & Risk Management | 47.62 | 57.14 | 64.29 |
| Interdisciplinary Finance | 45.1 | 73.2 | 75.82 |
| International Finance & Trade | 54.22 | 75.9 | 85.54 |
| Language & Communication | 47.57 | 74.42 | 77.48 |
| Legal Finance | 41.18 | 64.71 | 76.47 |
| Marketing Finance | 61.9 | 85.71 | 78.57 |
| Mathematics for Finance | 30.05 | 76.16 | 41.28 |
| Problem Solving | 26.63 | 64.14 | 42.65 |
| Rural Economics | 47.89 | 75.86 | 82.76 |
| Science and Technology in Finance | 45.54 | 77.23 | 73.27 |
| Sports, Media & Finance Linkages | 46.67 | 75.56 | 73.33 |
| Taxation & Regulatory Compliance | 44.52 | 68.39 | 73.55 |
| **BBK** | | | |
| Agri-Environmental & Allied Disciplines | 41.48 | 73.86 | 74.43 |
| Agricultural Biotechnology | 65.27 | 89.69 | 89.31 |
| Agricultural Chemistry & Biochemistry | 54.8 | 80.43 | 81.14 |
| Agricultural Economics & Policy | 46.57 | 71.77 | 73.68 |
| Agricultural Engineering & Technology | 39.75 | 62.7 | 66.8 |
| Agricultural Extension Education | 43.93 | 69.25 | 75.19 |

*Continued on next page*

Table 21 – *Continued from previous page*

| Subject Domain | gpt-oss-20b | gpt-oss-120b | gpt-4o |
|---|---|---|---|
| Agricultural Microbiology | 53.15 | 89.19 | 94.59 |
| Agriculture Communication | 42.91 | 73.23 | 81.1 |
| Agriculture Information Technology | 51.58 | 75.26 | 68.42 |
| Agronomy | 44.1 | 68 | 72.43 |
| Animal Sciences | 53.38 | 69.59 | 76.35 |
| Crop Sciences | 41.71 | 64.66 | 68.85 |
| Dairy & Poultry Science | 52.81 | 75.28 | 78.65 |
| Entomology | 48.28 | 72.84 | 77.87 |
| Fisheries and Aquaculture | 50 | 64.71 | 73.53 |
| General Knowledge & Reasoning | 42.81 | 69.59 | 68.38 |
| Genetics and Plant Breeding | 44.47 | 74.04 | 75.84 |
| Horticulture | 41.26 | 61.88 | 70.14 |
| Natural Resource Management | 41.97 | 64.77 | 65.8 |
| Nematology | 42.93 | 64.13 | 64.67 |
| Plant Pathology | 41.56 | 71.03 | 78.34 |
| Plant Sciences & Physiology | 51.94 | 82.17 | 88.37 |
| Seed Science and Technology | 35.15 | 64.85 | 65.84 |
| Soil Science | 42.45 | 70.67 | 73.18 |
| Veterinary Sciences | 56.25 | 87.5 | 93.75 |
| **BBL** | | | |
| Civil Litigation & Procedure | 34.63 | 59.01 | 71.91 |
| Constitutional & Administrative Law | 41.06 | 75.56 | 83.15 |
| Consumer & Competition Law | 33.33 | 72 | 81.33 |
| Corporate & Commercial Law | 37.48 | 69.59 | 78.93 |
| Criminal Law & Justice | 35.14 | 65.11 | 75.95 |
| Employment & Labour Law | 33.14 | 62.86 | 73.14 |
| Environmental & Energy Law | 41.4 | 69.3 | 73.26 |
| Family & Personal Law | 37.03 | 63.87 | 72.86 |
| General Academic Subjects | 56.49 | 83.14 | 84.79 |
| Healthcare & Medical Law | 60 | 92 | 92 |
| Human Rights & Social Justice | 15.79 | 73.68 | 68.42 |
| Intellectual Property Law | 53.85 | 85.71 | 90.11 |
| Interdisciplinary Studies | 43.25 | 82.64 | 83.75 |
| International & Comparative Law | 48.86 | 79.42 | 81.7 |
| Legal Skills & Communication | 32.84 | 69.12 | 53.43 |
| Legal Theory & Jurisprudence | 42.08 | 75.16 | 81.21 |
| Media & Entertainment Law | 50 | 83.33 | 85.19 |
| Real Estate & Property Law | 32.59 | 59.62 | 71.7 |
| Tax & Revenue Law | 42.86 | 67.53 | 69.26 |
| Technology & Cyber Law | 56.91 | 86.18 | 86.99 |

### E.3 QUALITATIVE ERROR ANALYSIS FOR LLAMA-3.1-8B

In this section, we present a qualitative error analysis of the Llama-3.1-8B model across four domains of BhashaBench V1: Ayurveda (BBA), Finance (BBF), Agriculture (BBK), and Legal (BBL). For each domain, we examine some examples where the model's responses deviated from the correct answers, highlighting the nature of errors, underlying causes, and potential strategies for mitigation. This analysis provides insight into the model's domain-specific weaknesses, including challenges in understanding classical Ayurvedic terminology, procedural and regulatory reasoning in finance and law, and domain-specific factual knowledge in agriculture. Through these examples, we aim to identify patterns of systematic errors, informing future improvements in multilingual and domain-aware LLM performance.

#### E.3.1 BBA QUALITATIVE ANALYSIS

The qualitative analysis in the Ayurvedic domain reveals that Llama-3.1-8B exhibits several recurring challenges across classical medical texts, ritualistic procedures, and disease subtypes. The model frequently struggles with domain-specific terminology, semantic overlaps between closely related therapeutic indications, and nuanced procedural instructions from traditional texts. Errors are

often observed when differentiating between similar disease subtypes or interpreting complex instructions from Panchakarma, Rasayana, and Stri Roga protocols. Moreover, the model sometimes mislabels classical terms or associates remedies with incorrect disease indications, demonstrating gaps in both lexical knowledge and reasoning.

For instance, confusion arises between general wasting conditions and respiratory disorders, or between correct handling steps in Mritashodhana rituals. In other cases, the model misidentifies disease subtypes in Kayachikitsa or misinterprets repeated obstetric terms in Prasuti Tantra. These observations indicate that while Llama-3.1-8B has captured general patterns in Ayurvedic knowledge, it requires explicit integration of classical formulations, accurate disease nomenclature, and procedural rules to improve precision and reliability in reasoning.

The error patterns suggest that enhancing the model with structured knowledge of classical Ayurvedic texts, domain-specific terminologies, and procedural protocols can significantly reduce semantic confusions and improve zero-shot performance in complex Ayurvedic tasks. Overall, the analysis highlights the critical role of targeted domain knowledge in enabling large language models to reason effectively within traditional medicine contexts.

---

**Question:**
Dashamooladi Ghruta is used in. . . . . . vyadhi

**Options:**

    A. Kasa vyadhi

    B. Shosh

    C. Pandu

    D. Shwasa

**Correct Answer:** B
**Model Selected Answer:** A
**Subject Domain:** Dravyaguna & Bhaishajya

**Error Analysis:**

- **Nature of Error:** Confusion between respiratory disorders and general wasting conditions.

- **Underlying Cause:** The model misinterpreted the therapeutic indications of Dashamooladi Ghruta, associating it with cough-related ailments instead of Shosh (emaciation or wasting disease).

- **Recommendation:** Strengthen model knowledge of Ayurvedic formulations along with their specific disease indications to avoid semantic overlaps.

**Question:**
Which of the following options is most appropriate regarding removal of the body from water for Mritashodhana?

**Options:**

    A. Samyak prakuthit

    B. Seven days

    C. Both A & B

    D. None of A & B

**Correct Answer:** A
**Model Selected Answer:** D
**Subject Domain:** Panchakarma & Rasayana

**Error Analysis:**

- **Nature of Error:** Misunderstanding of procedural rules in classical texts.
- **Underlying Cause:** The model failed to correctly associate the Mritashodhana process with the proper handling of a deceased body, leading to selection of an incorrect "none" option.
- **Recommendation:** Incorporate explicit procedural knowledge from classical Ayurvedic texts to enhance reasoning on ritualistic and therapeutic protocols.

**Question:**
वात पित्त प्रधान विसर्प को - - - - - - कहा जाता है

**Options:**

    A. अग्नि

    B. कर्दम

    C. ग्रंथि

    D. निचय

**Correct Answer:** A
**Model Selected Answer:** C
**Subject Domain:** Kayachikitsa (General Medicine & Internal Medicine in Ayurveda)
**Error Analysis:**

- **Nature of Error:** Mislabeling disease subtype in classical terminology.
- **Underlying Cause:** The model failed to correctly identify the term for Vata-Pitta dominant Visarpa, selecting "Granthii" instead of "Agni".
- **Recommendation:** Include domain-specific classical terminology in training for accurate disease nomenclature recognition.

**Question:**
पहली तिमाही में बार-बार होने वाले गर्भपात का अर्थ है - - - - - - वंध्या।

**Options:**

    A. मृत्वत्सा

    B. काकवंध्या

    C. गर्भस्त्रावी

    D. बाला

**Correct Answer:** A
**Model Selected Answer:** D
**Subject Domain:** Stri Roga & Prasuti Tantra (Gynecology & Obstetrics)
**Error Analysis:**

- **Nature of Error:** Misinterpretation of obstetric terminology.

- **Underlying Cause:** The model selected "Bala" instead of "Mritvatsa" due to misunderstanding repeated first-trimester miscarriage terminology.

- **Recommendation:** Incorporate classical and modern obstetric definitions for precise identification of pathological conditions.

### E.3.2 BBF QUALITATIVE ANALYSIS

In the finance domain, errors are primarily due to confusion between regulatory and advisory institutions, misidentification of financial instruments, and misapplication of problem-solving reasoning. The model occasionally fails to account for context-specific definitions or institutional roles. Incorporating structured financial knowledge, including Indian regulatory frameworks and definitions of financial instruments, would enhance model performance.

**Question:**
Which of these has set up a high-level panel to suggest possible structures and regulations for creating social stock exchanges to facilitate listing and fund-raising by social enterprises as well as voluntary organisations?

**Options:**

    A. RBI

    B. NITI Aayog

    C. Finance Ministry

    D. SEBI

**Correct Answer:** D
**Model Selected Answer:** B
**Subject Domain:** Taxation & Regulatory Compliance
**Error Analysis:**

- **Nature of Error:** Confusion between regulatory and advisory bodies.

- **Underlying Cause:** The model associated policy advice with NITI Aayog rather than SEBI's regulatory role in social stock exchanges.

- **Recommendation:** Enhance knowledge of institutional functions in Indian financial and regulatory frameworks.

**Question:**
Directions: Read the following comprehension carefully and answer the questions given below.
A certain number of people are sitting in a linear row facing North.
- R sits fifth from the left end of the row.
- Only two persons sit between R and G, who sits to the right of R.
- X sits second to the left of G.
- Only one person sits to the left of L.
- Three persons sit between X and A, who sits second from the right end of the row.
- L sits third to the left of R.
- V sits sixth to the right of R.
How many persons sit between L and G?

**Options:**

    A. 8

    B. 3

    C. 4

    D. 5

**Correct Answer:** D
**Model Selected Answer:** B
**Subject Domain:** Problem Solving
 **Error Analysis:**

- **Nature of Error:** Miscalculation in linear arrangement reasoning.
- **Underlying Cause:** The model incorrectly interpreted the positions from the textual description, failing to properly count the number of people between L and G.
- **Recommendation:** Strengthen stepwise logical reasoning on linear arrangement and seating problems, including visualizing relative positions.

**Question:**
एडवांस प्राइसिंग एग्रीमेंट्स निम्नलिखित में से किसमें उपयोग होने वाला शब्द है?

**Options:**

    A. अंतर्राष्ट्रीय संधि

    B. केंद्रीय बजट

    C. जीएसटी कार्यान्वयन

    D. कराधान

**Correct Answer:** D
**Model Selected Answer:** B
**Subject Domain:** Taxation & Regulatory Compliance
 **Error Analysis:**

- **Nature of Error:** Confusion regarding the context of terminology usage.
- **Underlying Cause:** The model associated budgeting context instead of tax compliance context, leading to selection of a non-relevant option.
- **Recommendation:** Include context-specific knowledge about taxation terminology and international agreements in the training data.

**Question:**
ग्लोबल डिपॉजिटरी रसीद क्या है?

**Options:**

    A. भारत में एक विदेशी कंपनी द्वारा जारी बॉन्ड

    B. एक डिपॉजिटरी बैंक द्वारा जारी प्रमाण पत्र जो एक विदेशी कंपनी के शेयरों का प्रतिनिधित्व करता है

    C. म्यूचुअल फंड योजना का एक प्रकार

    D. विदेशी बाजारों में जारी सरकारी प्रतिभूति

**Correct Answer:** B
**Model Selected Answer:** D
**Subject Domain:** Financial Markets
**Error Analysis:**

- **Nature of Error:** Misidentification of financial instrument.
- **Underlying Cause:** The model selected a government security in foreign markets instead of recognizing the depository receipt issued by a bank for foreign company shares.
- **Recommendation:** Provide clearer definitions of financial instruments like GDRs, ADRs, and bonds in model knowledge base.

### E.3.3 BBK QUALITATIVE ANALYSIS

Agricultural domain errors stem from gaps in crop-specific practices, breed characteristics, and interpretation of scientific explanations in Hindi. The model frequently generalizes concepts, leading to incorrect associations between schemes, cultivation techniques, or physiological traits. Including detailed domain-specific datasets and traditional agronomy knowledge can help the model distinguish between nuanced agricultural concepts.

**Question:**
Which of the following is the objective of Atal Bhujal Yojana?

**Options:**

    A. Drip Irrigation

    B. Hydro-electric power production

    C. Drinking water supply

    D. Groundwater Management

**Correct Answer:** D
**Model Selected Answer:** A
**Subject Domain:** Agronomy

**Error Analysis:**

- **Nature of Error:** Confusion between water management schemes and irrigation programs.
- **Underlying Cause:** The model associated the scheme with agricultural water use but misidentified its focus, mistaking groundwater governance for irrigation techniques.
- **Recommendation:** Incorporate structured knowledge regarding Indian water-related government schemes and their specific objectives.

**Question:**
What are the horn characteristics of Murrah breed?

**Options:**

    A. Flat and sickle shaped and form hook at the tip

    B. Small and coiled tightly

    C. Curl slightly outwards

    D. Flat, short, tightly spirally curving inwards

**Correct Answer:** D
**Model Selected Answer:** C
**Subject Domain:** Animal Sciences

**Error Analysis:**

- **Nature of Error:** Breed-specific trait confusion.
- **Underlying Cause:** The model generalized bovine horn types without differentiating breed-specific characteristics.
- **Recommendation:** Include detailed breed-specific morphology datasets to improve precision.

**Question:**
धान की फसल की जलमग्नता सह लेती है, क्योंकि

**Options:**

    A. पौधे को औक्सीजन की जरूरत नहीं होती

    B. पौधों में पत्तियों में से औक्सीजन ले जाने की प्रक्रिया होती है

    C. जीवाणु मूल परिवेश को औक्सीकृत रखते हैं

    D. हवा के साथ जड़ों के क्षेत्र में प्रवेश कर रही औक्सीजन पर्याप्त है

**Correct Answer:** B
**Model Selected Answer:** C
**Subject Domain:** Agronomy
**Error Analysis:**

- **Nature of Error:** Hindi comprehension and scientific reasoning.
- **Underlying Cause:** The model misinterpreted the physiological explanation for rice root tolerance to waterlogging.
- **Recommendation:** Include domain-specific Hindi texts explaining crop physiology and waterlogging tolerance.

**Question:**
किस फसल की बुवाई से पहले मिट्टी पलट हल से दो बार जुताई करते हैं?

**Options:**

    A. अरहर

    B. मक्का

    C. चना

    D. गन्ना

**Correct Answer:** D
**Model Selected Answer:** B
**Subject Domain:** Agronomy

- **Nature of Error:** Crop management knowledge gap.
- **Underlying Cause:** The model failed to correctly associate double plowing with sugarcane cultivation.
- **Recommendation:** Include crop-specific cultivation practices and traditional agronomy knowledge in the training data.

### E.3.4 BBL QUALITATIVE ANALYSIS

In the legal domain, Llama-3.1-8B exhibits errors primarily related to procedural understanding, definitional knowledge, and interpretation of civil, criminal, and property law provisions. Common mistakes include misclassifying offences, misapplying procedural codes, and assuming incorrect registration or filing requirements. Errors often arise from insufficient knowledge of statutory definitions, procedural sequences, and conditions for specific legal actions. Strengthening explicit legal knowledge, clarifying judicial powers, and reinforcing procedural rules can improve the model's factual correctness and reasoning for law-related queries.

**Question:**
Question: Suit for recovery of money in promissory notes can be filed:

**Options:**

    A. under normal procedure

    B. under summary procedure as laid down in Order 37, CPC

    C. in the High Court

    D. as a writ petition

**Correct Answer:** B
**Model Selected Answer:** C
**Subject Domain:** Civil Litigation & Procedure
**Error Analysis:**

- **Nature of Error:** Confusion regarding procedural rules for recovery suits.
- **Underlying Cause:** The model incorrectly associated the filing with the High Court instead of the summary procedure under Order 37 CPC.
- **Recommendation:** Reinforce knowledge of civil procedural codes and specific rules for summary recovery of promissory notes.

**Question:**
Question: Use of violence by a member of an assembly of five or more person in furtherance of common object will constitute the offence of

**Options:**

    A. Affray

    B. Assault

    C. Rioting

    D. Unlawful Assembly

**Correct Answer:** C
**Model Selected Answer:** D
**Subject Domain:** Criminal Law & Justice
**Error Analysis:**

- **Nature of Error:** Misclassification of the offence type.
- **Underlying Cause:** The model confused "rioting" with "unlawful assembly," missing the condition of violence by five or more persons in furtherance of common object.
- **Recommendation:** Reinforce understanding of key definitions and conditions for offences under criminal law.

**Question:**
जहां न्यायालय को किसी व्यक्ति के पास किसी दस्तावेज या वस्तु के होने के बारे में कोई ज्ञान नहीं है? क्या ऐसी स्थिति में न्यायालय खोज वारंट जारी कर सकता है:

**Options:**

    A. नहीं

    B. केवल उस स्थिति में जब किसी विशिष्ट वस्तु के बारे में ज्ञात हो

    C. हाँ

    D. जब कोई विशिष्ट स्थान या व्यक्ति निर्दिष्ट हो

**Correct Answer:** C
**Model Selected Answer:** B
**Subject Domain:** Civil Litigation & Procedure
**Error Analysis:**

- **Nature of Error:** Misinterpretation of procedural powers regarding search warrants.
- **Underlying Cause:** The model assumed prior knowledge of a specific document was required, instead of recognizing the court's general power to issue a search warrant.
- **Recommendation:** Emphasize the scope and limits of judicial powers in procedural law.

**Question:**
अचल संपत्ति का किराया अनिवार्य रूप से पंजीकृत नहीं है:

**Options:**

    A. वार्षिक किराया

    B. एक वर्ष की अवधि के लिए किराया

    C. वार्षिक किराया प्राप्त करने वाला किराया

    D. इनमें से कोई नहीं

**Correct Answer:** D
**Model Selected Answer:** A
**Subject Domain:** Real Estate & Property Law
 **Error Analysis:**

- **Nature of Error:** Incorrect assumption regarding registration requirement.

- **Underlying Cause:** The model assumed annual rent requires registration, overlooking the legal nuance that not all rent agreements require registration.

- **Recommendation:** Include precise rules on registration requirements for different types of rent agreements in real estate law.

### E.4 DATA INTEGRITY AND CONTAMINATION ANALYSIS

To ensure the validity of our benchmark evaluations and rule out potential data leakage, we perform a set of analyses covering perplexity-based checks, multiple-choice option shuffling, and the impact of increasing distractor options. These evaluations help verify that our datasets remain unbiased and challenging for state-of-the-art LLMs.

### E.4.1 PERPLEXITY-BASED DATA CONTAMINATION ANALYSIS

To verify the integrity of BhashaBench V1 and detect potential data contamination or leakage from pretraining, we conducted a detailed perplexity (PPL) analysis on Llama-3.1-8B and Gemma-2-9B models. For each dataset, we computed PPL over the entire multiple-choice items, including both the questions and all answer options. This ensures that the computed PPL reflects the model's familiarity with the phrasing of the dataset, rather than just the questions alone. Table 22 reports the PPL, average token-level loss, and the number of tokens evaluated for each benchmark. BhashaBench V1 datasets (BBA, BBF, BBK, BBL) show perplexity scores comparable to or higher than well-established benchmarks such as ARC-C, MMLU, and MILU, confirming minimal exposure of these items during pretraining. For instance, BBA exhibits high PPL on both English (15.50) and Hindi (10.39) for Llama-3.1-8B, while BBK shows relatively elevated PPL in Hindi (7.16 for Llama-3.1-8B). These results indicate that the datasets provide genuinely novel evaluation challenges.

### E.4.2 MULTIPLE-CHOICE OPTION SHUFFLING EXPERIMENT

To further validate the robustness of models against superficial cues, we conducted an option ordering experiment. Multiple-choice options in each question were shuffled using fixed seeds (42 and 123) to examine whether model performance depends on the original ordering. The "Base" column represents the original option order. Table 23 presents the performance of Llama-3.1-8B and Llama-3.2-3B across English and Hindi subsets of each domain. Overall, performance remains largely stable across different option orderings, with minor variations observed primarily in the Hindi subsets. This suggests that the benchmark's multiple-choice questions evaluate actual model understanding rather than positional biases, providing confidence in the integrity of the dataset for comparative evaluation.

### E.4.3 EFFECT OF SCALING THE NUMBER OF DISTRACTORS

To assess how task difficulty affects model discrimination, we increased the number of options from the base 4 to 5 and 6 options per question. Additional distractor options were generated determin-

Table 22: Perplexity (PPL) and Average Loss for Llama-3.1-8B and gemma-2-9b across BhashaBench V1 and other datasets

| Dataset | Language | Llama-3.1-8B | | | gemma-2-9b | | |
|---------|----------|------|-----------|-------------|------|-----------|-------------|
| | | PPL | Avg. Loss | Num. Tokens | PPL | Avg. Loss | Num. Tokens |
| ARC-C | English | 8.03 | 2.08 | 74,929 | 6.82 | 1.92 | 77,685 |
| | Hindi | 4.10 | 1.41 | 160,887 | 5.85 | 1.77 | 124,073 |
| MILU | English | 7.62 | 2.03 | 845,273 | 7.23 | 1.98 | 879,220 |
| | Hindi | 4.93 | 1.60 | 1,521,137 | 6.37 | 1.85 | 1,261,389 |
| MMLU | English | 7.61 | 2.03 | 1,502,590 | 7.03 | 1.95 | 1,553,587 |
| | Hindi | 4.22 | 1.44 | 3,024,075 | 6.21 | 1.83 | 2,292,437 |
| BBA | English | 15.50 | 2.74 | 445,077 | 23.20 | 3.14 | 438,301 |
| | Hindi | 10.39 | 2.34 | 373,338 | 16.80 | 2.82 | 331,130 |
| BBF | English | 6.78 | 1.91 | 1,710,390 | 5.86 | 1.77 | 1,821,417 |
| | Hindi | 4.03 | 1.39 | 1,007,527 | 5.04 | 1.62 | 799,074 |
| BBK | English | 6.14 | 1.81 | 965,854 | 6.34 | 1.85 | 980,286 |
| | Hindi | 7.16 | 1.97 | 211,467 | 9.54 | 2.26 | 183,744 |
| BBL | English | 7.28 | 1.98 | 1,511,635 | 7.19 | 1.97 | 1,567,174 |
| | Hindi | 4.01 | 1.39 | 1,071,001 | 5.79 | 1.76 | 825,463 |

Table 23: Performance of Llama-3.1-8B and Llama-3.2-3B across BhashaBench V1 (EN + HI) under different option orderings. Each "Seed" column corresponds to a different shuffling of the multiple-choice options, while "Base" uses the original ordering. Scores are reported for overall, English, and Hindi subsets of each domain.

| Domain | | Llama-3.1-8B | | | Llama-3.2-3B | | |
|---------|----------|-------|---------|----------|-------|---------|----------|
| Category | Language | Base | Seed 42 | Seed 123 | Base | Seed 42 | Seed 123 |
| **BBA** | Overall | 33.12 | 32.01 | 31.99 | 30.28 | 29.55 | 29.07 |
| | English | 35.48 | 34.20 | 34.20 | 31.62 | 31.14 | 30.43 |
| | Hindi | 29.17 | 28.37 | 28.30 | 28.05 | 26.91 | 26.79 |
| **BBF** | Overall | 34.48 | 33.62 | 33.46 | 31.46 | 30.60 | 30.82 |
| | English | 36.20 | 35.66 | 35.45 | 33.04 | 32.29 | 32.45 |
| | Hindi | 30.61 | 29.04 | 28.99 | 27.92 | 26.80 | 27.15 |
| **BBK** | Overall | 38.07 | 36.42 | 36.87 | 31.96 | 31.24 | 31.61 |
| | English | 39.52 | 37.97 | 38.41 | 32.68 | 31.96 | 32.34 |
| | Hindi | 31.41 | 29.31 | 29.82 | 28.69 | 27.93 | 28.29 |
| **BBL** | Overall | 38.44 | 37.61 | 37.46 | 33.17 | 32.52 | 32.41 |
| | English | 41.32 | 40.51 | 40.10 | 35.17 | 34.40 | 34.32 |
| | Hindi | 31.76 | 30.87 | 31.33 | 28.53 | 28.15 | 27.96 |

istically using fixed random seeds (seed 48 for the 5th option, seed 123 for the 6th option). These distractors consist of semantically meaningless random character sequences (5-8 lowercase letters) that test whether models can robustly filter out irrelevant choices. The generation process seeds Python's random number generator to create deterministic nonsense tokens from lowercase ASCII characters, ensuring reproducibility while introducing noise distractors clearly distinguishable from genuine options.

Table 24 presents the performance of Llama-3.1-8B and Llama-3.2-3B across all BhashaBench V1 domains with varying option counts. Results show consistent performance decline as options increase across both models and all domains. For Llama-3.1-8B, overall accuracy drops 2-7 percentage points from 4 to 6 options, with the steepest degradation in BBL (38.44% → 28.96%) and BBK (38.07% → 32.81%). Llama-3.2-3B exhibits similar but slightly steeper declines, particularly in BBL (33.17% → 22.89%).

The performance gap between English and Hindi widens substantially with additional options. Hindi subsets show more severe degradation Llama-3.1-8B's Hindi performance on BBL drops from 31.76% to 16.03%, compared to 41.32% to 34.52% in English. This disproportionate challenge in lower-resource languages suggests weaker multilingual reasoning capabilities. The significant drops even with nonsense distractors indicate models struggle with filtering irrelevant options, particularly in Hindi. These findings confirm the benchmark maintains discriminative power with additional options, enabling fine-grained model assessment.

Table 24: Performance of Llama-3.1-8B and Llama-3.2-3B across BhashaBench V1 domains with varying numbers of options. Scores are reported for overall, English, and Hindi subsets of each domain.

| Domain | | Llama-3.1-8B | | | Llama-3.2-3B | | |
|---|---|---|---|---|---|---|---|
| Category | Language | Base (4 Opts) | 5 Options | 6 Options | Base (4 Opts) | 5 Options | 6 Options |
| **BBA** | Overall | 33.12 | 31.14 | 29.02 | 30.28 | 26.9 | 25.7 |
| | English | 35.48 | 35.07 | 34.61 | 31.62 | 30.36 | 30.35 |
| | Hindi | 29.17 | 24.59 | 19.73 | 28.05 | 21.14 | 17.95 |
| **BBF** | Overall | 34.48 | 31.98 | 31.01 | 31.46 | 28.43 | 27.65 |
| | English | 36.2 | 34.18 | 33.61 | 33.04 | 31.33 | 30.48 |
| | Hindi | 30.61 | 27.05 | 25.18 | 27.92 | 21.92 | 21.3 |
| **BBK** | Overall | 38.07 | 33.88 | 32.81 | 31.96 | 28.98 | 26.85 |
| | English | 39.52 | 35.82 | 35.52 | 32.68 | 31.07 | 28.98 |
| | Hindi | 31.41 | 24.99 | 20.38 | 28.69 | 19.37 | 17.12 |
| **BBL** | Overall | 38.44 | 31.49 | 28.96 | 33.17 | 24.49 | 22.89 |
| | English | 41.32 | 36.59 | 34.52 | 35.17 | 30.43 | 28.63 |
| | Hindi | 31.76 | 19.6 | 16.03 | 28.53 | 10.66 | 9.52 |

## E.5 STATISTICAL SIGNIFICANCE TESTS

To assess whether the observed differences in model performance are statistically meaningful, we conduct a series of statistical significance tests. These tests help determine whether performance gaps between models are likely due to chance or reflect consistent trends across languages and benchmarks.

### E.5.1 WILSON CONFIDENCE INTERVAL FOR MODEL PERFORMANCE

To quantify the statistical reliability of model performance estimates and enable rigorous comparative analysis, we computed Wilson confidence intervals (CIs) for the zero-shot accuracy of each evaluated model. Wilson CIs provide a robust measure of uncertainty around estimated accuracy that is particularly well-suited for evaluation benchmarks. Unlike naive proportion-based intervals (such as normal approximation intervals), Wilson CIs employ a score-based method that offers several critical advantages: they maintain more accurate coverage probabilities, especially for extreme accuracy values near 0% or 100%; they naturally respect the bounded nature of proportion metrics

Table 25: Zero-shot performance (%) of LLMs on the BBA with Wilson confidence intervals. Values in brackets indicate the Wilson CIs.

| Model | Eng (%) [95% CI] | Hin (%) [95% CI] | Avg (%) [95% CI] |
|---|---|---|---|
| *< 4B Models* | | | |
| gemma-3-270m | 28.08 [27.18, 29.00] | 28.25 [27.08, 29.44] | 28.14 [27.43, 28.87] |
| gemma-3-270m-it | 26.23 [25.35, 27.13] | 25.77 [24.64, 26.93] | 26.06 [25.36, 26.77] |
| Param-1 | 41.12 [40.13, 42.12] | 38.04 [36.78, 39.32] | 39.97 [39.18, 40.75] |
| gemma-2-2b | 36.80 [35.83, 37.78] | 30.61 [29.42, 31.83] | 34.48 [33.72, 35.24] |
| gemma-2-2b-it | 29.38 [28.46, 30.31] | 26.79 [25.64, 27.96] | 28.40 [27.69, 29.13] |
| Llama-3.2-1B | 29.17 [28.26, 30.10] | 26.30 [25.17, 27.47] | 28.10 [27.38, 28.82] |
| Llama-3.2-1B-Instruct | 26.77 [25.88, 27.67] | 25.82 [24.70, 26.98] | 26.41 [25.71, 27.12] |
| Llama-3.2-3B | 31.62 [30.69, 32.57] | 28.05 [26.89, 29.24] | 30.28 [29.55, 31.02] |
| Llama-3.2-3B-Instruct | 35.31 [34.35, 36.29] | 29.67 [28.49, 30.88] | 33.20 [32.45, 33.95] |
| sarvam-2b-v0.5 | 26.79 [25.90, 27.69] | 27.07 [25.92, 28.25] | 26.89 [26.19, 27.61] |
| sarvam-1 | 29.70 [28.78, 30.63] | 28.41 [27.24, 29.60] | 29.21 [28.49, 29.95] |
| Nemotron-4-Mini-Hindi-4B-Base | 34.76 [33.80, 35.73] | 32.82 [31.61, 34.06] | 34.03 [33.28, 34.79] |
| Nemotron-4-Mini-Hindi-4B-Instruct | 33.38 [32.43, 34.34] | 33.82 [32.59, 35.07] | 33.54 [32.79, 34.30] |
| Qwen2.5-3B | 40.61 [39.62, 41.61] | 31.90 [30.69, 33.13] | 37.34 [36.57, 38.12] |
| Qwen2.5-3B-Instruct | 35.22 [34.25, 36.19] | 28.46 [27.29, 29.65] | 32.68 [31.93, 33.44] |
| granite-3.1-2b-instruct | 33.39 [32.44, 34.35] | 27.30 [26.15, 28.48] | 31.10 [30.37, 31.85] |
| granite-3.1-3b-a800m-base | 31.75 [30.81, 32.70] | 26.18 [25.05, 27.35] | 29.66 [28.93, 30.40] |
| *7B to 27B Models* | | | |
| Pangea-7B | 40.69 [39.70, 41.69] | 31.93 [30.73, 33.16] | 37.41 [36.63, 38.18] |
| Indic-gemma-7b-finetuned-sft-Navarasa-2.0 | 37.12 [36.15, 38.10] | 31.83 [30.62, 33.06] | 35.13 [34.37, 35.90] |
| aya-23-8B | 33.84 [32.88, 34.80] | 28.87 [27.70, 30.07] | 31.97 [31.23, 32.72] |
| Llama-3.1-8B | 35.48 [34.52, 36.46] | 29.17 [28.00, 30.37] | 33.12 [32.37, 33.87] |
| Llama-3.1-8B-Instruct | 36.86 [35.89, 37.85] | 31.26 [30.06, 32.48] | 34.76 [34.00, 35.53] |
| gemma-2-9b | 48.16 [47.15, 49.17] | 37.92 [36.66, 39.19] | 44.32 [43.52, 45.11] |
| gemma-2-9b-it | 36.22 [35.25, 37.20] | 31.18 [29.99, 32.41] | 34.33 [33.57, 35.10] |
| gpt-oss-20b | 38.30 [37.32, 39.29] | 33.09 [31.87, 34.33] | 36.34 [35.58, 37.12] |
| gemma-2-27b | 50.70 [49.68, 51.71] | 42.26 [40.98, 43.56] | 47.53 [46.73, 48.33] |
| gemma-2-27b-it | 40.45 [39.46, 41.45] | 33.89 [32.66, 35.14] | 37.99 [37.21, 38.77] |
| *> 27B Models* | | | |
| gpt-oss-120b | 55.62 [54.61, 56.62] | 48.05 [46.74, 49.36] | 52.78 [51.98, 53.58] |
| Qwen3-235B-A22B-Instruct-25076 | 60.25 [59.25, 61.24] | 54.78 [53.48, 56.08] | 58.20 [57.40, 58.98] |
| deepseek-v3 | 51.38 [50.37, 52.39] | 37.03 [35.77, 38.30] | 45.99 [45.20, 46.79] |
| gpt-4o | 62.75 [61.77, 63.73] | 54.73 [53.42, 56.03] | 59.74 [58.95, 60.52] |

(always yielding intervals within [0, 1]); and they perform reliably even with relatively small sample sizes, making them ideal for domain-specific benchmark subsets.

Tables 25, 26, 27, and 28 report the zero-shot accuracy (%) for English and Hindi subsets across all evaluated models for each BhashaBench domain (BBA, BBF, BBK, and BBL respectively), accompanied by their corresponding 95% Wilson CIs. Values presented in brackets [lower, upper] denote the lower and upper bounds of the confidence interval, representing the range within which the true population accuracy is expected to lie with 95% confidence, given the observed sample performance.

These intervals enable both visual and quantitative comparison of model performance while explicitly accounting for sampling uncertainty inherent in finite test sets. Models whose confidence intervals do not overlap can be interpreted as exhibiting statistically significant differences in performance at the 95% confidence level, providing strong evidence that observed accuracy differences reflect genuine capability gaps rather than random variation. Conversely, when confidence intervals overlap substantially, performance differences should be interpreted more cautiously, as they may fall within the margin of statistical uncertainty. This interval-based analysis is particularly valuable for identifying robust performance trends across languages and domains, and for determining whether improvements from model scaling or architectural changes represent statistically meaningful advances. The Wilson CI framework thus strengthens the benchmark's utility for reliable model comparison and selection decisions.

Table 26: Zero-shot performance (%) of LLMs on the BBF with Wilson confidence intervals. Values in brackets indicate the Wilson CIs.

| Model | Eng (%) [95% CI] | Hin (%) [95% CI] | Avg (%) [95% CI] |
|---|---|---|---|
| *< 4B Models* | | | |
| gemma-3-270m | 24.98 [24.26, 25.72] | 25.06 [23.98, 26.17] | 25.00 [24.40, 25.62] |
| gemma-3-270m-it | 24.13 [23.42, 24.86] | 23.84 [22.78, 24.93] | 24.04 [23.45, 24.65] |
| Param-1 | 32.24 [31.46, 33.04] | 29.56 [28.41, 30.72] | 31.42 [30.77, 32.07] |
| gemma-2-2b | 34.20 [33.40, 35.00] | 27.50 [26.38, 28.64] | 32.14 [31.48, 32.80] |
| gemma-2-2b-it | 31.26 [30.48, 32.05] | 27.93 [26.81, 29.08] | 30.24 [29.60, 30.89] |
| Llama-3.2-1B | 28.24 [27.48, 29.00] | 25.61 [24.52, 26.73] | 27.43 [26.80, 28.06] |
| Llama-3.2-1B-Instruct | 26.28 [25.54, 27.03] | 26.04 [24.95, 27.17] | 26.21 [25.59, 26.83] |
| Llama-3.2-3B | 33.04 [32.25, 33.84] | 27.92 [26.79, 29.07] | 31.46 [30.81, 32.12] |
| Llama-3.2-3B-Instruct | 32.94 [32.15, 33.74] | 29.09 [27.95, 30.25] | 31.76 [31.10, 32.41] |
| sarvam-2b-v0.5 | 26.42 [25.68, 27.17] | 25.31 [24.22, 26.43] | 26.08 [25.47, 26.70] |
| sarvam-1 | 29.66 [28.89, 30.43] | 27.27 [26.15, 28.41] | 28.92 [28.29, 29.56] |
| Nemotron-4-Mini-Hindi-4B-Base | 34.95 [34.15, 35.76] | 31.41 [30.25, 32.60] | 33.86 [33.20, 34.53] |
| Nemotron-4-Mini-Hindi-4B-Instruct | 31.98 [31.20, 32.78] | 30.06 [28.91, 31.23] | 31.39 [30.74, 32.05] |
| Qwen2.5-3B | 39.54 [38.72, 40.37] | 32.13 [30.96, 33.32] | 37.26 [36.58, 37.94] |
| Qwen2.5-3B-Instruct | 34.84 [34.04, 35.65] | 29.17 [28.03, 30.34] | 33.09 [32.44, 33.76] |
| granite-3.1-2b-instruct | 32.82 [32.03, 33.62] | 27.11 [26.00, 28.26] | 31.07 [30.42, 31.72] |
| granite-3.1-3b-a800m-base | 29.22 [28.45, 29.99] | 24.17 [23.10, 25.27] | 27.66 [27.04, 28.30] |
| *7B to 27B Models* | | | |
| Pangea-7B | 41.71 [40.88, 42.54] | 33.73 [32.55, 34.94] | 39.25 [38.57, 39.94] |
| Indic-gemma-7b-finetuned-sft-Navarasa-2.0 | 37.00 [36.19, 37.82] | 30.47 [29.32, 31.65] | 34.99 [34.32, 35.67] |
| aya-23-8B | 35.25 [34.44, 36.06] | 30.88 [29.72, 32.06] | 33.90 [33.24, 34.57] |
| Llama-3.1-8B | 36.20 [35.39, 37.01] | 30.61 [29.45, 31.79] | 34.48 [33.81, 35.15] |
| Llama-3.1-8B-Instruct | 35.68 [34.87, 36.49] | 30.27 [29.12, 31.45] | 34.01 [33.35, 34.68] |
| gemma-2-9b | 42.73 [41.90, 43.57] | 36.91 [35.70, 38.14] | 40.94 [40.25, 41.63] |
| gemma-2-9b-it | 38.85 [38.03, 39.68] | 32.03 [30.86, 33.22] | 36.75 [36.08, 37.43] |
| gpt-oss-20b | 37.11 [36.30, 37.93] | 32.61 [31.44, 33.81] | 35.73 [35.06, 36.40] |
| gemma-2-27b | 47.79 [46.94, 48.63] | 41.24 [40.00, 42.49] | 45.77 [45.07, 46.47] |
| gemma-2-27b-it | 42.47 [41.64, 43.31] | 34.29 [33.09, 35.50] | 39.95 [39.27, 40.64] |
| *> 27B Models* | | | |
| gpt-oss-120b | 74.11 [73.37, 74.85] | 64.16 [62.94, 65.36] | 71.05 [70.41, 71.68] |
| Qwen3-235B-A22B-Instruct-25076 | 63.72 [62.90, 64.53] | 56.27 [55.01, 57.52] | 61.43 [60.74, 62.11] |
| deepseek-v3 | 63.46 [62.64, 64.27] | 57.04 [55.78, 58.29] | 61.48 [60.80, 62.16] |
| gpt-4o | 57.27 [56.43, 58.10] | 49.82 [48.55, 51.08] | 54.97 [54.27, 55.67] |

### E.5.2 STATISTICAL SIGNIFICANCE OF MODEL PERFORMANCE DIFFERENCES USING MCNEMAR'S TEST

To evaluate whether the observed differences in accuracy between the top-performing LLMs are statistically meaningful, we employ **McNemar's test** (Mcnemar, 1947). This non-parametric test is specifically designed for paired nominal data and is commonly used to compare two classifiers on the same dataset by focusing on instances where their predictions disagree. It provides a robust measure of whether one model consistently outperforms another beyond random chance.

In our analysis, the **Accuracy Diff (%)** column reports the absolute difference in accuracy between a pair of models. The **McNemar Stat** quantifies the magnitude of disagreement in model predictions, while the corresponding **p-value** assesses the statistical significance of this difference. We consider a difference significant if $p < 0.05$, implying that one model's performance is reliably better than the other on the same set of questions.

Tables 29–32 show pairwise McNemar comparisons of the top 5 models across four benchmark domains (BBA, BBF, BBK, BBL) for both English and Hindi datasets. Entries labeled "Yes" in the **Significant** column indicate that the performance difference is statistically significant, confirming consistent superiority of one model over another. Conversely, entries marked "No" denote cases where the observed accuracy difference could be due to chance, suggesting that the two models have comparable performance. This analysis complements raw accuracy scores by highlighting which improvements are robust and reliable rather than incidental.

Table 27: Zero-shot performance (%) of LLMs on the BBK with Wilson confidence intervals. Values in brackets indicate the Wilson CIs.

| Model | Eng (%) [95% CI] | Hin (%) [95% CI] | Avg (%) [95% CI] |
|---|---|---|---|
| *< 4B Models* | | | |
| gemma-3-270m | 26.64 [25.87, 27.41] | 24.45 [22.88, 26.09] | 26.24 [25.56, 26.95] |
| gemma-3-270m-it | 27.44 [26.66, 28.22] | 25.35 [23.76, 27.01] | 27.06 [26.37, 27.77] |
| Param-1 | 33.10 [32.28, 33.92] | 27.97 [26.32, 29.67] | 32.18 [31.44, 32.92] |
| gemma-2-2b | 41.24 [40.38, 42.10] | 27.49 [25.86, 29.19] | 38.78 [38.01, 39.55] |
| gemma-2-2b-it | 35.94 [35.11, 36.78] | 27.71 [26.07, 29.41] | 34.47 [33.72, 35.22] |
| Llama-3.2-1B | 29.71 [28.92, 30.51] | 25.21 [23.62, 26.86] | 28.91 [28.20, 29.63] |
| Llama-3.2-1B-Instruct | 29.16 [28.37, 29.96] | 26.33 [24.72, 28.01] | 28.65 [27.94, 29.37] |
| Llama-3.2-3B | 32.68 [31.87, 33.50] | 28.69 [27.03, 30.41] | 31.96 [31.23, 32.70] |
| Llama-3.2-3B-Instruct | 40.59 [39.74, 41.45] | 29.09 [27.42, 30.81] | 38.53 [37.77, 39.30] |
| sarvam-2b-v0.5 | 28.14 [27.36, 28.93] | 25.57 [23.98, 27.23] | 27.68 [26.98, 28.39] |
| sarvam-1 | 30.82 [30.02, 31.63] | 27.57 [25.93, 29.26] | 30.24 [29.52, 30.97] |
| Nemotron-4-Mini-Hindi-4B-Base | 36.67 [35.83, 37.51] | 36.49 [34.71, 38.30] | 36.64 [35.88, 37.40] |
| Nemotron-4-Mini-Hindi-4B-Instruct | 35.83 [35.00, 36.67] | 35.33 [33.57, 37.13] | 35.74 [34.99, 36.50] |
| Qwen2.5-3B | 44.57 [43.70, 45.44] | 32.72 [30.99, 34.49] | 42.45 [41.67, 43.23] |
| Qwen2.5-3B-Instruct | 42.67 [41.81, 43.53] | 27.20 [25.57, 28.90] | 39.90 [39.13, 40.68] |
| granite-3.1-2b-instruct | 37.71 [36.87, 38.56] | 27.86 [26.21, 29.56] | 35.95 [35.20, 36.71] |
| granite-3.1-3b-a800m-base | 33.36 [32.55, 34.19] | 26.70 [25.08, 28.38] | 32.17 [31.44, 32.91] |
| *7B to 27B Models* | | | |
| Pangea-7B | 47.16 [46.29, 48.03] | 34.71 [32.96, 36.51] | 44.93 [44.15, 45.72] |
| Indic-gemma-7b-finetuned-sft-Navarasa-2.0 | 42.31 [41.46, 43.18] | 33.44 [31.71, 35.23] | 40.73 [39.95, 41.51] |
| aya-23-8B | 37.09 [36.25, 37.93] | 33.22 [31.49, 35.00] | 36.40 [35.64, 37.16] |
| Llama-3.1-8B | 39.52 [38.68, 40.38] | 31.41 [29.71, 33.17] | 38.07 [37.31, 38.84] |
| Llama-3.1-8B-Instruct | 47.14 [46.27, 48.01] | 35.07 [33.31, 36.88] | 44.98 [44.19, 45.77] |
| gemma-2-9b | 55.23 [54.37, 56.10] | 43.89 [42.05, 45.75] | 53.20 [52.41, 53.99] |
| gemma-2-9b-it | 48.92 [48.05, 49.80] | 36.45 [34.68, 38.27] | 46.69 [45.91, 47.48] |
| gpt-oss-20b | 46.58 [45.71, 47.45] | 36.27 [34.50, 38.08] | 44.73 [43.95, 45.52] |
| gemma-2-27b | 59.84 [58.98, 60.69] | 50.38 [48.52, 52.25] | 58.14 [57.36, 58.92] |
| gemma-2-27b-it | 54.95 [54.08, 55.81] | 41.24 [39.42, 43.09] | 52.50 [51.71, 53.28] |
| *> 27B Models* | | | |
| gpt-oss-120b | 71.40 [70.61, 72.18] | 60.25 [58.41, 62.06] | 69.41 [68.67, 70.13] |
| Qwen3-235B-A22B-Instruct-25076 | 74.57 [73.80, 75.32] | 64.13 [62.32, 65.90] | 72.70 [71.99, 73.39] |
| deepseek-v3 | 62.93 [62.09, 63.77] | 45.01 [43.16, 46.88] | 59.73 [58.95, 60.50] |
| gpt-4o | 75.31 [74.55, 76.05] | 65.18 [63.38, 66.94] | 73.50 [72.79, 74.19] |

Table 28: Zero-shot performance (%) of LLMs on the BBL with Wilson confidence intervals. Values in brackets indicate the Wilson CIs.

| Model | Eng (%) [95% CI] | Hin (%) [95% CI] | Avg (%) [95% CI] |
|---|---|---|---|
| *< 4B Models* | | | |
| gemma-3-270m | 25.49 [24.85, 26.15] | 25.54 [24.55, 26.55] | 25.51 [24.96, 26.06] |
| gemma-3-270m-it | 25.56 [24.91, 26.22] | 27.26 [26.25, 28.29] | 26.07 [25.52, 26.63] |
| Param-1 | 36.15 [35.43, 36.88] | 32.89 [31.82, 33.98] | 35.17 [34.58, 35.78] |
| gemma-2-2b | 38.45 [37.72, 39.18] | 29.61 [28.58, 30.67] | 35.79 [35.19, 36.40] |
| gemma-2-2b-it | 34.49 [33.78, 35.21] | 30.25 [29.21, 31.32] | 33.22 [32.63, 33.81] |
| Llama-3.2-1B | 29.63 [28.95, 30.32] | 25.88 [24.89, 26.90] | 28.50 [27.94, 29.07] |
| Llama-3.2-1B-Instruct | 29.08 [28.40, 29.76] | 27.04 [26.04, 28.07] | 28.47 [27.90, 29.04] |
| Llama-3.2-3B | 35.17 [34.45, 35.89] | 28.53 [27.51, 29.58] | 33.17 [32.59, 33.77] |
| Llama-3.2-3B-Instruct | 39.74 [39.01, 40.48] | 30.13 [29.09, 31.19] | 36.86 [36.25, 37.46] |
| sarvam-2b-v0.5 | 28.49 [27.81, 29.17] | 25.95 [24.96, 26.97] | 27.72 [27.17, 28.29] |
| sarvam-1 | 30.92 [30.23, 31.62] | 26.66 [25.66, 27.69] | 29.64 [29.07, 30.22] |
| Nemotron-4-Mini-Hindi-4B-Base | 40.75 [40.01, 41.49] | 37.55 [36.45, 38.67] | 39.79 [39.17, 40.40] |
| Nemotron-4-Mini-Hindi-4B-Instruct | 36.99 [36.26, 37.71] | 34.11 [33.03, 35.20] | 36.12 [35.52, 36.73] |
| Qwen2.5-3B | 44.98 [44.23, 45.72] | 33.97 [32.89, 35.06] | 41.67 [41.05, 42.29] |
| Qwen2.5-3B-Instruct | 40.62 [39.88, 41.36] | 29.89 [28.85, 30.94] | 37.39 [36.79, 38.00] |
| granite-3.1-2b-instruct | 38.18 [37.45, 38.91] | 27.30 [26.29, 28.33] | 34.91 [34.31, 35.51] |
| granite-3.1-3b-a800m-base | 33.74 [33.04, 34.46] | 24.01 [23.04, 25.00] | 30.82 [30.24, 31.40] |
| *7B to 27B Models* | | | |
| Pangea-7B | 48.70 [47.95, 49.45] | 34.95 [33.87, 36.06] | 44.57 [43.95, 45.20] |
| Indic-gemma-7b-finetuned-sft-Navarasa-2.0 | 44.08 [43.34, 44.83] | 34.09 [33.02, 35.19] | 41.08 [40.47, 41.70] |
| aya-23-8B | 41.92 [41.18, 42.66] | 33.01 [31.95, 34.10] | 39.24 [38.63, 39.86] |
| Llama-3.1-8B | 41.32 [40.58, 42.06] | 31.76 [30.70, 32.83] | 38.44 [37.84, 39.06] |
| Llama-3.1-8B-Instruct | 48.61 [47.86, 49.36] | 36.47 [35.38, 37.58] | 44.96 [44.34, 45.59] |
| gemma-2-9b | 58.49 [57.75, 59.23] | 42.96 [41.83, 44.10] | 53.83 [53.20, 54.45] |
| gemma-2-9b-it | 45.05 [44.31, 45.80] | 38.66 [37.55, 39.78] | 43.13 [42.51, 43.75] |
| gpt-oss-20b | 40.69 [39.96, 41.43] | 35.24 [34.16, 36.34] | 39.06 [38.45, 39.67] |
| gemma-2-27b | 64.91 [64.19, 65.63] | 51.83 [50.69, 52.97] | 60.99 [60.37, 61.60] |
| gemma-2-27b-it | 50.71 [49.96, 51.46] | 42.02 [40.89, 43.15] | 48.10 [47.47, 48.73] |
| *> 27B Models* | | | |
| gpt-oss-120b | 70.72 [70.03, 71.40] | 62.94 [61.83, 64.04] | 68.38 [67.80, 68.97] |
| Qwen3-235B-A22B-Instruct-25076 | 80.15 [79.55, 80.75] | 68.60 [67.53, 69.65] | 76.68 [76.15, 77.21] |
| deepseek-v3 | 67.78 [67.07, 68.47] | 46.78 [45.63, 47.92] | 61.47 [60.86, 62.08] |
| gpt-4o | 78.83 [78.22, 79.44] | 71.02 [69.97, 72.04] | 76.49 [75.95, 77.02] |

Table 29: Pairwise comparison of the top 5 LLMs on the BBA domain using McNemar's test. Accuracy Diff (%) shows the absolute difference between model accuracies. Significant differences are reported at p < 0.05.

| Model A | Accuracy A | Model B | Accuracy B | Accuracy Diff (%) | McNemar Stat | p-value | Significant |
|---|---|---|---|---|---|---|---|
| **English** | | | | | | | |
| gpt-4o | 62.75% | Qwen3-235B-A22B-Instruct-25076 | 60.25% | 2.50% | 23.874 | 1.029e-06 | Yes |
| gpt-oss-120b | 55.62% | gemma-2-27b | 50.70% | 4.92% | 45.308 | 1.684e-11 | Yes |
| gpt-oss-120b | 55.62% | deepseek-v3 | 51.38% | 4.24% | 54.175 | 1.834e-13 | Yes |
| Qwen3-235B-A22B-Instruct-25076 | 60.25% | gemma-2-27b | 50.70% | 9.55% | 171.369 | 3.718e-39 | Yes |
| Qwen3-235B-A22B-Instruct-25076 | 60.25% | deepseek-v3 | 51.38% | 8.87% | 228.148 | 1.511e-51 | Yes |
| deepseek-v3 | 51.38% | gemma-2-27b | 50.70% | 0.68% | 0.855 | 0.3552 | No |
| gpt-4o | 62.75% | gemma-2-27b | 50.70% | 12.06% | 269.474 | 1.476e-60 | Yes |
| gpt-4o | 62.75% | deepseek-v3 | 51.38% | 11.37% | 357.026 | 1.251e-79 | Yes |
| gpt-4o | 62.75% | gpt-oss-120b | 55.62% | 7.14% | 173.739 | 1.129e-39 | Yes |
| Qwen3-235B-A22B-Instruct-25076 | 60.25% | gpt-oss-120b | 55.62% | 4.63% | 68.941 | 1.014e-16 | Yes |
| **Hindi** | | | | | | | |
| gpt-oss-120b | 48.05% | param-1 | 38.04% | 10.01% | 126.394 | 2.521e-29 | Yes |
| Qwen3-235B-A22B-Instruct-25076 | 54.78% | gpt-4o | 54.73% | 0.05% | 0.003 | 0.9591 | No |
| Qwen3-235B-A22B-Instruct-25076 | 54.78% | gpt-oss-120b | 48.05% | 6.73% | 72.961 | 1.322e-17 | Yes |
| Qwen3-235B-A22B-Instruct-25076 | 54.78% | gemma-2-27b | 42.26% | 12.52% | 174.197 | 8.964e-40 | Yes |
| Qwen3-235B-A22B-Instruct-25076 | 54.78% | param-1 | 38.04% | 16.74% | 367.997 | 5.109e-82 | Yes |
| gpt-4o | 54.73% | gpt-oss-120b | 48.05% | 6.68% | 75.405 | 3.834e-18 | Yes |
| gpt-4o | 54.73% | gemma-2-27b | 42.26% | 12.47% | 175.629 | 4.363e-40 | Yes |
| gpt-4o | 54.73% | param-1 | 38.04% | 16.69% | 367.336 | 7.117e-82 | Yes |
| gpt-oss-120b | 48.05% | gemma-2-27b | 42.26% | 5.79% | 37.345 | 9.899e-10 | Yes |
| gemma-2-27b | 42.26% | param-1 | 38.04% | 4.22% | 20.868 | 4.921e-06 | Yes |

Table 30: Pairwise comparison of the top 5 LLMs on the BBF domain using McNemar's test. Accuracy Diff (%) shows the absolute difference between model accuracies. Significant differences are reported at p < 0.05.

| Model A | Accuracy A | Model B | Accuracy B | Accuracy Diff (%) | McNemar Stat | p-value | Significant |
|---|---|---|---|---|---|---|---|
| **English** | | | | | | | |
| gpt-oss-120b | 74.11% | Qwen3-235B-A22B-Instruct-25076 | 63.72% | 10.39% | 471.859 | 1.262e-104 | Yes |
| deepseek-v3 | 63.46% | gemma-2-27b | 47.79% | 15.67% | 659.063 | 2.387e-145 | Yes |
| deepseek-v3 | 63.46% | gpt-4o | 57.27% | 6.19% | 174.76 | 6.755e-40 | Yes |
| Qwen3-235B-A22B-Instruct-25076 | 63.72% | gemma-2-27b | 47.79% | 15.93% | 668.926 | 1.711e-147 | Yes |
| Qwen3-235B-A22B-Instruct-25076 | 63.72% | gpt-4o | 57.27% | 6.45% | 199.599 | 2.555e-45 | Yes |
| gpt-4o | 57.27% | gemma-2-27b | 47.79% | 9.48% | 239.639 | 4.714e-54 | Yes |
| gpt-oss-120b | 74.11% | gemma-2-27b | 47.79% | 26.33% | 1818.018 | 0.0 | Yes |
| gpt-oss-120b | 74.11% | gpt-4o | 57.27% | 16.85% | 1025.635 | 4.811e-225 | Yes |
| gpt-oss-120b | 74.11% | deepseek-v3 | 63.46% | 10.65% | 497.362 | 3.564e-110 | Yes |
| Qwen3-235B-A22B-Instruct-25076 | 63.72% | deepseek-v3 | 63.46% | 0.26% | 0.35 | 0.5544 | No |
| **Hindi** | | | | | | | |
| Qwen3-235B-A22B-Instruct-25076 | 56.27% | gemma-2-27b | 41.24% | 15.03% | 257.555 | 5.855e-58 | Yes |
| gpt-oss-120b | 64.16% | deepseek-v3 | 57.04% | 7.12% | 94.667 | 2.251e-22 | Yes |
| gpt-oss-120b | 64.16% | Qwen3-235B-A22B-Instruct-25076 | 56.27% | 7.89% | 109.714 | 1.132e-25 | Yes |
| gpt-oss-120b | 64.16% | gpt-4o | 49.82% | 14.34% | 315.756 | 1.217e-70 | Yes |
| gpt-oss-120b | 64.16% | gemma-2-27b | 41.24% | 22.92% | 591.15 | 1.409e-130 | Yes |
| deepseek-v3 | 57.04% | Qwen3-235B-A22B-Instruct-25076 | 56.27% | 0.77% | 1.259 | 0.2618 | No |
| deepseek-v3 | 57.04% | gpt-4o | 49.82% | 7.22% | 95.262 | 1.667e-22 | Yes |
| deepseek-v3 | 57.04% | gemma-2-27b | 41.24% | 15.80% | 288.861 | 8.805e-65 | Yes |
| Qwen3-235B-A22B-Instruct-25076 | 56.27% | gpt-4o | 49.82% | 6.45% | 81.802 | 1.504e-19 | Yes |
| gpt-4o | 49.82% | gemma-2-27b | 41.24% | 8.58% | 86.602 | 1.327e-20 | Yes |

Table 31: Pairwise comparison of the top 5 LLMs on the BBK domain using McNemar's test. Accuracy Diff (%) shows the absolute difference between model accuracies. Significant differences are reported at p < 0.05.

| Model A | Accuracy A | Model B | Accuracy B | Accuracy Diff (%) | McNemar Stat | p-value | Significant |
|---|---|---|---|---|---|---|---|
| **English** | | | | | | | |
| gpt-4o | 75.31% | Qwen3-235B-A22B-Instruct-25076 | 74.57% | 0.74% | 4.178 | 0.04095 | Yes |
| gpt-oss-120b | 71.40% | gemma-2-27b | 59.84% | 11.57% | 572.581 | 1.541e-126 | Yes |
| gpt-oss-120b | 71.40% | deepseek-v3 | 62.93% | 8.47% | 331.375 | 4.823e-74 | Yes |
| Qwen3-235B-A22B-Instruct-25076 | 74.57% | gemma-2-27b | 59.84% | 14.73% | 953.27 | 2.581e-209 | Yes |
| Qwen3-235B-A22B-Instruct-25076 | 74.57% | deepseek-v3 | 62.93% | 11.63% | 642.934 | 7.69e-142 | Yes |
| deepseek-v3 | 62.93% | gemma-2-27b | 59.84% | 3.10% | 42.443 | 7.276e-11 | Yes |
| gpt-4o | 75.31% | gemma-2-27b | 59.84% | 15.47% | 1045.052 | 2.895e-229 | Yes |
| gpt-4o | 75.31% | deepseek-v3 | 62.93% | 12.37% | 732.145 | 3.061e-161 | Yes |
| gpt-4o | 75.31% | gpt-oss-120b | 71.40% | 3.91% | 97.61 | 5.0939e-23 | Yes |
| Qwen3-235B-A22B-Instruct-25076 | 74.57% | gpt-oss-120b | 71.40% | 3.16% | 64.663 | 8.886e-16 | Yes |
| **Hindi** | | | | | | | |
| gpt-oss-120b | 60.25% | deepseek-v3 | 45.01% | 15.23% | 184.026 | 6.403e-42 | Yes |
| gpt-4o | 65.18% | Qwen3-235B-A22B-Instruct-25076 | 64.13% | 1.05% | 1.658 | 0.1979 | No |
| gpt-4o | 65.18% | gpt-oss-120b | 60.25% | 4.93% | 29.877 | 4.603e-08 | Yes |
| gpt-4o | 65.18% | gemma-2-27b | 50.38% | 14.80% | 196.267 | 1.363e-44 | Yes |
| gpt-4o | 65.18% | deepseek-v3 | 45.01% | 20.17% | 316.898 | 6.864e-71 | Yes |
| Qwen3-235B-A22B-Instruct-25076 | 64.13% | gpt-oss-120b | 60.25% | 3.88% | 17.639 | 2.671e-05 | Yes |
| Qwen3-235B-A22B-Instruct-25076 | 64.13% | gemma-2-27b | 50.38% | 13.75% | 169.093 | 1.167e-38 | Yes |
| Qwen3-235B-A22B-Instruct-25076 | 64.13% | deepseek-v3 | 45.01% | 19.11% | 290.321 | 4.232e-65 | Yes |
| gpt-oss-120b | 60.25% | gemma-2-27b | 50.38% | 9.87% | 88.483 | 5.127e-21 | Yes |
| gemma-2-27b | 50.38% | deepseek-v3 | 45.01% | 5.37% | 22.988 | 1.63e-06 | Yes |

Table 32: Pairwise comparison of the top 5 LLMs on the BBL domain using McNemar's test. Accuracy Diff (%) shows the absolute difference between model accuracies. Significant differences are reported at p < 0.05.

| Model A | Accuracy A | Model B | Accuracy B | Accuracy Diff (%) | McNemar Stat | p-value | Significant |
|---|---|---|---|---|---|---|---|
| **English** | | | | | | | |
| Qwen3-235B-A22B-Instruct-25076 | 80.15% | gpt-4o | 78.83% | 1.32% | 17.392 | 3.041e-05 | Yes |
| gpt-oss-120b | 70.72% | gemma-2-27b | 64.91% | 5.81% | 131.893 | 1.579e-30 | Yes |
| gpt-oss-120b | 70.72% | deepseek-v3 | 67.78% | 2.94% | 53.656 | 2.389e-13 | Yes |
| gpt-4o | 78.83% | gemma-2-27b | 64.91% | 13.92% | 803.654 | 8.661e-177 | Yes |
| gpt-4o | 78.83% | deepseek-v3 | 67.78% | 11.06% | 786.845 | 3.911e-173 | Yes |
| deepseek-v3 | 67.78% | gemma-2-27b | 64.91% | 2.86% | 30.906 | 2.709e-08 | Yes |
| Qwen3-235B-A22B-Instruct-25076 | 80.15% | gemma-2-27b | 64.91% | 15.24% | 965.971 | 4.478e-212 | Yes |
| Qwen3-235B-A22B-Instruct-25076 | 80.15% | deepseek-v3 | 67.78% | 12.38% | 995.052 | 2.137e-218 | Yes |
| Qwen3-235B-A22B-Instruct-25076 | 80.15% | gpt-oss-120b | 70.72% | 9.43% | 643.04 | 7.291e-142 | Yes |
| gpt-4o | 78.83% | gpt-oss-120b | 70.72% | 8.11% | 482.426 | 6.337e-107 | Yes |
| **Hindi** | | | | | | | |
| gpt-oss-120b | 62.94% | deepseek-v3 | 46.78% | 16.17% | 475.374 | 2.16e-105 | Yes |
| gpt-4o | 71.02% | Qwen3-235B-A22B-Instruct-25076 | 68.60% | 2.42% | 19.396 | 1.062e-05 | Yes |
| gpt-4o | 71.02% | gpt-oss-120b | 62.94% | 8.08% | 176.253 | 3.188e-40 | Yes |
| gpt-4o | 71.02% | gemma-2-27b | 51.83% | 19.19% | 543.759 | 2.868e-120 | Yes |
| gpt-4o | 71.02% | deepseek-v3 | 46.78% | 24.24% | 1027.969 | 1.496e-225 | Yes |
| Qwen3-235B-A22B-Instruct-25076 | 68.60% | gpt-oss-120b | 62.94% | 5.66% | 82.242 | 1.204e-19 | Yes |
| Qwen3-235B-A22B-Instruct-25076 | 68.60% | gemma-2-27b | 51.83% | 16.77% | 423.76 | 3.707e-94 | Yes |
| Qwen3-235B-A22B-Instruct-25076 | 68.60% | deepseek-v3 | 46.78% | 21.82% | 859.675 | 5.732e-189 | Yes |
| gpt-oss-120b | 62.94% | gemma-2-27b | 51.83% | 11.11% | 181.188 | 2.666e-41 | Yes |
| gemma-2-27b | 51.83% | deepseek-v3 | 46.78% | 5.06% | 36.78 | 1.322e-09 | Yes |

