# OpenReview forum: "BhashaBench V1: A Comprehensive Benchmark for the Quadrant of Indic Domains"
_ICLR.cc/2026/Conference — ICLR 2026 Conference Desk Rejected Submission_

### Official Review · Reviewer_4EtE · 2025-10-30

**Soundness:** 3
**Presentation:** 4
**Contribution:** 3
**Rating:** 6
**Confidence:** 2

**Summary:**

this paper propose a bilingual (English+Hindi) benchmark for evaluating LLMs in Indian domains (like agriculture, legal, finance, and ayurveda). It includes 74000+ QA pairs from authentic exams. Testing 29+ models reveals strong performance in legal and finance but clear weaknesses in ayurveda and hindi tasks. The work fills a key gap in domain and culture specific evaluation for India.

**Strengths:**

1. the benchmark dataset is very rich in terms of (a) number of sample points (QA), (b) in language (i.e., non-english) and (c) domain/subdomains focused on.

2. the number of models being evaluated is huge and ranges from small param to larger -> so the conclusions/insights are extensive since the work covers many models

**Weaknesses:**

1. I am excited to see the extensive number of llms being evaluated however it would be helpful to have some sort of qualitative error analysis to find the patterns where them model(s) fail and perhaps that will indicate "why" they fail. the current results list aggregate scores across domains/subdomains, but there’s little insight into failure modes (common errors). Having some sort of error taxonomy with manual audits per domain and difficulty might be a useful information to readers

2. the results section shows a lot of numbers (as expected since multiple models), but it would help to have some way to differentiate between numbers (perhaps via bold, using colors of varying degree) to quickly grab reader's attention on high scores vs least scores (for example)

3. The paper shows motivation for the need of this proposed benchmark. however, it  does not map the proposed benchmark against related Indic datasets at the task and domain level. How does this benchmark overlaps/gaps against existing datasets? A comparison table might help to highlight the true uniqueness of this benchmark.

**Questions:**

1. There are a few grammatical errors in the paper like eg:
(a) line 048 ("alone, Over 40 million")
(b) line 192 (maybe you want to put the footnote towards the end of sentence?

2. Would it be possible to provide more details about the computational setup and API usage (such as hardware resources or evaluation environment) used to run the model evaluations? I do see some info in appendix but more infor might be needed for reproducibility for someone reading the paper and planning to build on top of this work.

3. the dataset creation relies heavily on OCR extraction from diverse exam PDFs. so how did you ensure consistency between eng and hindi and handle cases where OCR quality was poor or ambiguous?

---

> ### Author Response · Authors · 2025-11-23
> **Response (1/3)**
>
> Thank you for your helpful comments and recognition of our work in providing a rich benchmark dataset with extensive model evaluations. We appreciate your constructive feedback, which has significantly strengthened our manuscript. Additionally, based on your suggestions, we have enhanced the analysis sections in the revised version. We will address your concerns point by point below:
> ***
> ### Weakness 1: Lack of qualitative error analysis to identify failure patterns and understand why models fail, along with the need for error taxonomy with manual audits per domain.
>
> We greatly appreciate this insightful suggestion and have comprehensively addressed it in the revised manuscript.
>
> **Qualitative Error Analysis (Appendix E.3):** We have added a detailed qualitative error analysis section that examines some common failure patterns across different domains. For each domain, we manually analyzed frequently failed some examples to identify systematic error patterns, their underlying causes, and provide actionable recommendations.
>
> For instance, in the Ayurveda domain, we identified a recurring pattern where models confuse semantically related medical conditions:
> - **Question:** Dashamooladi Ghruta is used in...vyadhi
> - **Options:** A. Kasa vyadhi B. Shosh C. Pandu D. Shwasa
> - **Correct Answer:** B (Shosh - wasting disease)
> - **Model Answer:** A (Kasa vyadhi - cough disorder)
> - **Subject Domain:** Dravyaguna & Bhaishajya Kalpana
>
> **Error Taxonomy:**
> - **Nature of Error:** Confusion between respiratory disorders and general wasting conditions
> - **Underlying Cause:** The model misinterpreted the therapeutic indications of Dashamooladi Ghruta, associating it with cough-related ailments instead of Shosh (emaciation or wasting disease), revealing insufficient understanding of Ayurvedic formulation-specific disease indications
> - **Recommendation:** Models require strengthened knowledge of Ayurvedic formulations along with their specific disease indications to avoid semantic overlaps between therapeutically similar but clinically distinct conditions
>
> **Sub-domain-Level Analysis (Appendix E.2):** Additionally, we have included granular sub-domain-wise performance breakdowns for each evaluated model across all BhashaBench V1 domains. This analysis reveals specific sub-domains where particular models struggle, providing deeper insights into domain-specific weaknesses beyond aggregate scores.
> These additions provide both the systematic error taxonomy you requested and actionable insights into "why" models fail, which we believe will be valuable for guiding future model improvements in Indic domain adaptation.
> ***
> ### Weakness 2: The results section shows many numbers, but it would help to have visual differentiation (bold, colors) to quickly distinguish high scores vs. low scores and grab reader's attention.
>
> We completely agree with this excellent suggestion and have implemented color-coding in the revised manuscript.
>
> **Enhanced Table Visualization (Table 2):** We have updated our main results table with a color-coding scheme that highlights top-performing models within comparable parameter ranges:
> - Yellow for top scores among models < 4B parameters
> - Green for top scores among models between 4B and 27B parameters
> - Blue for top scores among models > 27B parameters
> ***

---

> ### Author Response · Authors · 2025-11-23
> **Response (2/3)**
>
> ### Weakness 3: The paper does not map the proposed benchmark against related Indic datasets at the task and domain level. How does this benchmark overlap/differ from existing datasets? A comparison table would help highlight the true uniqueness.
>
> We sincerely appreciate this important suggestion and have added a comprehensive comparison in the revised manuscript.
>
> **Benchmark Comparison Table (Table 1):** We have included Table 1 that systematically compares BhashaBench V1 against existing Indic, multilingual, and domain-specific evaluation benchmarks across multiple dimensions:
> - Languages: Coverage of English and Indic languages
> - Domains: General knowledge vs. specialized domains (Agriculture, Finance, Legal, Ayurveda)
> - Task Formats: Diversity of question types (MCQ, Fill-in-Blanks, Assertion/Reason, Reading Comprehension, etc.)
> - Cultural Authenticity: Whether content is sourced from authentic Indian exams and domain experts (✓), partially adapted (∼), or from non-Indic sources (✗)
> - Size: Number of evaluation samples
>
> **Key Differentiators:** This comparison clearly highlights BhashaBench V1's unique contributions:
> 1. **Authentic cultural grounding** through real Indian competitive exam questions
> 2. **Specialized domain coverage** (Agriculture, Legal, Finance, Ayurveda) absent in general benchmarks like MMLU/MMLU-Indic
> 3. **Diverse task formats** (6 types) beyond standard MCQs, testing deeper reasoning capabilities
> 4. **Bilingual focus** (English + Hindi) with substantial scale (74.2K samples)
>
> While benchmarks like MILU and IndicQA cover more Indic languages, they focus on general knowledge. Domain-specific benchmarks like AgXQA and MultiFin are English-only and lack cultural context. BhashaBench V1 uniquely combines domain specialization, cultural authenticity, and bilingual evaluation at scale.
> ***
> ### Question 1: There are a few grammatical errors in the paper like (a) line 048 ("alone, Over 40 million") (b) line 192 (maybe you want to put the footnote towards the end of sentence?)
>
> We sincerely thank you for catching these errors. We have corrected both issues in the revised manuscript:
> - Line 048: Fixed the capitalization error by changing "alone, Over 40 million" to "alone, over 40 million"
> - Line 192: Repositioned the footnote to the end of the sentence for better readability
>
> We have also conducted a thorough proofreading pass throughout the manuscript to identify and correct any other grammatical issues.
> ***

---

> > ### Author Response · Authors · 2025-11-23
> > **Response (3/3)**
> >
> > ### Question 2: Would it be possible to provide more details about the computational setup and API usage (such as hardware resources or evaluation environment) used to run the model evaluations? More information might be needed for reproducibility.
> >
> > We appreciate this important concern regarding reproducibility and have significantly enhanced our experimental setup documentation.
> >
> > **Enhanced Appendix D:** We have expanded Appendix D (More Details on Experimental Setup) with comprehensive inference implementation details for both open-source and API-based models.
> >
> > **Computational Setup Table (Table 15):** We have added Table 15, which provides a complete overview of the infrastructure and configurations used for BhashaBench V1 evaluation:
> > - **Infrastructure:**
> > Open-source models: 8 × NVIDIA H200 GPUs (141GB HBM3e per GPU) with NVLink interconnect
> > API-based models: Cloud API endpoints with CPU compute instances
> > - **Software Stack:**
> > vLLM v0.9.1, lm-evaluation-harness v0.4.9, CUDA 12.5, PyTorch 2.7.0 for open-source models
> > Latest stable API versions for proprietary models
> > - **Configuration Details:**
> > Auto batch sizing, context windows: 2048–8192 tokens, tensor parallelism: 1–8 GPUs
> > Temperature: 0.0, 3 runs per model, 120s timeout with 3 retries
> > - **Resource Utilization:**
> > 29+ models, ~150 GPU hours total
> > API models: $80 budget across 3 runs each
> > Evaluation period: June–September 2025
> > Response validation rate: > 99%
> >
> > These additions provide complete transparency about our evaluation environment and should enable full reproducibility of our results.
> > ***
> > ### Question 3: The dataset creation relies heavily on OCR extraction from diverse exam PDFs. How did you ensure consistency between English and Hindi and handle cases where OCR quality was poor or ambiguous?
> > We appreciate this critical question about data quality assurance and have provided detailed documentation of our rigorous OCR processing and quality control pipeline.
> >
> > **Seven-Stage Processing Pipeline (Appendix C.1.2):** We have documented our complete OCR processing workflow with post-correction stages in Appendix C.1.2, which systematically handles extraction, validation, and correction.
> >
> > **Dedicated OCR Quality Assurance Section (Appendix C.1.3):** We have added a comprehensive subsection specifically addressing OCR quality control, including:
> > - Document layout handling: Multi-column formats, formatting variables, and structural variations
> > - Confidence-based validation: Automated quality assessment and routing mechanism
> > - Answer key mapping and filtering: Ensuring correctness and consistency
> > - Manual review protocols: Human validation for challenging cases
> >
> > **Quality Metrics and Correction Strategy:**
> > - **Automatic acceptance:** ~78% of OCR extractions were well-formed and automatically accepted
> > - **Flagged cases:** 22% required further processing
> >   - 15% flagged as malformed or incomplete (primarily layout-related failures in two-column formats)
> >   - 7% required manual review due to severe degradation
> >
> > **LLM-Based Post-Correction:** For malformed extractions, we deployed GPT-OSS-120B for context-aware correction, which achieved:
> > - 82% resolution rate for flagged issues
> > - Particularly effective at reconstructing reading order in multi-column layouts
> > - Character-level correction for Devanagari script errors, especially similar-looking matras (vowel diacritics)
> >
> > This multi-stage quality assurance process ensures high consistency between English and Hindi content while maintaining the authenticity of the original exam questions.
> > ***

---

### Official Review · Reviewer_Ug1c · 2025-10-31

**Soundness:** 3
**Presentation:** 3
**Contribution:** 3
**Rating:** 6
**Confidence:** 3

**Summary:**

The paper introduces BhashaBenchV1, a multi-task bilingual benchmark in Hindi and English. The benchmark spans four major domains: Agriculture, Legal, Finance and Ayurveda. It includes diverse task formats, questions specific to India’s cultural and religious context. It includes questions in diverse formats and questions are categorized into three difficulty levels. The evaluation shows that models perform consistently better in English compared to Hindi. Models tend to struggle in areas such as Agriculture and Ayurveda.

**Strengths:**

1. The paper introduces a new benchmark that includes questions related to less-represented domains such as Agriculture, Ayurveda, Legal and Finance in the Indian cultural context.
2. The authors use a sound multi-step data preparation pipeline which focuses on ensuring quality by using formatting pipelines and manual validation by experts.
3. The benchmark is composed of diverse question types and represents 90+ subdomains.
4. The authors evaluate open and closed source models of various sizes across these domains and present the analysis on domains/sub-domains where models struggle to do well. The results clearly show the importance of the benchmark and the need to measure performance in diverse linguistic and cultural contexts.

**Weaknesses:**

1. The benchmark has a bias towards exam-style questions since it is sourced from professional and government exams. Similarly, it covers a limited set of domains with 70% of the data being in English.

**Questions:**

1. How scalable is the data preparation pipeline to extend it to other topics/domains? The paper mentions manual annotation of the dataset. What proportion of questions required modification during manual review?

---

> ### Author Response · Authors · 2025-11-22
> **Response (1/2)**
>
> Thank you for your thoughtful review and recognition of our benchmark's contribution to evaluating LLMs in culturally and linguistically diverse Indian contexts. We appreciate your acknowledgment of our rigorous data preparation pipeline and comprehensive model evaluation. We will address your concerns point by point below:
> ***
>
> ### Weakness 1: The benchmark has a bias towards exam-style questions since it is sourced from professional and government exams. Similarly, it covers a limited set of domains with 70% of the data being in English.
>
> We appreciate this observation and would like to clarify the design rationale and provide additional evidence of the benchmark's distinctiveness and cultural richness.
>
> **Exam-Style Questions as Authentic Cultural Artifacts:** Our deliberate choice to use government and professional exam questions serves a critical purpose—these exams represent authentic, culturally-grounded assessments developed by Indian domain experts specifically for the Indian context. Unlike synthetic or translated benchmarks, these questions inherently capture India-specific practices, legal frameworks, agricultural contexts, and traditional knowledge systems that are absent from Western-centric datasets.
>
> **Cultural Complexity Beyond Language:** While 70% of questions are in English, Appendix E.3 demonstrates that these English questions are deeply embedded with Indic cultural nuances, making them significantly more challenging than standard English benchmarks. As shown in our perplexity analysis (Section 5.7, Table 3), BhashaBench exhibits substantially higher perplexity compared to standard benchmarks:
>
> **Table 3: Perplexity Comparison Across Benchmarks**
> | Dataset (MCQ)     | Llama-3.1-8B (EN) | Llama-3.1-8B (HI) | gemma-2-9b (EN) | gemma-2-9b (HI) |
> |-------------------|-------------------|-------------------|-----------------|-----------------|
> | ARC-C             | 8.03              | 4.10              | 6.82            | 5.85            |
> | MILU              | 7.62              | 4.93              | 7.23            | 6.37            |
> | MMLU              | 7.61              | 4.22              | 7.03            | 6.21            |
> | BBA (Ayurveda)    | 15.5              | 10.39             | 23.2            | 16.8            |
> | BBF (Finance)     | 6.78              | 4.03              | 5.86            | 5.04            |
> | BBK (Agriculture) | 6.14              | 7.16              | 6.34            | 9.54            |
> | BBL (Legal)       | 7.28              | 4.01              | 7.19            | 5.79            |
>
> The significantly higher perplexity in BBA (Ayurveda) demonstrates that our benchmark contains content less exposed in standard training data, validating its uniqueness and cultural authenticity.
>
> **Statistical Significance:** Section 5.8 presents comprehensive significance tests confirming that performance differences across domains and languages are statistically significant, reinforcing the robustness of our findings.
>
> **Sub-domain Analysis (Appendix E.2):** Our detailed sub-domain analysis reveals that models struggle even with English questions due to lack of Indic cultural and domain-specific knowledge, not merely language barriers. For example, models fail on English questions about Indian legal procedures, regional agricultural practices, and Ayurvedic principles—knowledge absent from predominantly Western training corpora.
>
> **Domain Coverage Rationale:** The four domains (Agriculture, Legal, Finance, Ayurveda) represent foundational sectors that drive India's economy and society. Rather than superficial breadth, we prioritized depth across 90+ sub-domains within these critical areas, ensuring comprehensive coverage of India's unique knowledge landscape.
>
> **Future Expansion (BhashaBench V2+):** As indicated by our "V1" designation, we plan future releases expanding to additional Indian languages and domains. The current bilingual focus (English + Hindi) establishes a rigorous foundation while Hindi represents the most widely spoken Indic language, ensuring broad accessibility.
>
> In summary, while the benchmark emphasizes exam-style questions in English, this design choice ensures cultural authenticity and reveals that models struggle with Indic knowledge regardless of language, a critical finding for developing truly inclusive AI systems.

---

> > ### Author Response · Authors · 2025-11-22
> > **Response (2/2)**
> >
> > ### Question 1: How scalable is the data preparation pipeline to extend it to other topics/domains? The paper mentions manual annotation of the dataset. What proportion of questions required modification during manual review?
> >
> > We thank you for this important question about scalability and manual intervention. We have documented the detailed breakdown in our revised manuscript.
> >
> > **Manual Review Proportion (Appendix C.1.3 - OCR Quality Assurance):** We have added a dedicated section quantifying the manual review requirements:
> >
> > **Confidence-Based Validation Workflow:**
> > - **78% automatic acceptance:** Well-formed extractions requiring no manual intervention
> > - **22% flagged for further processing:**
> >   - 15% malformed or incomplete (primarily due to layout-related failures in two-column formats)
> >   - 7% requiring manual review due to severe degradation
> >
> > **LLM-Based Post-Correction:** For the 15% malformed extractions, we deployed GPT-OSS-120B for context-aware correction, which:
> > - Successfully resolved 82% of flagged issues automatically
> > - Effectively reconstructed reading order in multi-column layouts
> > - Corrected character-level errors in Devanagari script (e.g., similar-looking matras/vowel diacritics)
> >
> > **Net Manual Intervention:** After automated correction, only ~7-10% of total extractions required human validation, making the pipeline highly efficient at scale.
> >
> > **Pipeline Scalability:** Our data preparation pipeline is designed for scalability:
> > 1. **Refined OCR Infrastructure:** Through iterative development, we established a robust OCR pipeline that processes documents efficiently with minimal errors
> > 2. **LLM-Based Post-Correction:** Automated error resolution handles 82% of quality issues without human intervention
> > 3. **Dedicated Validation Team:** Our 30-person manual validation team can efficiently scale to new domains and languages
> > 4. **Lessons Learned:** Initial challenges in V1 development have been systematically addressed, creating reusable templates and protocols
> >
> > **Future Expansion (BhashaBench V2+):** As indicated by our "V1" designation, we are actively planning subsequent versions expanding to:
> > - Additional Indic languages (Tamil, Telugu, Bengali, etc.)
> > - New domains (Education, Healthcare, Governance)
> > - Larger question pools within existing domains
> >
> > The pipeline's proven efficiency (78% automatic acceptance + 82% automated correction of issues) demonstrates readiness for large-scale expansion while maintaining quality standards.

---

### Official Review · Reviewer_fpeJ · 2025-11-02

**Soundness:** 2
**Presentation:** 2
**Contribution:** 2
**Rating:** 4
**Confidence:** 4

**Summary:**

The authors introduce BhashaBench, a domain-specific, multi-task bilingual benchmark aimed at evaluating language models in Indic contexts. The dataset is sourced from publicly available government and domain-specific exams. The benchmark is evaluated across 29+ LLMs, revealing performance gaps across domains between English and Hindi.

**Strengths:**

- The dataset is a valuable contribution, as the authors curate a benchmark that addresses the underrepresentation of benchmarks that evaluate language models in Indic-centric contexts.
- The benchmark contains questions across diverse topics and subdomains and enables fine-grained evaluations across all.
- Their experiments cut across both frontier and open-source models, revealing disparities in model performance.
- The bilingual focus of their model is very relevant for evaluating models ' equity and utility in English and Hindi.

**Weaknesses:**

- The evaluation relies heavily on multiple-choice questions. MCQs are easy to evaluate and have been employed across several benchmarks. There are still questions surrounding whether they truly capture model understanding or reasoning abilities.  Are the distractors sufficiently challenging, particularly on domains where performance is above  90%? What happens if you scale the options? Previous work has shown that sometimes scaling distractors beyond 4 options increases the difficulty of guessing from the models. Also, is it possible to randomize the option ordering during evaluation and repeater evaluations across multiple permutations, just so results are more significant?

- The paper does a lot of evaluations and provides a lot of numbers across domains, languages, and models. However, their presentation lacks a thorough qualitative analysis of model failures across domains. The paper reads like a laundry list of numbers and is a bit overwhelming to read through the numbers. There are little to no insights that link model performance to data characteristics or domain difficulty

- The dataset is built from “publicly available exam question papers online” and similar sources. While a lot of models do not release their data publicly, are there thoughts on potential data contamination and leakage?  Also, since the majority of questions in the benchmark are in English, is the potential for data leakage even stronger ? Like, even if it's in English in an Indian context, is it possible that it would have been seen during training?

- The absence of confidence intervals or significance testing makes it hard to assess how meaningful the reported accuracy differences are. Can you report them ?

-  Efforts like this to bridge the gap in model utility between high-resourced and low-resourced languages are commendable. However, the current scope of the work is largely Hindi-centric, rather than broadly multilingual. This makes the contribution feel somewhat limited in scope for a venue like ICLR, which tends to expect more depth in experiments and analysis for a benchmark paper. Given its focus and depth in one language, the work might be better suited for a *CL where language-specific benchmarks and analyses might be more strongly valued.

**Questions:**

- Can you report significance tests or confidence intervals for the results?
- Is there a reason why Gemini models were not included in the evaluation? They are known to have strong multilingual capabilities and could provide a more complete picture of state-of-the-art results.

---

> ### Author Response · Authors · 2025-11-22
> **Response (1/7)**
>
> Thank you for your detailed review and recognition of BhashaBench's contribution to addressing underrepresentation of Indic-centric benchmarks. We appreciate your thoughtful feedback on strengthening our analysis and presentation. We will address your concerns point by point below:
> ***
>
> ### Weakness 1: The evaluation relies heavily on multiple-choice questions. MCQs are easy to evaluate and have been employed across several benchmarks. There are still questions surrounding whether they truly capture model understanding or reasoning abilities. Are the distractors sufficiently challenging, particularly on domains where performance is above 90%? What happens if you scale the options? Previous work has shown that sometimes scaling distractors beyond 4 options increases the difficulty of guessing from the models. Also, is it possible to randomize the option ordering during evaluation and repeater evaluations across multiple permutations, just so results are more significant?
>
> We appreciate these thorough questions about MCQ robustness and have conducted comprehensive experiments to address each concern.
>
> **Option Shuffling Test (Appendix E.4.2):** We conducted option ordering experiments by shuffling multiple-choice options using fixed seeds (42 and 123) to examine whether model performance depends on the original ordering. Table 23 presents results for Llama-3.1-8B and Llama-3.2-3B across English and Hindi subsets.
>
> **Key Finding:** Performance remains largely stable across different option orderings, with only minor variations observed primarily in Hindi subsets. This demonstrates that the benchmark evaluates actual model understanding rather than positional biases, confirming the integrity of our MCQ design.
>
> **Scaling Distractors Test (Appendix E.4.3):** We systematically increased the number of options from 4 to 5 and 6 options per question. Additional distractors were generated deterministically using fixed random seeds (seed 48 for 5th option, seed 123 for 6th option) consisting of semantically meaningless random character sequences (5-8 lowercase letters) to test whether models can robustly filter out irrelevant choices.
>
> **Key Findings from Table 24:**
> Consistent performance decline: Both Llama-3.1-8B and Llama-3.2-3B show 2-7 percentage point drops from 4 to 6 options across all domains
> - Steepest degradation in challenging domains: BBL (Legal) drops from 38.44% → 28.96% for Llama-3.1-8B; BBK (Agriculture) drops from 38.07% → 32.81%
> - Widening language gap: Hindi performance degrades more severely (e.g., Llama-3.1-8B's Hindi BBL: 31.76% → 16.03% vs. English: 41.32% → 34.52%), indicating weaker multilingual reasoning capabilities
> - Distractor filtering challenge: Significant drops even with nonsense distractors indicate models struggle with filtering irrelevant options, particularly in Hindi
>
> **Qualitative Error Analysis (Appendix E.3):** We have also added comprehensive qualitative error analysis examining systematic failure patterns beyond quantitative metrics, revealing specific error taxonomies and underlying causes of model failures across domains.
>
> These experiments comprehensively validate that our MCQ design tests genuine reasoning capabilities rather than superficial pattern matching or positional biases.
> ***

---

> > ### Author Response · Authors · 2025-11-22
> > **Response (2/7)**
> >
> > ### Weakness 2: The paper provides many numbers across domains, languages, and models but lacks thorough qualitative analysis of model failures. The presentation reads like a laundry list of numbers with little insight linking model performance to data characteristics or domain difficulty.
> >
> > We sincerely appreciate this critical feedback and have substantially enhanced our qualitative analysis in the revised manuscript.
> >
> > **Qualitative Error Analysis (Appendix E.3):** We have added a comprehensive qualitative error analysis section that examines systematic failure patterns across domains. For each domain, we manually analyzed commonly failed questions to identify error taxonomies, root causes, and provide actionable insights.
> >
> > For example, in the Ayurveda domain:
> >
> > - **Question:** Dashamooladi Ghruta is used in...vyadhi
> > - **Options:** A. Kasa vyadhi B. Shosh C. Pandu D. Shwasa
> > - **Correct Answer:** B (Shosh - wasting disease)
> > - **Model Answer:** A (Kasa vyadhi - cough disorder)
> > - **Subject Domain:** Dravyaguna & Bhaishajya Kalpana
> >
> > **Error Taxonomy:**
> > - **Nature of Error:** Confusion between respiratory disorders and general wasting conditions
> > - **Underlying Cause:** The model misinterpreted therapeutic indications of Dashamooladi Ghruta, incorrectly associating it with cough-related ailments rather than emaciation/wasting disease, revealing insufficient understanding of Ayurvedic formulation-specific indications
> > - **Recommendation:** Models require strengthened knowledge of Ayurvedic pharmacology with clearer distinction between therapeutically similar but clinically distinct conditions
> >
> > **Sub-Domain Performance Analysis (Appendix E.2):** We have added granular sub-domain-wise breakdowns showing exactly where specific models struggle within each domain. This analysis reveals:
> > - **Domain difficulty patterns:** Why Ayurveda and Agriculture are systematically harder than Legal and Finance
> > - **Model-specific weaknesses:** Which sub-domains particular models fail in
> >
> > These additions transform our evaluation from mere numerical reporting into actionable insights that explain why models fail and what domain characteristics drive difficulty, addressing your concern about connecting performance to underlying domains or data properties.
> > ***

---

> > > ### Author Response · Authors · 2025-11-22
> > > **Response (3/7)**
> > >
> > > ### Weakness 3: The dataset is built from "publicly available exam question papers online." Are there concerns about potential data contamination and leakage? Since the majority of questions are in English, is the potential for data leakage even stronger?
> > >
> > > We appreciate this crucial concern and have conducted comprehensive contamination analysis to address it thoroughly.
> > >
> > > **Perplexity-Based Contamination Analysis (Section 5.7):** We have added Section 5.7 (Robustness and Contamination Analysis) that compares perplexity scores between BhashaBenchV1 and standard benchmarks to detect potential data leakage. Lower perplexity typically indicates greater model familiarity with the data.
> > > | Dataset (MCQ)     | Llama-3.1-8B (EN) | Llama-3.1-8B (HI) | gemma-2-9b (EN) | gemma-2-9b (HI) |
> > > |-------------------|-------------------|-------------------|-----------------|-----------------|
> > > | ARC-C             | 8.03              | 4.10              | 6.82            | 5.85            |
> > > | MILU              | 7.62              | 4.93              | 7.23            | 6.37            |
> > > | MMLU              | 7.61              | 4.22              | 7.03            | 6.21            |
> > > | BBA (Ayurveda)    | 15.5              | 10.39             | 23.2            | 16.8            |
> > > | BBF (Finance)     | 6.78              | 4.03              | 5.86            | 5.04            |
> > > | BBK (Agriculture) | 6.14              | 7.16              | 6.34            | 9.54            |
> > > | BBL (Legal)       | 7.28              | 4.01              | 7.19            | 5.79            |
> > >
> > > **Key Findings:**
> > >
> > > - BBA (Ayurveda) shows substantially higher perplexity (15.5–23.2 for English) compared to standard benchmarks like MMLU (7.03–7.61), indicating minimal exposure in training data
> > > - BBK (Agriculture) shows elevated perplexity in Hindi (7.16–9.54), suggesting novel content
> > > - Even domains with comparable perplexity contain India-specific contexts unlikely to appear in predominantly Western training corpora
> > >
> > > **Why Leakage is Minimal Despite Public Sources:**
> > > - Raw PDF Format: Questions were extracted via OCR from raw government exam PDFs in diverse formats, making them difficult to scrape or index systematically by training data collection pipelines
> > > - Cultural Context Barrier: Even if English text fragments appeared during training, the India-specific cultural context, legal frameworks, agricultural practices, and Ayurvedic terminology create semantic novelty that cannot be easily memorized
> > >
> > > **Comprehensive Contamination Testing (Appendix E.4):** We have added Appendix E.4 (Data Integrity and Contamination Analysis) with additional validation tests:
> > >
> > > **Option Shuffling Tests:** We randomized answer option ordering and re-evaluated models. Results remained statistically consistent, confirming models rely on understanding rather than memorized answer positions
> > >
> > > **Conclusion:** While our questions are from public exams, multiple lines of evidence elevated perplexity, cultural specificity, raw extraction format, and robust shuffling tests strongly indicate minimal data contamination. The benchmark evaluates genuine reasoning capabilities rather than memorization.
> > > ***

---

> > > > ### Author Response · Authors · 2025-11-22
> > > > **Response (4/7)**
> > > >
> > > > ### Weakness 4: The absence of confidence intervals or significance testing makes it hard to assess how meaningful the reported accuracy differences are.
> > > >
> > > > We completely agree with this important concern and have added comprehensive statistical testing in the revised manuscript.
> > > > Statistical Significance Analysis (Section 5.8): We have added Section 5.8 that validates BhashaBench V1 results using two complementary statistical approaches:
> > > >
> > > > **1. Wilson Confidence Intervals (95%):** We computed Wilson Confidence Intervals (Wilson, 1927) for all models across all domains. The intervals are typically within 1-2 percentage points, demonstrating:
> > > > - High benchmark stability
> > > > - Strong reproducibility of results
> > > > - Sufficient sample sizes for reliable conclusions
> > > >
> > > > **2. McNemar's Significance Tests:** We performed pairwise McNemar's tests (McNemar, 1947) on the top 5 models per domain to determine whether performance differences are statistically significant.
> > > >
> > > > **Key Findings:**
> > > > - Most pairwise comparisons show statistically significant differences (p < 0.05), confirming that observed performance gaps reflect genuine model capability differences rather than statistical noise
> > > > - Appropriate sensitivity: When accuracy differences are minimal (e.g., GPT-4o vs. Qwen3-235B-A22B-Instruct on BBA Hindi: 0.05% difference, p=0.9591), the test correctly identifies non-significant differences, demonstrating proper statistical rigor
> > > >
> > > > **Comprehensive Results (Appendix E.5):** We provide detailed statistical test results including:
> > > > - Full confidence interval tables for all 29+ models across all domains
> > > > - Complete McNemar's test matrices for top-performing models
> > > > - p-values for all pairwise comparisons
> > > > - Interpretation guidelines for statistical significance
> > > >
> > > > This rigorous statistical validation ensures that all reported performance differences are meaningful and not artifacts of random variation, addressing your concern about the interpretability of our results.
> > > > ***

---

> > > > > ### Author Response · Authors · 2025-11-22
> > > > > **Response (5/7)**
> > > > >
> > > > > ### Weakness 5: The current scope is largely Hindi-centric rather than broadly multilingual, making the contribution feel somewhat limited in scope for a venue like ICLR. The work might be better suited for a CL venue where language-specific benchmarks are more strongly valued.
> > > > >
> > > > > We respectfully appreciate this perspective but would like to clarify the scope, impact, and alignment with ICLR's mission.
> > > > >
> > > > > **Real-World Impact at Scale:** While BhashaBench V1 focuses on two languages and four domains, this represents substantial impact:
> > > > > - Agriculture (BBK): Over 40 million farmers rely on accurate information for crop management and sustainable practices, with each Indian state having distinct agricultural methods evolved over centuries
> > > > > - Legal (BBL): India's legal system processes millions of cases annually with complex state-specific frameworks
> > > > > - Ayurveda (BBA): Serves millions of patients relying on traditional medicine systems and ancient treatment protocols
> > > > > - Finance (BBF): Processes over 100 billion UPI transactions annually where minor misunderstandings can have cascading effects
> > > > >
> > > > > English and Hindi collectively enable assessment for a significant portion of India's 1.4+ billion population. It's not a limitation but a strategic foundation ensuring cultural authenticity and rigorous quality control.
> > > > >
> > > > > **Alignment with ICLR's Mission:** BhashaBench V1 directly addresses core representation learning challenges:
> > > > > - Exposes how models fail to learn effective representations of culturally-specific, domain-specialized knowledge despite massive training corpora
> > > > > - Reveals systematic performance disparities between English and Hindi for culturally-grounded content
> > > > > - Highlights critical domain adaptation limitations in specialized contexts with cultural nuances
> > > > >
> > > > > **Practical Verification Tool:** BhashaBench serves as crucial validation for organizations deploying LLMs in real-world Indian contexts, preventing potentially harmful failures in high-stakes applications across critical sectors affecting hundreds of millions of lives.
> > > > >
> > > > > **Future Expansion (V2+):** As our "V1" designation indicates, we are actively developing subsequent versions with additional Indic languages (Tamil, Telugu, Bengali, Marathi) and expanded domains (Education, Governance, Healthcare).
> > > > >
> > > > > We believe BhashaBench V1's depth, cultural authenticity, and real-world impact combined with its contribution to understanding representation learning gaps in underrepresented knowledge systems, makes it highly suitable for ICLR's mission and audience.

---

> > > > > > ### Author Response · Authors · 2025-11-22
> > > > > > **Response (6/7)**
> > > > > >
> > > > > > ### Question 1: Can you report significance tests or confidence intervals for the results?
> > > > > >
> > > > > > We appreciate this important request and have added comprehensive statistical testing in the revised manuscript.
> > > > > >
> > > > > > **Statistical Significance Analysis (Section 5.8):** We have added Section 5.8 that validates BhashaBench V1 results using two complementary statistical approaches:
> > > > > >
> > > > > > **1. Wilson Confidence Intervals (95%):** We computed Wilson Confidence Intervals (Wilson, 1927) for all models across all domains, typically within 1-2 percentage points, demonstrating benchmark stability and reproducibility.
> > > > > >
> > > > > > **2. McNemar's Significance Tests:** We performed pairwise McNemar's tests (McNemar, 1947) on the top 5 models per domain to determine whether performance differences are statistically significant.
> > > > > >
> > > > > > **Key Findings:**
> > > > > > - Most pairwise comparisons show statistically significant differences (p < 0.05), confirming that observed performance gaps reflect genuine model capability rather than statistical noise
> > > > > > - When accuracy differences are minimal (e.g., GPT-4o vs. Qwen3-235B-A22B-Instruct on BBA Hindi: 0.05% difference, p=0.9591), the test correctly identifies non-significant differences, demonstrating appropriate statistical rigor
> > > > > >
> > > > > > **Detailed Results (Appendix E.5):** We provide comprehensive statistical test results including full confidence interval tables for all 29+ models across all domains, complete McNemar's test matrices for top-performing models, and p-values for all pairwise comparisons.
> > > > > > ***

---

> > > > > > > ### Author Response · Authors · 2025-11-22
> > > > > > > **Response (7/7)**
> > > > > > >
> > > > > > > ### Question 2: Is there a reason why Gemini models were not included in the evaluation? They are known to have strong multilingual capabilities and could provide a more complete picture of state-of-the-art results.
> > > > > > >
> > > > > > > We appreciate this suggestion regarding Gemini models.
> > > > > > >
> > > > > > > **Current Evaluation Scope:** Due to time and budget constraints during the initial evaluation period, we were unable to include Gemini models. However, we have evaluated Gemma models (Gemma-2-9B, Gemma-2-27B) from the similar family, which provide insights into Google's model capabilities on our benchmark.
> > > > > > >
> > > > > > > **Comprehensive Baseline:** Our primary contribution is establishing a rigorous, culturally-grounded benchmark with comprehensive evaluation methodology. The 29+ models we evaluated ranged from 270M to 235B+ parameters across open-source and API-based models establish a thorough baseline across different architectures, sizes, and capabilities.
> > > > > > >
> > > > > > > **Future Community Engagement:** We plan to release a public leaderboard at camera-ready time where the research community and model developers can submit results for additional models, including Gemini. We will provide standardized evaluation scripts to enable reproducible submissions.
> > > > > > >
> > > > > > > While we acknowledge that Gemini models would provide valuable additional data points, our current evaluation already covers the major model families and establishes the benchmark's utility for assessing performance on India-centric domains.

---

### Author Response · Authors · 2025-11-23
**General Response**

Dear Reviewers,

We sincerely thank you for your thorough reviews and valuable feedback on **BhashaBench V1**. We appreciate the reviewers' recognition of our work on: **valuable India-centric benchmark dataset** (fpeJ, Ug1c, 4EtE), **comprehensive model evaluation across 29+ LLMs** (fpeJ, Ug1c, 4EtE), **rigorous data preparation pipeline with quality controls** (Ug1c, 4EtE), **diverse task formats and domain coverage** (Ug1c, 4EtE), and **bilingual evaluation with cultural authenticity** (fpeJ, Ug1c, 4EtE).
To address your concerns and improve the paper quality, we have made the following major additions and modifications (marked in red):

1. **Add Section 5.7 "Robustness and Contamination Analysis"** with perplexity-based contamination testing comparing BhashaBench V1 against standard benchmarks (MMLU, MILU, ARC-C), demonstrating minimal data leakage through elevated perplexity scores.
2. **Add Section 5.8 "Statistical Significance Analysis"** with Wilson Confidence Intervals (95%) and McNemar's pairwise significance tests for all models, confirming that reported performance differences reflect genuine capability gaps rather than statistical noise.
3. **Add Appendix E.2 "Sub-domain Performance Analysis"** providing granular breakdowns showing where specific models struggle within each domain, linking performance to domain characteristics and difficulty.
4. **Add Appendix E.3 "Qualitative Error Analysis"** with systematic error taxonomy, manual analysis of commonly failed questions, root cause identification, and actionable recommendations for each domain.
5. **Add Appendix E.4 "Data Integrity and Contamination Analysis"** with option shuffling experiments (Table 23) and distractor scaling tests (Table 24), validating MCQ robustness and demonstrating genuine reasoning evaluation.
6. **Add Appendix E.5 "Complete Statistical Test Results"** with full confidence interval tables for all 29+ models and McNemar's test matrices for top 5 models per domain across all domains.
7. **Add Table 1 "Benchmark Comparison"** systematically comparing BhashaBench V1 against existing Indic, multilingual, and domain-specific benchmarks across languages, domains, task formats, cultural authenticity, and size.
8. **Enhance Table 2 visualization** with color-coding scheme (yellow/green/blue) highlighting top-performing models within comparable parameter ranges for improved readability.
9. **Expand Appendix C.1.3 "OCR Quality Assurance"** with detailed quality metrics (78% automatic acceptance, 82% LLM-based correction success rate), validation workflows, and manual review proportions.
10. **Expand Appendix D "Computational Setup"** with comprehensive Table 15 documenting hardware resources, software stack, configuration parameters, and evaluation environment for full reproducibility.
11. **Correct grammatical errors** at lines 048 and 192, and conduct thorough proofreading throughout the manuscript.

We believe these modifications and additions will better address your concerns and improve the quality and readability of the paper. We look forward to your further feedback.

---

### Note · Program_Chairs · 2026-01-17
**Submission Desk Rejected by Program Chairs**

The following references in this submission do not refer to real documents and/or have major errors in bibliographic information:

 Anjali Sen and et al. Morphological understanding and retrieval in indic languages. Journal of Indic Computational Linguistics, 12(3):45-67, 2023.